# Diffusion Transformers for Imputation: Statistical Efficiency and Uncertainty Quantification

**Zeqi Ye**
Northwestern University
Evanston, IL, USA
zeqiye2029@u.northwestern.edu

**Minshuo Chen**
Northwestern University
Evanston, IL, USA
minshuo.chen@northwestern.edu

## Abstract

Imputation methods play a critical role in enhancing the quality of practical time-series data, which often suffer from pervasive missing values. Recently, diffusion-based generative imputation methods have demonstrated remarkable success compared to autoregressive and conventional statistical approaches. Despite their empirical success, the theoretical understanding of how well diffusion-based models capture complex spatial and temporal dependencies between the missing values and observed ones remains limited. Our work addresses this gap by investigating the statistical efficiency of conditional diffusion transformers for imputation and quantifying the uncertainty in missing values. Specifically, we derive statistical sample complexity bounds based on a novel approximation theory for conditional score functions using transformers, and, through this, construct tight confidence regions for missing values. Our findings also reveal that the efficiency and accuracy of imputation are significantly influenced by the missing patterns. Furthermore, we validate these theoretical insights through simulation and propose a mixed-masking training strategy to enhance the imputation performance.

## 1 Introduction

Sequential data are ubiquitous in real-world applications such as finance [John et al., 2019, Chen et al., 2016], healthcare [Tonekaboni et al., 2021, Kazijevs and Samad, 2023], transportation [Li et al., 2020, Tedjopurnomo et al., 2020], and meteorology [Yozgatligil et al., 2013]. However, these datasets often suffer from missing values due to factors such as sensor malfunctions, data transmission errors, and human oversight [Greco et al., 2012, Yi et al., 2016]. Missing data can significantly degrade the performance of downstream tasks [Ribeiro and Castro, 2022, Alwateer et al., 2024], making accurate and robust imputation a critical challenge.

One of the earliest imputation methods dates back to Allan and Wishart [1930], which provided formulas for estimating single missing observations. Over the past century, this foundational idea of imputation has been extended to broader application domains. Statistical imputation methods have gained sustained attention due to their computational efficiency and ease of implementation. These approaches range from simple techniques, such as imputation using the mean or median of observations, to interpolation-based methods [Tukey, 1952], and more sophisticated model-based techniques, including Kalman filters and autoregressive models [Gómez and Maravall, 1994, Shumway et al., 2000]. However, these methods often rely on strong assumptions such as linearity and stationarity, which may not hold in complex real-world scenarios, thereby limiting their applicability and accuracy [Fuller, 2009].

To address the limitations of statistical methods, recent research has increasingly turned to machine learning approaches for imputation. These methods are capable of capturing complex spatio-temporal patterns and nonlinear dependencies without requiring strict assumptions [Fang and Wang, 2020]. Typical examples include training neural networks such as recurrent neural networks and transformer

39th Conference on Neural Information Processing Systems (NeurIPS 2025).

architectures for inferring missing values [Wang et al., 2024]. In parallel, generative models such as Variational AutoEncoders (VAEs) and Generative Adversarial Networks (GANs) have shown promise by introducing uncertainty-aware imputations [Fortuin et al., 2020, Miao et al., 2021]. However, these generative models often spell limitations in expressiveness or training stability. More recently, diffusion-based generative models have emerged as a powerful alternative, offering robust imputations and strong empirical performance across diverse and high-dimensional time series datasets [Tashiro et al., 2021, Zhou et al., 2024].

Despite their widespread empirical success, diffusion-based imputation methods exhibit two key challenges. First, their performance is highly sensitive to dataset characteristics, often displaying substantial variability across benchmarks [Zhang et al., 2024, Zheng and Charoenphakdee, 2022, Tashiro et al., 2021]. Second, they are significantly affected by missing patterns, leading to inconsistencies in imputation quality [Zhang et al., 2024, Ouyang et al., 2023, Zhou et al., 2024]. These observations motivate the following fundamental questions:

*How well can diffusion models capture the underlying conditional distribution of missing values?*

*How does the missing pattern affect the imputation performance?*

In this paper, we answer the two questions from a statistical learning perspective. Our analysis centers on Diffusion Transformers (DiT, Peebles and Xie [2022]) applied to imputation tasks with Gaussian process (GP) data. Despite their conceptual simplicity, GPs exhibit rich spatio-temporal dependencies and long-horizon dependencies that pose challenges for modeling and imputation. On the other hand, GPs are powerful statistical tools widely used in regression, classification, and forecasting tasks [Seeger, 2004, Banerjee et al., 2013, Borovitskiy et al., 2021].

We establish sample complexity bounds for DiTs in learning the underlying conditional distribution of missing values given observed ones. The obtained bounds demonstrate the role of missing patterns in imputation performance, highlighting how the condition number of the covariance matrix for the missing values and distribution shifts contribute to variability in accuracy. Furthermore, we derive confidence intervals for imputed values and show the coverage probability of them converging to the desired level. We summarize our contributions as follows.

- **Statistical Efficiency**. We show that DiTs capture the conditional distribution of missing values effectively. The sample complexity in Theorem 2 scales at a rate $\widetilde{\mathcal{O}}(\sqrt{H}d^2\kappa^5/\sqrt{n})$, where $n$ denotes the training sample size. We obtain a $n^{-1/2}$ convergence rate with a mild polynomial dependence on the sequence length $H$. In addition, $\kappa$ is the condition number induced by the missing patterns. To establish Theorem 2, we develop a novel score representation theory (Theorem 1) for DiTs, where we utilize an algorithm unrolling technique.

- **Uncertainty Quantification**. Leveraging the generative power of trained DiTs, we construct confidence regions (intervals) from massive generated missing values. This approach possesses its natural appeal and enjoys strong coverage guarantees (Corollary 1). We show that the coverage probability converges to the desired level at a $\widetilde{\mathcal{O}}(n^{-1/2})$ rate. Meanwhile, the missing patterns influences the convergence.

- **Mixed-Masking Training Strategy**. Motivated by our theoretical results, we propose a training strategy blending different masking schemes to cover diverse missing patterns. The performance of our method on synthetic datasets validates our findings and outperforms benchmark methods.

**Notations** We use bold lowercase letters to denote vectors and bold uppercase letters to denote matrices. For a vector $\mathbf{v}$, $\|\mathbf{v}\|_2$ denotes its Euclidean norm. For a matrix $\mathbf{A}$, $\|\mathbf{A}\|_2$ and $\|\mathbf{A}\|_{\mathrm{F}}$ denote its spectral norm and Frobenius norm, respectively, and $\|\mathbf{A}\|_\infty = \max_{i,j}|A_{ij}|$. When matrix $\mathbf{A}$ is positive definite, we denote $\lambda_{\max}(\mathbf{A})$ and $\lambda_{\min}(\mathbf{A})$ as its largest and smallest eigenvalues; its condition number is $\kappa(\mathbf{A}) = \lambda_{\max}(\mathbf{A})/\lambda_{\min}(\mathbf{A})$. We denote $f \lesssim g$ if there exists a constant $C > 0$ such that $f \leq Cg$. Notation $\mathcal{O}(\cdot)$ suppresses constants, while $\widetilde{\mathcal{O}}(\cdot)$ further hides logarithmic factors.

Due to space limit, the related work section is deferred to Appendix A.

## 2 Imputation in Gaussian Processes via Conditional Diffusion Models

In this section, we formalize the imputation task as a conditional distribution estimation problem. When the data are sampled from a Gaussian process, we identify rich structures in the conditional distribution. We then utilize a DiT to learn the distribution of missing values. Lastly, we summarize diffusion-based imputation method in Algorithm 1.

## 2.1 Imputation for Gaussian Process Data

Imputation refers to the task of inferring missing values given the observed ones. Denote by $\mathcal{I} = \{1, \ldots, H\}$ the set of all time indices. For a multivariate sequence $\mathbf{X} = [\mathbf{x}_1, \ldots, \mathbf{x}_H] \in \mathbb{R}^{d \times H}$ of length $H$, we consider a block-missing setting, where certain time frames are entirely unobserved. The subset of observed indices is denoted by $\mathcal{I}_{\mathrm{obs}} = \{i_1, \ldots, i_{|\mathcal{I}_{\mathrm{obs}}|}\}$, where $|\mathcal{I}_{\mathrm{obs}}|$ denotes the cardinality. Correspondingly, $\mathcal{I}_{\mathrm{miss}} = \mathcal{I} \setminus \mathcal{I}_{\mathrm{obs}}$ denotes the time indices of missing frames. To avoid degenerate cases, we assume $0 < |\mathcal{I}_{\mathrm{miss}}| < H$. In the sequel, we focus on the *Missing Completely at Random* case [Little, 1988], where each index in $\mathcal{I}_{\mathrm{miss}}$ is independently sampled from some underlying distribution.

We represent the vectorized observed partial sequence as $\mathbf{x}_{\mathrm{obs}} = [\mathbf{x}_{i_1}^\top, \ldots, \mathbf{x}_{i_{|\mathcal{I}_{\mathrm{obs}}|}}^\top]^\top \in \mathbb{R}^{d|\mathcal{I}_{\mathrm{obs}}|}$, and the vectorized missing part as $\mathbf{x}_{\mathrm{miss}} = [\mathbf{x}_{j_1}^\top, \ldots, \mathbf{x}_{j_{|\mathcal{I}_{\mathrm{miss}}|}}^\top]^\top \in \mathbb{R}^{d|\mathcal{I}_{\mathrm{miss}}|}$. *We estimate the missing values by learning the conditional distribution* $P(\mathbf{x}_{\mathrm{miss}} \mid \mathbf{x}_{\mathrm{obs}})$. Notably, learning the conditional distribution goes beyond point estimates of the missing values, but provides easy access to confidence regions. We slightly abuse the notation by using $\mathbf{x}$ to simultaneously refer to random vectors.

Throughout our theoretical analysis, we focus on $d$-dimensional Gaussian process data. To uniquely distinguish a Gaussian process, it suffices to specify its mean and covariance functions. In particular, we denote the mean as $\boldsymbol{\mu}_i = \mathbb{E}[\mathbf{x}_i]$ and we parameterize the covariance matrix by $\mathrm{Cov}[\mathbf{x}_i, \mathbf{x}_j] = \gamma(i, j)\boldsymbol{\Lambda}$, where $\boldsymbol{\Lambda} = \mathrm{Var}[\mathbf{x}_h] \in \mathbb{R}^{d \times d}$ for any $h$ and $\gamma$ is a kernel function. It is worth mentioning that $\boldsymbol{\Lambda}$ captures the spatial dependencies and function $\gamma$ represents temporal correlation. The kernel function $\gamma$ dictates the strength and decay of the temporal dependencies among different data frames. The joint distribution of a sequence $\mathrm{vec}(\mathbf{X}) = [\mathbf{x}_1^\top, \cdots, \mathbf{x}_H^\top]^\top$ is Gaussian $\mathcal{N}(\boldsymbol{\mu}, \boldsymbol{\Gamma} \otimes \boldsymbol{\Lambda})$, where

$$
\boldsymbol{\mu} = \begin{bmatrix} \boldsymbol{\mu}_1, \\ \vdots \\ \boldsymbol{\mu}_H \end{bmatrix} \quad \text{and} \quad \boldsymbol{\Gamma} \otimes \boldsymbol{\Lambda} = \begin{bmatrix} \gamma(1,1)\boldsymbol{\Lambda} & \cdots & \gamma(1,H)\boldsymbol{\Lambda} \\ \vdots & \ddots & \vdots \\ \gamma(H,1)\boldsymbol{\Lambda} & \cdots & \gamma(H,H)\boldsymbol{\Lambda} \end{bmatrix}.
$$

Here $\boldsymbol{\Gamma}_{ij} = \gamma(i, j)$ and $\otimes$ is the matrix Kronecker product. We impose the following assumption for characterizing the temporal dependencies.

**Assumption 1.** *There exists $d_e$-dimensional embedding $\{\mathbf{e}_i \in \mathbb{R}^{d_e}\}_{i=1}^H$ such that $\|\mathbf{e}_i\|_2 = r$ for a constant $r$. Moreover, for any $i, j$, it holds that $\|\mathbf{e}_i - \mathbf{e}_j\|_2 = f(|i - j|)$, and for $|i_1 - j_1| \neq |i_2 - j_2|$, $f(|i_1 - j_1|) \neq f(|i_2 - j_2|)$. Kernel function $\gamma(i, j)$ only depends on $\|\mathbf{e}_i - \mathbf{e}_j\|_2$. Furthermore, we assume $\boldsymbol{\Gamma}$ and $\boldsymbol{\Lambda}$ are positive definite.*

Assumption 1 ensures that the pairwise distances in the embedding uniquely identifies positional gaps. We do not specify a particular form of the kernel function, which encodes many commonly ones such as Gaussian Radial Basis Function (RBF), Ornstein–Uhlenbeck kernels, and Matérn kernels [Rasmussen and Williams, 2006]. As a concrete example, sinusoidal embedding is widely used in transformer networks [Vaswani et al., 2017]. Consider a two-dimensional embedding defined as $\mathbf{e}_i = [r \sin(2\pi i/C), r \cos(2\pi i/C)]^\top$, where $r > 0$ is a fixed radius and $C > 0$ is a scaling constant. The Euclidean distance between any two embedding is $\|\mathbf{e}_i - \mathbf{e}_j\|_2 = 2r|\sin(\pi(i - j)/C)|$, which is strictly positive for $i \neq j$, and approximately linear in $|i - j|$ when $C$ is sufficiently large.

Under the Gaussian process setting, the conditional distribution of $\mathbf{x}_{\mathrm{miss}}|\mathbf{x}_{\mathrm{obs}}$ is still Gaussian [Bishop and Nasrabadi, 2006]. The conditional mean and covariance are given by

$$
\boldsymbol{\mu}_{\mathrm{cond}}(\mathbf{x}_{\mathrm{obs}}) = \boldsymbol{\mu}_{\mathrm{miss}} + \boldsymbol{\Sigma}_{\mathrm{cor}}^\top \boldsymbol{\Sigma}_{\mathrm{obs}}^{-1}(\mathbf{x}_{\mathrm{obs}} - \boldsymbol{\mu}_{\mathrm{obs}}), \quad \boldsymbol{\Sigma}_{\mathrm{cond}} = \boldsymbol{\Sigma}_{\mathrm{miss}} - \boldsymbol{\Sigma}_{\mathrm{cor}}^\top \boldsymbol{\Sigma}_{\mathrm{obs}}^{-1} \boldsymbol{\Sigma}_{\mathrm{cor}},
$$

where we denote $\boldsymbol{\mu}_{\mathrm{obs}} = \mathbb{E}[\mathbf{x}_{\mathrm{obs}}]$ (the same holds for $\boldsymbol{\mu}_{\mathrm{miss}}$), $\boldsymbol{\Sigma}_{\mathrm{cor}} = \mathrm{Cov}[\mathbf{x}_{\mathrm{obs}}, \mathbf{x}_{\mathrm{miss}}]$, and $\boldsymbol{\Sigma}_{\mathrm{obs}}$ (resp. $\boldsymbol{\Sigma}_{\mathrm{miss}}$) as the covariance of $\mathbf{x}_{\mathrm{obs}}$ (resp. $\mathbf{x}_{\mathrm{miss}}$). See Figure 1 for a graphical demonstration. We check that $\boldsymbol{\Sigma}_{\mathrm{obs}} = \boldsymbol{\Gamma}_{\mathrm{obs}} \otimes \boldsymbol{\Lambda}$ with $\boldsymbol{\Gamma}_{\mathrm{obs}} \in \mathbb{R}^{|\mathcal{I}_{\mathrm{obs}}| \times |\mathcal{I}_{\mathrm{obs}}|}$ capturing correlation among index set $\mathcal{I}_{\mathrm{obs}}$.

## 2.2 Training Diffusion Transformers for Imputation

We estimate the conditional distribution $P(\mathbf{x}_{\mathrm{miss}} \mid \mathbf{x}_{\mathrm{obs}})$ using diffusion transformers. A diffusion model consists of two coupled processes—a forward and a backward process. We adopt a continuous-time description. In the forward process, we gradually corrupt data by

$$
\mathrm{d}\mathbf{x}_t = -\frac{1}{2}\mathbf{x}_t \mathrm{d}t + \mathrm{d}\mathbf{w}_t \quad \text{with} \quad \mathbf{x}_0 \sim P(\cdot \mid \mathbf{x}_{\mathrm{obs}}), \tag{1}
$$

and $\mathbf{w}_t$ is a Wiener process. The forward process terminates at a sufficiently large time $T$ and we denote the distribution of $\mathbf{x}_t$ as $P_t(\cdot|\mathbf{x}_{\mathrm{obs}})$ with density $p_t(\cdot|\mathbf{x}_{\mathrm{obs}})$. Note that we only corrupt the missing values by Gaussian noise, but keep the observed partial sequence $\mathbf{x}_{\mathrm{obs}}$ unchanged.

Corresponding to the forward process, the backward process simulates the reverse evolution of the forward process. As a result, it generates new samples by progressively removing noise:

$$d\mathbf{v}_t = \left[\frac{1}{2}\mathbf{v}_t + \nabla_{\mathbf{v}_t} \log p_{T-t}(\mathbf{v}_t \mid \mathbf{x}_{\text{obs}})\right] dt + d\bar{\mathbf{w}}_t \quad \text{with} \quad \mathbf{v}_0 \sim P_T(\cdot \mid \mathbf{x}_{\text{obs}}), \qquad (2)$$

where $\bar{\mathbf{w}}_t$ is another Wiener process and $\nabla_{\mathbf{v}_t} \log p_t(\mathbf{v}_t \mid \mathbf{x}_{\text{obs}})$ is the conditional score function. In the remaining of the paper, we drop the subscript $\mathbf{v}_t$ in the score function for simplicity. Unfortunately, $\nabla \log p_{T-t}(\mathbf{v}_t \mid \mathbf{x}_{\text{obs}})$ is typically unknown and must be estimated using a neural network. We denote the estimated score function by $\widehat{\mathbf{s}}(\mathbf{v}_t, \mathbf{x}_{\text{obs}}, t)$. Consequently, the sample generation process follows an alternative backward SDE:

$$d\widehat{\mathbf{v}}_t = \left[\frac{1}{2}\widehat{\mathbf{v}}_t + \widehat{\mathbf{s}}(\widehat{\mathbf{v}}_t, \mathbf{x}_{\text{obs}}, t)\right] dt + d\bar{\mathbf{w}}_t \quad \text{with} \quad \widehat{\mathbf{v}}_0 \sim \mathcal{N}(\mathbf{0}, \mathbf{I}_{d|I_{\text{miss}}|}). \qquad (3)$$

Here, we also replace the unknown $P_T$ by a standard Gaussian distribution.

When training the score estimator $\widehat{\mathbf{s}}$, we assume access to fully observed sequences. To simulate a partially observed sequence, we sample a masking sequence $\{\tau_1, \ldots, \tau_H\} \in \{0, 1\}^H$, where $0$ denotes missing the observation and $1$ keeping the observation. Then $\mathbf{x}_{\text{obs}}$ is extracted according to the masking sequence. In later context, we will investigate how to choose masking strategies. We summarize the diffusion-based method for sequence imputation in Algorithm 1.

---

**Algorithm 1** Diffusion-Based Sequence Imputation

---

1: `Module I: Training`
2: **Input:** Fully observed sequences $\mathcal{D} := \{\mathbf{X}_i\}_{i=1}^n$, a masking strategy.
3: Simulate $\{\mathbf{x}_{\text{obs}}^{(i)}, \mathbf{x}_{\text{miss}}^{(i)}\}_{i=1}^n$ pairs via the masking strategy, and train a conditional diffusion model.
4: **Output:** A well-trained conditional diffusion model.

5: `Module II: Imputation`
6: **Input:** Conditional diffusion model from `Module I`, a new partial sequence $\mathbf{x}_{\text{obs}}^*$, repetition time $Z$, and confidence level $1 - \alpha$.
7: Conditioned on $\mathbf{x}_{\text{obs}}^*$, independently generate $B$ missing sequences $\widehat{\mathbf{x}}_{\text{miss}}^{(z)}$ for $z = 1, \ldots, Z$.
8: ⋆ *Point estimate*: Mean $\widehat{\mathbf{x}}_{\text{miss}}^* = \frac{1}{Z}\sum_{z=1}^Z \widehat{\mathbf{x}}_{\text{miss}}^{(z)}$ (or median of the generated sequences).
9: ⋆ *Confidence region*: $\widehat{\mathcal{CR}}_{1-\alpha}^* = \{\mathbf{x}_{\text{miss}} : \|\mathbf{x}_{\text{miss}} - \widehat{\mathbf{x}}_{\text{miss}}^*\|_2 \leq \widehat{D}_{1-\alpha}^*\}$,
    where $\widehat{D}_{1-\alpha}^*$ is the $1 - \alpha$ upper quantile of $\|\widehat{\mathbf{x}}_{\text{miss}}^{(z)} - \widehat{\mathbf{x}}_{\text{miss}}^*\|_2$ for $z = 1, \ldots, Z$.
10: **Return:** $\widehat{\mathbf{x}}_{\text{miss}}^*$ and $\widehat{\mathcal{CR}}_{1-\alpha}^*$.

---

For the rest of the paper, we parameterize the conditional score function using a transformer network. A transformer [Vaswani et al., 2017], comprises a series of blocks and each block encompasses a multi-head attention layer and a feedforward layer. Let $\mathbf{Y} = [\mathbf{y}_1, \ldots, \mathbf{y}_H] \in \mathbb{R}^{D \times H}$ be the (column) stacking matrix of $H$ patches. In a transformer block, the multi-head attention layer computes

$$\text{Attn}(\mathbf{Y}) = \mathbf{Y} + \sum_{m=1}^M \mathbf{V}^m \mathbf{Y} \cdot \sigma\big((\mathbf{Q}^m \mathbf{Y})^\top \mathbf{K}^m \mathbf{Y}\big), \qquad (4)$$

where $\mathbf{V}^m, \mathbf{Q}^m, \mathbf{K}^m$ are weight matrices of corresponding sizes in the $m$-th attention head, and $\sigma$ is an activation function. The attention layer is followed by a feedforward layer, which computes

$$\text{FFN}(\mathbf{Y}) = \mathbf{Y} + \mathbf{W}_1 \cdot \text{ReLU}(\mathbf{W}_2 \mathbf{Y} + \mathbf{b}_2 \mathbf{1}^\top) + \mathbf{b}_1 \mathbf{1}^\top.$$

Here, $\mathbf{W}_1, \mathbf{W}_2$ are weight matrices, $\mathbf{b}_1$ and $\mathbf{b}_2$ are offset vectors, $\mathbf{1}$ denotes a vector of ones, and the ReLU activation function is applied entry-wise. This feedforward layer performs a linear transformation to the output of the attention module with more flexibility. For our study, the raw input to a transformer is $H$ patches of $d$-dimensional vectors and time $t$ in the backward process. We refer to $\mathcal{T}(D, L, M, B, R)$ as a transformer architecture defined by

$$\mathcal{T}(D, L, M, B, R) = \big\{ f : f = f_{\text{out}} \circ (\text{FFN}_L \circ \text{Attn}_L) \circ \cdots \circ (\text{FFN}_1 \circ \text{Attn}_1) \circ f_{\text{in}},$$
$$\text{Attn}_i \text{ uses entrywise ReLU activation for } i = 1, \ldots, L,$$
$$\text{number of heads in each Attn is bounded by } M,$$
$$\text{the Frobenius norm of each weight matrix is bounded by } B,$$
$$\text{the output range } \|f\|_2 \text{ is bounded by } R\}. \qquad (5)$$

## 3    Conditional Score Approximation via Algorithm Unrolling

Suggested by the sample generation process (3), the key is to learn the conditional score function. This section devotes to establishing a novel score approximation theory of transformers based on algorithm unrolling.

Since $\mathbf{x}_{\mathrm{miss}}|\mathbf{x}_{\mathrm{obs}}$ is Gaussian, the forward process (1) yields the following closed-form score function:

$$\nabla \log p_t(\mathbf{v}_t|\mathbf{x}_{\mathrm{obs}}) = -(\alpha_t^2 \mathbf{\Sigma}_{\mathrm{cond}} + \sigma_t^2 \mathbf{I})^{-1}(\mathbf{v}_t - \alpha_t \boldsymbol{\mu}_{\mathrm{cond}}(\mathbf{x}_{\mathrm{obs}})), \tag{6}$$

where $\alpha_t = e^{-\frac{t}{2}}$ and $\sigma_t = \sqrt{1 - e^{-t}}$. The matrix inverse poses a challenge in representing the score by a transformer, as it may deteriorate structures in $\mathbf{\Sigma}_{\mathrm{cond}}$. Therefore, we reformulate the conditional score function as the optimal solution of a quadratic optimization problem:

$$\nabla \log p_t(\mathbf{v}_t|\mathbf{x}_{\mathrm{obs}}) = \arg\min_{\mathbf{s}} \ \mathcal{L}_t(\mathbf{s}) := \frac{1}{2}\mathbf{s}^\top \left(\alpha_t^2 \mathbf{\Sigma}_{\mathrm{cond}} + \sigma_t^2 \mathbf{I}\right)\mathbf{s} + \mathbf{s}^\top \left(\mathbf{v}_t - \alpha_t \boldsymbol{\mu}_{\mathrm{cond}}(\mathbf{x}_{\mathrm{obs}})\right). \tag{7}$$

It suffices to obtain an approximate optimal solution of (7) using a gradient descent algorithm. At the $k$-th iteration, with a step size $\eta_t$, we have

$$\mathbf{s}^{(k+1)} = \mathbf{s}^{(k)} - \eta_t \underbrace{\left[(\sigma_t^2 \mathbf{I} + \alpha_t^2 \mathbf{\Sigma}_{\mathrm{miss}})\mathbf{s}^{(k)} + \alpha_t^2 \mathbf{\Sigma}_{\mathrm{cor}}^\top \mathbf{\Sigma}_{\mathrm{obs}}^{-1} \mathbf{\Sigma}_{\mathrm{cor}}\mathbf{s}^{(k)} + (\mathbf{v}_t - \alpha_t \boldsymbol{\mu}_{\mathrm{cond}}(\mathbf{x}_{\mathrm{obs}}))\right]}_{\nabla \mathcal{L}_t(\mathbf{s}^{(k)})}, \tag{8}$$

for $k = 0, \ldots, K - 1$. Unfortunately, we encounter another matrix inverse in $\mathbf{\Sigma}_{\mathrm{obs}}^{-1} \mathbf{\Sigma}_{\mathrm{cor}}\mathbf{s}^{(k)}$. Analogous to (7), we consider an auxiliary quadratic optimization problem:

$$\mathbf{\Sigma}_{\mathrm{obs}}^{-1} \mathbf{\Sigma}_{\mathrm{cor}}\mathbf{s}^{(k)} = \arg\min_{\mathbf{u}} \mathcal{L}_{\mathrm{aux}}^{(k)}(\mathbf{u}) := \frac{1}{2}\mathbf{u}^\top \mathbf{\Sigma}_{\mathrm{obs}}\mathbf{u} - \mathbf{u}^\top \mathbf{\Sigma}_{\mathrm{cor}}\mathbf{s}^{(k)}. \tag{9}$$

Via a gradient descent algorithm with step size $\theta$, the update reads

$$\mathbf{u}^{(k_{\mathrm{aux}}+1)} = \mathbf{u}^{(k_{\mathrm{aux}})} - \theta \nabla \mathcal{L}_{\mathrm{aux}}^{(k)}(\mathbf{u}) = \mathbf{u} - \theta \left(\mathbf{\Sigma}_{\mathrm{obs}}\mathbf{u} - \mathbf{\Sigma}_{\mathrm{cor}}\mathbf{s}^{(k)}\right), \tag{10}$$

where iteration index $k_{\mathrm{aux}} = 0, \ldots, K_{\mathrm{aux}} - 1$.

We substitute the last iterate $\mathbf{u}^{(K_{\mathrm{aux}})}$ into the right-hand side of (8) to obtain $\widetilde{\nabla}\mathcal{L}_t(\mathbf{s}^{(k)})$ as an approximation to $\nabla\mathcal{L}_t(\mathbf{s}^{(k)})$. We summarize the nested gradient descent algorithm for calculating the conditional score function in Algorithm 2.

---

**Algorithm 2** Nested Gradient Descent for Representing Score Function

---

1: **Input:** Observation $\mathbf{x}_{\mathrm{obs}}$, current state $\mathbf{v}_t$, time $t$, step sizes $\eta_t, \theta$, iteration counts $K_{\mathrm{aux}}, K$.
   *(Major) Gradient Descent:*
2: Initialize $\mathbf{s}^{(0)} = \mathbf{0}$.
3: **for** $k = 0, 1, \ldots, K - 1$ **do**
       *Auxiliary Gradient Descent:*
4:    Initialize $\mathbf{u}^{(0)} = \mathbf{0}$.
5:    **for** $k_{\mathrm{aux}} = 0, 1, \ldots, K_{\mathrm{aux}} - 1$ **do**
6:        $\mathbf{u}^{(k_{\mathrm{aux}}+1)} = \mathbf{u}^{(k_{\mathrm{aux}})} - \theta \nabla \mathcal{L}_{\mathrm{aux}}^{(k)}(\mathbf{u}^{(k_{\mathrm{aux}})})$.
7:    Calculate $\widetilde{\nabla}\mathcal{L}_t(\mathbf{s}^{(k)})$ using $\mathbf{u}^{(K_{\mathrm{aux}})}$.
8:    $\mathbf{s}^{(k+1)} = \mathbf{s}^{(k)} - \eta_t \widetilde{\nabla}\mathcal{L}_t(\mathbf{s}^{(k)})$.
9: **Return:** $\mathbf{s}^{(K)}$.

---

With sufficiently large $K_{\mathrm{aux}}$ and $K$, the representation error of Algorithm 2 can be well-controlled.

**Lemma 1** (Representation error of Algorithm 2)**.** *Suppose Assumption 1 holds. For an arbitrarily fixed time $t \in (0, T]$, given an error tolerance $\epsilon \in (0, 1)$, choose $K, K_{\mathrm{aux}}$ as*

$$K = \mathcal{O}\left(\kappa(\mathbf{\Sigma}_{\mathrm{cond}})\log\left(\frac{Hd\kappa(\mathbf{\Sigma}_{\mathrm{cond}})\kappa(\mathbf{\Sigma}_{\mathrm{obs}})}{\epsilon}\right)\right), K_{\mathrm{aux}} = \mathcal{O}\left(\kappa(\mathbf{\Sigma}_{\mathrm{obs}})\log\left(\frac{Hd\kappa(\mathbf{\Sigma}_{\mathrm{obs}})}{\sigma_t \epsilon}\right)\right).$$

*Then, given $\delta > 0$, for any $\mathbf{x}_{\mathrm{obs}}$ and $\mathbf{v}_t$ in a compact region $\mathcal{C}_\delta$, there exist step sizes $\eta_t$ and $\theta$ such that running Algorithm 2 gives rise to*

$$\|\mathbf{s}^{(K)} - \nabla \log p_t(\mathbf{v}_t \mid \mathbf{x}_{\mathrm{obs}})\|_2 \leq \sigma_t^{-1}\epsilon.$$

Detailed proof of Lemma 1 is provided in Appendix B. The compact region $\mathcal{C}_\delta$ truncates the norm of $\mathbf{x}_{\text{obs}}$ and $\mathbf{v}_t$, which is plausible due to the Gaussian tail; see a precise definition of $\mathcal{C}_\delta$ in Appendix Equation (13). Lemma 1 suggests that the computational complexity of Algorithm 2 for approximating the score function is governed by the condition numbers of $\boldsymbol{\Sigma}_{\text{cond}}$ and $\boldsymbol{\Sigma}_{\text{obs}}$. A large condition number on $\boldsymbol{\Sigma}_{\text{cond}}$ implies that the variability of missing values among different directions changes significantly. Equivalently, with a large condition number, given $\mathbf{x}_{\text{obs}}$, the missing values exhibit strong anistropic uncertainty that complicates the imputation.

Representing the conditional score function by a nested gradient descent algorithm enables an effective transformer network approximation. We show that transformers can realize each gradient descent iteration using a constant number of attention blocks. We provide the following score approximation theory using transformers.

**Theorem 1.** *Suppose Assumption 1 holds. Given an early stopping time $t_0 \in (0, T]$ and an error level $\epsilon \in (0, 1)$, for any $\mathbf{x}_{\text{obs}}, \mathbf{v}_t \in \mathcal{C}_\delta$, there exists a transformer architecture $\mathcal{T}(D, L, M, B, R)$ such that, with proper weight parameters, it yields an approximation $\widetilde{\mathbf{s}}$ satisfying*

$$\|\widetilde{\mathbf{s}}(\mathbf{v}_t, \mathbf{x}_{\text{obs}}, t) - \nabla \log p_t(\mathbf{v}_t | \mathbf{x}_{\text{obs}})\|_2 \leq \sigma_t^{-1} \epsilon \quad \text{for all } t \in [t_0, T].$$

*The configuration of the transformer architecture satisfies*

$$D = \mathcal{O}(d + d_e), \quad L = \mathcal{O}\left(\kappa_{\max}^2(\boldsymbol{\Sigma}_{\text{cond}})\kappa_{\max}(\boldsymbol{\Sigma}_{\text{obs}}) \log^3\left(\frac{Hd\kappa_{\max}^2(\boldsymbol{\Sigma}_{\text{cond}})\kappa_{\max}(\boldsymbol{\Sigma}_{\text{obs}})}{\epsilon}\right)\right),$$

$$M = 4H, \quad B = \mathcal{O}\left(\sqrt{Hd^3}(r^2 + \kappa_{\max}(\boldsymbol{\Sigma}_{\text{obs}})\sigma_{t_0}^{-1})\right), \quad R = \mathcal{O}(\sigma_{t_0}^{-2}\sqrt{Hd}\kappa_{\max}(\boldsymbol{\Sigma}_{\text{obs}})),$$

*where we define $\kappa_{\max}(\cdot) = \sup_{\mathcal{I}_{\text{obs}}} \kappa(\cdot)$.*

The proof is provided in Appendix C. Figure 1 depicts the transformer architecture in our constructive proof, which unrolls Algorithm 2 efficiently. To obtain the approximation error bound, we develop a careful analysis of the error propagation in the auxiliary gradient descent for calculating $\widetilde{\nabla}\mathcal{L}_t$. Theorem 1 also reinforces the insights from Lemma 1, where we observe that the size of the transformer network scales with the worst-case condition number. We will further discuss the relation between missing patterns and the condition number in Theorem 2.

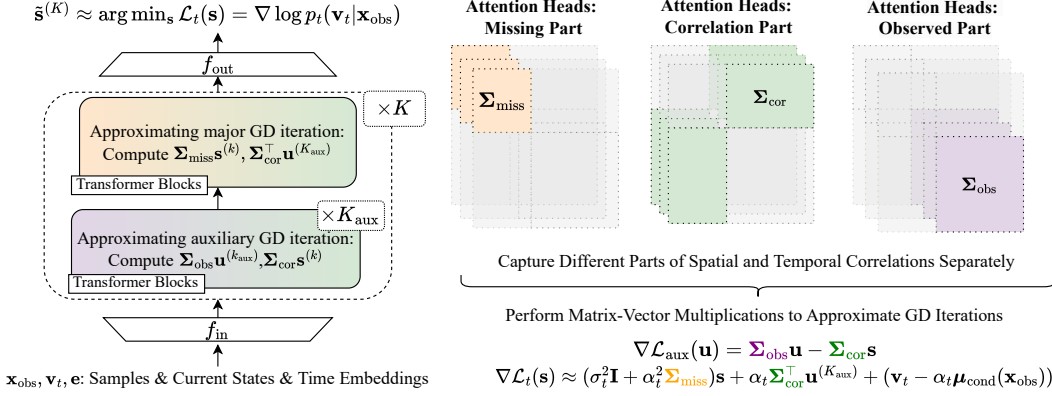

Figure 1: Constructed transformer architecture: Within each transformer block, attention heads focus on capturing information of different covariance components ($\boldsymbol{\Sigma}_{\text{obs}}, \boldsymbol{\Sigma}_{\text{cor}}, \boldsymbol{\Sigma}_{\text{miss}}$) separately, and approximate corresponding matrix–vector multiplications. A total of $K$ block groups perform major GD steps, with $K_{\text{aux}}$ inner blocks in each group dedicated to solving the auxiliary problem.

## 4 Capturing Conditional Distribution and Uncertainty Quantification

Given a properly chosen transformer architecture, we establish guarantees for learning the conditional distribution of missing values and uncertainty quantification. We consider an estimated score network $\widehat{\mathbf{s}}$ obtained by minimizing the following empirical score matching loss (a detailed derivation is deferred to Appendix E):

$$\widehat{\mathbf{s}} \in \arg\min_{\mathbf{s} \in \mathcal{T}} \widehat{\mathcal{L}}(\mathbf{s}) \coloneqq \frac{1}{n} \sum_{i=1}^n \ell(\mathbf{x}_{\text{miss}}^{(i)}, \mathbf{x}_{\text{obs}}^{(i)}; \mathbf{s}), \tag{11}$$

where

$$\ell(\mathbf{x}_{\mathrm{miss}}^{(i)}, \mathbf{x}_{\mathrm{obs}}^{(i)}; \mathbf{s}) = \int_{t_0}^{T} \mathbb{E}_{\mathbf{v}_t | \mathbf{v}_0 = \mathbf{x}_{\mathrm{miss}}^{(i)}} \left[ \| \mathbf{s}(\mathbf{v}_t, \mathbf{x}_{\mathrm{obs}}^{(i)}, t) - (\mathbf{v}_t - \alpha_t \mathbf{v}_0) / \sigma_t^2 \|_2^2 \right] dt. \quad (12)$$

Substituting the learned score $\widehat{\mathbf{s}}$ into the backward SDE (3) yields generated distribution $\mathbf{v}_{t_0} \sim \widehat{P}_{t_0}(\cdot \mid \mathbf{x}_{\mathrm{obs}})$. We introduce an early-stopping time $t_0$ to stabilize the training and sample generation [Song et al., 2020]. We now present a convergence guarantee of $\widehat{P}_{t_0}$ to the true conditional distribution.

**Theorem 2.** *Referring to the training procedure in Algorithm 1, by choosing the transformer architecture as in Theorem 1 with $\epsilon = n^{-\frac{1}{2}}$, terminal time $T = \mathcal{O}(\log n)$, and early-stopping time $t_0 = \mathcal{O}(\lambda_{\min}(\boldsymbol{\Sigma}_{\mathrm{cond}}) n^{-\frac{1}{2}})$, it holds that*

$$\epsilon_{\mathrm{dist}}^{(n)} := \mathbb{E}_{\mathcal{D}^{(n)}} \left[ \mathbb{E}_{\mathbf{x}_{\mathrm{obs}}} \left[ \mathrm{TV}(P(\cdot | \mathbf{x}_{\mathrm{obs}}), \widehat{P}_{t_0}(\cdot | \mathbf{x}_{\mathrm{obs}})) \right] \right] = \widetilde{O}(\sqrt{H d^2 \kappa^5(\boldsymbol{\Sigma}_{\mathrm{cond}}) \kappa^2(\boldsymbol{\Sigma}_{\mathrm{obs}})} / \sqrt{n}).$$

The proof of Theorem 2 is provided in Appendix D. This result establishes that DiT can efficiently learn the true conditional distribution of missing values. The sample complexity mildly depends on the sequence length. More importantly, the bound highlights that the estimation error depends on the condition numbers of $\boldsymbol{\Sigma}_{\mathrm{cond}}$ and $\boldsymbol{\Sigma}_{\mathrm{obs}}$, reflecting the discussion after Lemma 1.

We provide an example to demonstrate that different missing patterns can lead to distinct condition numbers. Consider data of length $H = 96$ with time correlation modeled by a Laplace kernel $\gamma(i, j) = \exp(-\|\mathbf{e}_i - \mathbf{e}_j\|_2 / 128)$, and missing length $|\mathcal{I}_{\mathrm{miss}}| = 16$. Clustered missingness—16 consecutive missing entries at the tail—yields a large condition number $\kappa(\boldsymbol{\Sigma}_{\mathrm{cond}}) = 415.40$, making the task challenging. In contrast, dispersed missing patterns, 16 randomly placed missing entries, result in much smaller $\kappa(\boldsymbol{\Sigma}_{\mathrm{cond}}) = 3.00$, making estimation easier. We provide numerical results on this example in Section 5.

**Confidence Region Construction**  Given the learned conditional distribution $\widehat{P}_{t_0}$ and a new observed sequence $\mathbf{x}_{\mathrm{obs}}^*$, we deploy the model to generate samples and form point estimates and confidence regions as in Algorithm 1. Since $\mathbf{x}_{\mathrm{obs}}^*$ may not be seen in the training samples, we encounter a distribution shift, meaning that we need to transfer the knowledge in the learned model to the new testing instance. The subtlety here is how to quantify the knowledge transfer rate. Our proposal is the following class-dependent distribution shift coefficient.

**Definition 1.** *The distribution shift between two probability distributions $P_1$ and $P_2$ with respect to a function class $\mathcal{G}$ is defined as $\mathsf{DS}(P_1, P_2; \mathcal{G}) = \sup_{g \in \mathcal{G}} \frac{\mathbb{E}_{\mathbf{y} \sim P_1}[g(\mathbf{y})]}{\mathbb{E}_{\mathbf{y} \sim P_2}[g(\mathbf{y})]}$.*

In our analysis, we specialize $\mathcal{G}$ to a function class induced by the transformer network:

$$\mathcal{G} = \{ g(\mathbf{y}) = \mathbb{E}_{\mathbf{x}_{\mathrm{miss}}}[\ell(\mathbf{x}_{\mathrm{miss}}, \mathbf{y}; \mathbf{s})] : \mathbf{s} \in \mathcal{T}(D, L, M, B, R) \} .$$

Since $\mathcal{G}$ might be insensitive to certain distinctions, it introduces some smoothing effect to capture the difference between $P_1$ and $P_2$. We consider $P_1$ and $P_2$ as the marginal training distribution of $\mathbf{x}_{\mathrm{obs}}$ and the point mass of the testing distribution $\mathbb{1}\{\cdot = \mathbf{x}_{\mathrm{obs}}^*\}$, denoted as $P_{\mathbf{x}_{\mathrm{obs}}}$ and $P_{\mathbf{x}_{\mathrm{obs}}^*}$, respectively. The following corollary provides a guarantee for the coverage probability of the constructed CR.

**Corollary 1.** *Under the setting of Theorem 2, given $\mathbf{x}_{\mathrm{obs}}^*$, Algorithm 1 yields $\widehat{\mathcal{CR}}_{1-\alpha}^*$ satisfying*

$$\mathbb{E}_{\mathcal{D}^{(n)}} \left[ \mathbb{P}(\mathbf{x}_{\mathrm{miss}}^* \in \widehat{\mathcal{CR}}_{1-\alpha}^*) \right] \geq (1 - \alpha) - \epsilon_{\mathrm{dist}}^{(n)} \cdot \sqrt{\mathsf{DS}(P_{\mathbf{x}_{\mathrm{obs}}^*}, P_{\mathbf{x}_{\mathrm{obs}}}; \mathcal{G})} - n^{-\frac{1}{2}} \psi(\mathbf{x}_{\mathrm{obs}}^*),$$

*where $\psi(\mathbf{x}_{\mathrm{obs}}^*)$ is independent of $n$ and proportional to $\|\mathbf{x}_{\mathrm{obs}}^*\|_2$ and $\kappa(\boldsymbol{\Sigma}_{\mathrm{cond}})$.*

Detailed proof is provided in Appendix D. Corollary 1 says that the coverage probability of the constructed CR converges to the desired level at the same rate of the conditional distribution estimation. More importantly, the distribution shift coefficient directly influences the coverage probability. We present a detailed discussion in the following remark.

**Remark 1.** There are two factors controlling the distribution shift coefficient: 1) the observed values in $\mathbf{x}_{\mathrm{obs}}^*$ and 2) the missing pattern. From our theoretical analysis, we identify a profound impact of the missingness patterns on the learning efficiency and the choice of transformer architectures. Indeed, when the masking strategy in Algorithm 1 is relatively easy, $\epsilon_{\mathrm{dist}}^{(n)}$ is small. However, $\mathbf{x}_{\mathrm{obs}}^*$ can deviate significantly from the training samples, causing a large distribution shift. On the contrary,

including harder masks can effectively reduce the distribution shift, but elevates learning difficulty. As a result, there is a trade-off between the masking strategy and the reliability of the trained diffusion transformer for imputation. In Section 5, we introduce a mixed-masking training strategy to enhance the performance of diffusion transformers, where diverse masking patterns are randomly sampled. This reduces distribution shift and improves robustness to varying imputation difficulty.

# 5 Experiments

We evaluate the performance of DiT through simulation to validate our theoretical results on imputation efficiency, uncertainty quantification, and the effectiveness of the mixed-masking training strategy. Experiments are conducted on Gaussian processes and, additionally, on more complex latent Gaussian processes to assess generalization beyond our theoretical scope. The DiT implementation builds on the DiT codebase [Peebles and Xie, 2022]. Further experimental details and real-world dataset experiments are provided in Appendix F. Our code is available at https://github.com/liamyzq/DiT_time_series_imputation.

## 5.1 Gaussian Processes

We generate Gaussian process data with sequence length $H = 96$, dimension $d = 8$, and define the missing segment length as $|\mathcal{I}_{\text{miss}}| = 16$. In addition to applying Algorithm 1 to construct $95\%$ confidence regions (CRs), we sample from the true conditional distribution to evaluate CR coverage—the proportion of true values that fall within the estimated CR for comparison.

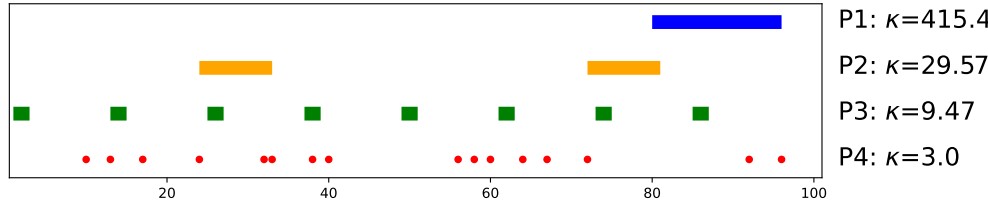

Figure 2: Visualization of the four missing patterns for a sequence of length 96. Each horizontal line shows the positions of missing values (highlighted in blue, orange, green and red for Patterns 1-4), and annotations on the right indicate the pattern number and its condition number $\kappa(\mathbf{\Sigma}_{\text{cond}})$.

We first vary two factors: training sample size $n \in \{10^3, 10^{3.5}, 10^4, 10^{4.5}, 10^5\}$, and missing patterns 1-4 (denoted as P1-P4) as shown in Figure 2. As discussed in Theorem 2, $\kappa(\mathbf{\Sigma}_{\text{cond}})$ acts as a key varying parameter. To mitigate distribution shift, we apply the same missing patterns to both training and test data. Results in Figure 3 show that small training sets ($n = 10^3, 10^{3.5}$) result in low variability and poor distribution estimation. As sample size increases, DiT yields CRs that significantly better match the true distribution. We further vary sequence length ($H$) and report the results in Table 1. The results suggest that CR coverage rate decreases as sequence length increases, which supports our theoretical findings. Regarding missing patterns, those with lower condition numbers reduce the sample complexity needed for effective estimation. These findings are consistent with our theory, suggesting that the conditional covariance condition number serves as a practical measure of estimation difficulty. Patterns with lower condition numbers retain richer temporal correlations, enabling accurate estimation with fewer samples.

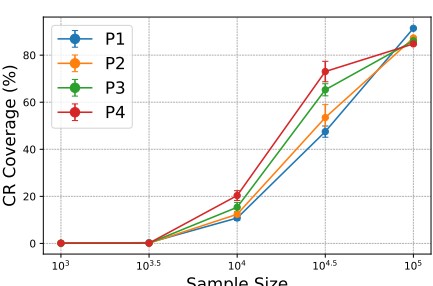

Figure 3: Percentage of real data samples that fall within the DiT-generated 95% CR.

Table 1: Sequence length vs CR coverage rates (%)(↑).

| H | 16 | 32 | 64 | 96 | 128 |
|---|---|---|---|---|---|
| CR | 92.67 (±1.95) | 88.63 (±2.01) | 82.14 (±1.70) | 80.25 (±1.64) | 77.81 (±1.87) |

Table 2: CR coverage rates (%) (↑) of models trained using different strategies on different missing patterns.

| | P1 | P2 | P3 | P4 |
|---|---|---|---|---|
| S1 | 34.58 (±1.22) | 58.46 (±1.89) | 72.42 (±1.66) | 80.25 (±1.64) |
| S2 | **66.22** (±3.86) | **83.71** (±2.86) | 74.04 (±1.90) | 81.50 (±2.12) |
| S3 | 56.04 (±6.48) | 81.05 (±2.09) | **74.59** (±1.27) | **83.09** (±1.48) |
| S4 | 57.27 (±5.34) | 79.00 (±2.42) | 74.38 (±3.00) | 82.74 (±2.40) |
| Only 8×2 | 36.74 (±1.31) | 60.51 (±1.65) | 71.24 (±1.52) | 80.46 (±2.01) |
| Only 4×4 | 34.15 (±1.16) | 59.23 (±1.88) | 73.08 (±1.10) | 79.83 (±1.84) |
| Only 1×16 | 32.68 (±1.50) | 54.23 (±1.76) | 69.46 (±1.53) | 76.72 (±2.20) |

**Mixed-Masking training strategy.** Based on our insights from our distribution shift analysis, we introduce mixed-masking training strategy. Remark 1 highlights that discrepancies between training and test distributions can impair CR estimation, especially in real-world settings with limited training data. A common practice is to train on fully random masks, which tend to have lower condition numbers and thus pose easier estimation tasks. However, this intensifies the mismatch with test cases featuring more challenging, clustered missing patterns, limiting model adaptability. To address this, we propose mixed-masking training strategy. Using the same $n = 10^5$ training samples, we evaluate the four test patterns in Figure 2. During training, we define four different mixed-masking strategies (each with 16 missing entries):

- **S1**: 100% random missing pattern (16×1, sixteen randomly placed missing entries).
- **S2**: 50% random (16×1) + 50% weakly grouped (8×2, eight randomly placed blocks of two consecutive missing entries).
- **S3**: 33.3% random (16×1) + 33.3% weakly grouped (8×2) + 33.3% moderately grouped (4×4, four randomly placed blocks of four consecutive missing entries).
- **S4**: 25% random (16×1) + 25% weakly grouped (8×2) + 25% moderately grouped (4×4) + 25% strongly grouped (1×16, one randomly placed block of sixteen consecutive missing entries).

Results in Table 2 show that models trained with mixed masking consistently outperform the baseline trained with completely random placed masks (S1). We also evaluate the strategies only containing individual patterns (8×2, 4×4, and 1×16 separately), and the results suggest that they yield inferior imputation performance compared to appropriately mixing different patterns. This supports our proposed mixed-masking strategies and aligns well with our theoretical insights. Yet determining optimal mixing ratios is instance based and remains an open question for future work.

Regarding how these strategies relate to our theoretical results, intuitively, different missing patterns during training lead to different training distributions $P_{\mathbf{x}_{\text{obs}}}$, resulting in varying condition numbers and consequently different DS values. Training with diverse missing patterns—ranging from easy to hard—helps the model adapt to imputation tasks with varying levels of difficulty by effectively covering more scenarios. As for a more concrete example, let us denote the training distributions corresponding to S1 and S4 as $P_{\mathbf{x}_{\text{obs}}}^{(1)}$ and $P_{\mathbf{x}_{\text{obs}}}^{(4)}$, respectively. Consider a test sample $\mathbf{x}_{\text{obs}}^*$ following the strongly grouped missing pattern P1 (consecutive missing entries). Intuitively, the resulting distribution $P_{\mathbf{x}_{\text{obs}}^*}$ is closer to $P_{\mathbf{x}_{\text{obs}}}^{(4)}$ than to $P_{\mathbf{x}_{\text{obs}}}^{(1)}$, which implies the distribution shift coefficient of $P_{\mathbf{x}_{\text{obs}}}^{(4)}$ is smaller than the one of $P_{\mathbf{x}_{\text{obs}}}^{(1)}$. Empirically, we calculate the average ratio across all test samples with missing pattern P1 and find that:

$$\frac{\mathsf{DS}(P_{\mathbf{x}_{\text{obs}}^*}, P_{\mathbf{x}_{\text{obs}}}^{(1)}, \mathcal{G})}{\mathsf{DS}(P_{\mathbf{x}_{\text{obs}}^*}, P_{\mathbf{x}_{\text{obs}}}^{(4)}, \mathcal{G})} \approx 47.93.$$

This clearly indicates that the mixed-masking training strategy (S4) yields significantly smaller distribution-shift coefficients compared to purely random missingness (S1). According to Corollary 1, this provides strong theoretical support for the superior empirical performance achieved by our mixed-masking strategy.

### 5.2 Latent Gaussian Processes

We conduct additional experiments to assess whether our findings generalize beyond the theoretical setting—specifically, whether different missing patterns affect imputation and uncertainty quantification performance, and whether the mixed-masking training strategy improves them. For $\mathbf{X}$ drawn from the Gaussian process in Section 5.1, we consider a corresponding latent Gaussian process: $\mathbf{Y} = \phi(\mathbf{X}) + \boldsymbol{\epsilon}$ with $\text{vec}(\boldsymbol{\epsilon}) \sim \mathcal{N}(\mathbf{0}, 0.1 \cdot \mathbf{I}_{dH})$, where the non-linear transform $\phi(x) = x + \exp(-x^2) + 2\sin(x)$ is applied entry-wise. We adopt a training sample size of $n = 10^5$. This introduces nonlinearity and noise, increasing the difficulty of distribution estimation.

We evaluate DiT on this transformed dataset using the same four missing patterns and four training strategies from Section 5.1. For comparison, we implement two representative generative imputation models—CSDI [Tashiro et al., 2021] and GPVAE [Fortuin et al., 2020], ensuring all models have comparable numbers of trainable parameters. We report Mean Squared Error (MSE) against the true conditional mean and CR coverage rates, following the setup in Section 5.1. Results are shown in

Table 3: MSE (↓) on latent Gaussian process data.

| | | DiT | CSDI | GPVAE |
|---|---|---|---|---|
| | S1 | 0.70 (±0.03) | 0.75 (±0.03) | 5.24 (±0.75) |
| | S2 | 0.68 (±0.02) | 0.69 (±0.02) | 5.45 (±1.05) |
| P1 | S3 | **0.67 (±0.03)** | 0.70 (±0.03) | 5.13 (±0.49) |
| | S4 | 0.67 (±0.02) | 0.68 (±0.02) | 5.28 (±0.68) |
| | S1 | 0.64 (±0.03) | 0.66 (±0.03) | 5.09 (±0.70) |
| | S2 | **0.62 (±0.02)** | 0.63 (±0.03) | 5.01 (±0.62) |
| P2 | S3 | 0.60 (±0.03) | 0.62 (±0.02) | 4.94 (±0.56) |
| | S4 | 0.62 (±0.03) | 0.63 (±0.03) | 4.84 (±0.60) |
| | S1 | 0.62 (±0.02) | 0.65 (±0.02) | 4.63 (±0.58) |
| | S2 | 0.60 (±0.03) | 0.64 (±0.03) | 5.12 (±1.00) |
| P3 | S3 | **0.58 (±0.02)** | 0.63 (±0.03) | 4.50 (±0.52) |
| | S4 | 0.58 (±0.03) | 0.61 (±0.02) | 4.59 (±0.54) |
| | S1 | 0.56 (±0.01) | 0.59 (±0.03) | 4.89 (±0.69) |
| | S2 | **0.53 (±0.03)** | 0.60 (±0.02) | 4.79 (±0.61) |
| P4 | S3 | 0.53 (±0.01) | 0.58 (±0.03) | 4.39 (±0.49) |
| | S4 | 0.53 (±0.02) | 0.58 (±0.02) | 4.45 (±0.54) |

Table 4: CR coverage rates (%) (↑).

| | | DiT | CSDI |
|---|---|---|---|
| | S1 | 36.46 (±1.62) | 54.75 (±1.89) |
| | S2 | 53.68 (±3.26) | 56.68 (±2.75) |
| P1 | S3 | 54.26 (±2.79) | **58.64 (±3.11)** |
| | S4 | 56.43 (±3.76) | 55.67 (±4.03) |
| | S1 | 55.81 (±1.55) | 63.67 (±1.77) |
| | S2 | 65.77 (±2.87) | 64.89 (±3.43) |
| P2 | S3 | **66.24 (±3.22)** | 63.13 (±2.95) |
| | S4 | 63.95 (±4.38) | 65.97 (±3.59) |
| | S1 | 63.53 (±1.72) | 61.35 (±1.49) |
| | S2 | 71.29 (±2.99) | 65.69 (±2.79) |
| P3 | S3 | 70.89 (±2.45) | 63.48 (±2.90) |
| | S4 | **73.36 (±4.37)** | 67.17 (±3.93) |
| | S1 | 76.46 (±1.33) | 68.60 (±1.74) |
| | S2 | 78.63 (±2.62) | 70.48 (±2.34) |
| P4 | S3 | 78.79 (±2.67) | 73.46 (±2.53) |
| | S4 | **80.64 (±3.72)** | 72.89 (±3.78) |

Tables 3 and 4. Since GPVAE performs poorly in point estimation, we omit its CR coverage. DiT consistently outperforms in both MSE and CR coverage, indicating transformers may better suit this task than CSDI's convolutional design. Moreover, mixed-masking training improves performance not only for DiT but also for other models, demonstrating its broader benefit. These findings reinforce our conclusions from Gaussian process experiments and support the generalization of our theory and training methodology to more complex, nonlinear settings.

## 6 Conclusion and Discussion

Our work addresses a critical gap in the theoretical understanding of diffusion-based time series imputation and uncertainty quantification by investigating the statistical efficiency of diffusion transformers on Gaussian process data. This result enables efficient and accurate imputation and confidence region construction. Motivated by the theory, we propose a mixed-masking training strategy that introduces diverse missing patterns during training, rather than relying solely on completely random masks. Our experiments validate the theoretical findings and further demonstrate that the proposed strategy performs well and generalizes to more complex data beyond our analytical scope.

Looking ahead, investigating the behavior of diffusion transformers on heavy-tailed time series (e.g., financial data) would further clarify their limitations and guide practical design choices. Moreover, a more detailed analysis of optimal mixed-masking training strategies—especially those leveraging prior knowledge—could significantly improve the performance of imputation models.

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

## A Related Work

In the early stages of time series imputation, statisticians developed a wide range of traditional statistical methods aimed at both imputation (point estimation) and quantifying uncertainty, often by leveraging well-established statistical tools to construct confidence intervals [Cox et al., 1981, Shumway et al., 2000]. Initial techniques were relatively simple, such as imputing missing values using the mean or median of observed entries. These were later followed by more advanced interpolation approaches based on regression models, including linear regression and splines [Shumway et al., 2000]. To better exploit the spatio-temporal structure inherent in time series data, model-based methods emerged, such as ARIMA, GARCH, Kalman filters, and Bayesian inference frameworks [Fuller, 2009]. These approaches are advantageous for their interpretability, ability to incorporate domain knowledge, and support for formal statistical testing. Moreover, many of them naturally allow for uncertainty quantification through predictive intervals or posterior distributions. However, these methods come with notable limitations: they typically rely on strong assumptions about stationarity, linearity, or noise distributions, making them less effective for complex real-world data with nonlinear or high-dimensional spatio-temporal dependencies [Anderson, 2011]. Additionally, their computational cost often scales poorly with data dimensionality, posing challenges for modern large-scale applications.

To address the limitations of statistical approaches, machine-based imputation methods have become increasingly popular in recent years. Early approaches include classical machine learning models [Jerez et al., 2010] such as support vector machines [Wu et al., 2015] and tree-based methods (including bagging and boosting techniques) [Vateekul and Sarinnapakorn, 2009, Yang et al., 2017]. With the advancement of model architectures and increasing computational power, deep learning-based models have gained prominence for their ability to capture complex temporal dependencies [Fang and Wang, 2020, Wang et al., 2024, Du et al., 2024]. Predictive models such as RNNs [Che et al., 2018, Yoon et al., 2018b, Cao et al., 2018], CNNs [Wu et al., 2022, Fu et al., 2024a], GNNs [Cini et al., 2021], and transformer-based networks [Bansal et al., 2021, Du et al., 2023] directly estimate missing values using well-designed architectures. Generative imputation methods model the distribution of missing data and perform better in quantifying uncertainty; representative techniques include GAN-based methods [Luo et al., 2018, Yoon et al., 2018a, Miao et al., 2021], VAE-based approaches [Mattei and Frellsen, 2019, Fortuin et al., 2020, Mulyadi et al., 2021, Peis et al., 2022, Kim et al., 2023], and diffusion models. Among diffusion approaches, CSDI [Tashiro et al., 2021] introduced conditional diffusion for time series imputation, and subsequent work [Alcaraz and Strodthoff, 2022, Wang et al., 2023, Liu et al., 2023, Zhou et al., 2024] improved conditioning strategies and computational efficiency. DiT [Peebles and Xie, 2022, Cao et al., 2024] extends this line by integrating a transformer backbone into the diffusion framework, achieving better imputation accuracy and uncertainty quantification. These methods resolve certain issues and perform well empirically, however, are still limited by lacks of uncertainty quantification in many methods and theoretical understanding.

Our work also contributes towards the theoretical foundations of diffusion models [Chen et al., 2024, Tang and Zhao, 2024]. Some prior works have established sample efficiency and learning guarantees for diffusion models when modeling the original data distribution. Chen et al. [2022], Benton et al. [2023], Li et al. [2024] show that the generated distribution remains close to the target distribution, assuming access to an relatively accurate score function. By incorporating score approximation procedures and corresponding theoretical analysis, Chen et al. [2023], Oko et al. [2023], Mei and Wu [2025] provide end-to-end guarantees, covering various types of data including manifold data and graphical models. In the case of conditional diffusion models, sharp statistical bounds of distribution estimation have been derived in Fu et al. [2024c]. Additionally, Fu et al. [2024b] explores the theoretical regime of modeling spatio-temporal dependencies in sequential data. However, these results do not directly apply to more concrete and complex scenarios, such as how conditional DiT models can learn intricate dependencies to accomplish time series imputation tasks.

## B Proof of Lemma 1

We provide the detailed proof of Lemma 1 in this section.

To simplify our analysis, we begin by making some assumptions. Firstly, without loss of generality, we assume the mean of the Gaussian process data $\boldsymbol{\mu} = \mathbf{0}$. Large norms in $\mathbf{x}$ and $\mathbf{v}_t$ often lead to training instability, making it practical to perform clipping. Inspired by this, leveraging the Gaussian and light-tailed nature of $\mathbf{x}$ and $\mathbf{v}_t$, we truncate the domain of the data and diffused samples by

defining an event that occurs with high probability $1 - \delta$:

$$\mathcal{C}_\delta = \{\|\mathbf{x}\|_2 \le C_{\mathrm{data}}^\delta, \|\mathbf{v}_t\|_2 \le C_{\mathrm{data}}^\delta\}, \tag{13}$$

where $C_{\mathrm{data}}^\delta = \mathcal{O}(\sqrt{Hd})$ is a threshold depending on $\delta$. **Our score approximation analysis of Lemma 1 and Theorem 1 is conducted under the condition of event** $\mathcal{C}_\delta$ (ensuring the conclusions hold with high probability $1 - \delta$), which significantly simplifies the process. The relationship between the truncation range $C_{\mathrm{data}}^\delta$, and high probability $\delta$ is deferred to Lemma 11. Outside event $\mathcal{C}_\delta$ (i.e., on $\mathcal{C}_\delta^c$), the unbounded range complicates obtaining a meaningful score approximation in the second-norm sense. However, as $\mathcal{C}_\delta^c$ occurs with a small probability, we can still achieve reliable results in distribution estimation, where evaluation is based on expectation.

**Some Useful Results**   In this part, we present some key results regarding the eigenvalues and condition numbers of covariance matrices, which will be instrumental in our analysis.

We first define:

$$\kappa_t := \frac{\lambda_{\max}\left(\alpha_t^2 \boldsymbol{\Sigma}_{\mathrm{cond}} + \sigma_t^2 \mathbf{I}\right)}{\lambda_{\min}\left(\alpha_t^2 \boldsymbol{\Sigma}_{\mathrm{cond}} + \sigma_t^2 \mathbf{I}\right)}.$$

Using the positive definiteness of $\boldsymbol{\Gamma}$ and $\boldsymbol{\Lambda}$, we obtain:

$$\lambda_{\max}(\boldsymbol{\Lambda}) = \|\boldsymbol{\Lambda}\|_2 > 0, \quad \lambda_{\min}(\boldsymbol{\Lambda}) = \|\boldsymbol{\Lambda}^{-1}\|_2^{-1} > 0.$$

Furthermore, by the properties of the Kronecker product, we derive:

$$\lambda_{\max}(\boldsymbol{\Sigma}_{\mathrm{obs}}) = \lambda_{\max}(\boldsymbol{\Gamma}_{\mathrm{obs}})\lambda_{\max}(\boldsymbol{\Lambda}), \quad \lambda_{\min}(\boldsymbol{\Sigma}_{\mathrm{obs}}) = \lambda_{\min}(\boldsymbol{\Gamma}_{\mathrm{obs}})\lambda_{\min}(\boldsymbol{\Lambda}),$$

$$\kappa(\boldsymbol{\Sigma}_{\mathrm{obs}}) = \frac{\lambda_{\max}(\boldsymbol{\Lambda})\lambda_{\max}(\boldsymbol{\Gamma}_{\mathrm{obs}})}{\lambda_{\min}(\boldsymbol{\Lambda})\lambda_{\min}(\boldsymbol{\Gamma}_{\mathrm{obs}})} = \kappa(\boldsymbol{\Gamma}_{\mathrm{obs}})\kappa(\boldsymbol{\Lambda}).$$

Finally, we assume:

$$\lambda_{\max}(\boldsymbol{\Gamma}_{\mathrm{obs}}), \lambda_{\max}(\boldsymbol{\Gamma}_{\mathrm{miss}}), \lambda_{\max}(\boldsymbol{\Lambda}) = \mathcal{O}(1).$$

## B.1   Key Steps for Proving Lemma 1

In Lemma 1, we aim to show that the gradient-based Algorithm 2 provides a good approximation of the conditional score function.

The algorithm employs gradient descent to solve two types of optimization problems: the major GD problem (7), and the auxiliary GD problem (9), which is solved within each update step of the major GD. It is critical to note that the major GD updates are inherently noisy due to various reasons, such as the auxiliary GD approximating certain quantities at each step, and later using transformers to approximate each step. Therefore, to establish the result in Lemma 1, our proof consists of two key steps:

**Step 1.** We demonstrate that, with a sufficient number of auxiliary iterations $K_{\mathrm{aux}}$, the approximation error of the auxiliary GD loop's result can be controlled below a specified threshold.

**Step 2.** We then show that, by controlling the perturbation level in each major GD update step, the score approximation error (i.e., the gap between the output of the major GD and the ground truth score function) can also be bounded, provided there are enough major iterations $N$.

In the following, we elaborate on each step by providing precise statements and subsequently use them to prove Lemma 1. All supporting results are deferred to later sections.

## B.2   Detailed Statements in Steps 1-2 and Proof of Lemma 1

Now we present formal statements in **Step 1-2** and use them to prove Lemma 1.

### B.2.1   Formal Statements in Steps 1-2

This section contains the statements of Lemma 2 and Lemma 3.

**Lemma 2.** *For an arbitrarily fixed time $t \in (0, T]$ and given an error tolerance $\epsilon_0 \in (0, 1)$, let $\mathbf{b} := \boldsymbol{\Sigma}_{\mathrm{cor}}\mathbf{s}^{(k)} \in \mathbb{R}^{d|\mathcal{I}_{\mathrm{miss}}|}$, running the auxiliary gradient descent in (10) with a suitable step size $\theta = 2/(\lambda_{\min}(\boldsymbol{\Sigma}_{\mathrm{obs}}) + \lambda_{\max}(\boldsymbol{\Sigma}_{\mathrm{obs}}))$ for*

$$K_{\mathrm{aux}} = \left\lceil \frac{\kappa(\boldsymbol{\Sigma}_{\mathrm{obs}}) + 1}{2} \log\left(\frac{\|\mathbf{b}\|_2}{\lambda_{\min}(\boldsymbol{\Gamma}_{\mathrm{obs}})\lambda_{\min}(\boldsymbol{\Lambda})\epsilon_0}\right) \right\rceil$$

*iterations produces a solution $\mathbf{u}^{(K_{\mathrm{aux}})}$ that satisfies*

$$\|\mathbf{u}^{(K_{\mathrm{aux}})} - \mathbf{u}\|_2 \le \epsilon_0.$$

Here, we introduce $\epsilon_0$ to distinguish the noise arising from the auxiliary GD loop approximation from the error level $\epsilon$ stated in Lemma 1. This distinction provides additional flexibility to adjust $\epsilon_0$ in subsequent proofs.

Next, we establish a lemma for the convergence of the major GD. In each major GD step (referring to (8)), we incorporate an error term and represent our gradient update as:

$$\mathbf{s}^{(k+1)} = \mathbf{s}^{(k)} - \eta_t \nabla \mathcal{L}_t(\mathbf{s}^{(k)}) + \xi^{(k)}, \tag{14}$$

where $\xi^{(k)}$ represents the error term in each perturbed major GD step. Explicitly accounting for the noise present in each perturbed gradient step, we can establish:

**Lemma 3.** *For an arbitrarily fixed time $t \in (0, T]$ and given an error tolerance $\epsilon \in (0, 1)$, suppose $\|\xi^{(k)}\|_2 \leq \epsilon$, then running the major gradient descent in (14) and a suitable step size $\eta_t = 2/(\lambda_{\min}(\alpha_t^2 \boldsymbol{\Sigma}_{\mathrm{cond}} + \sigma_t^2 \mathbf{I}) + \lambda_{\max}(\alpha_t^2 \boldsymbol{\Sigma}_{\mathrm{cond}} + \sigma_t^2 \mathbf{I}))$ for*

$$K = \mathcal{O}\left(\kappa_t \log\left(\frac{Hd\kappa(\boldsymbol{\Lambda})\kappa(\boldsymbol{\Gamma}_{\mathrm{obs}})}{\sigma_t \epsilon}\right)\right)$$

*iterations produces a solution $\mathbf{s}^{(K)}$ that satisfies*

$$\|\mathbf{s}^{(K)} - \mathbf{s}\|_2 \leq \left(\frac{\kappa_t}{2} + 1\right)\epsilon,$$

*where $\kappa_t := \kappa(\alpha_t^2 \boldsymbol{\Sigma}_{\mathrm{cond}} + \sigma_t^2 \mathbf{I})$.*

With the convergence of both the auxiliary and major GD established, we are ready to prove Lemma 1.

### B.2.2 Proof of Lemma 1

*Proof.* By the statement in Lemma 3, we need to control the noise level in each major GD step, i.e. ensure $\xi^{(k)} \leq \epsilon$. We analyze this error as

$$\begin{aligned}\|\xi^{(k)}\|_2 &\leq \|\alpha_t^2 \boldsymbol{\Sigma}_{\mathrm{cor}}^\top \boldsymbol{\Sigma}_{\mathrm{obs}}^{-1} \boldsymbol{\Sigma}_{\mathrm{cor}} \mathbf{s}^{(k)} - \alpha_t^2 \boldsymbol{\Sigma}_{\mathrm{cor}}^\top \widehat{\boldsymbol{\Sigma}}_{\mathrm{obs}}^{-1} \boldsymbol{\Sigma}_{\mathrm{cor}} \mathbf{s}^{(k)}\|_2 \\ &\quad + \|\boldsymbol{\Sigma}_{\mathrm{cor}}^\top \boldsymbol{\Sigma}_{\mathrm{obs}}^{-1} \mathbf{x}_{\mathrm{obs}} - \boldsymbol{\Sigma}_{\mathrm{cor}}^\top \widehat{\boldsymbol{\Sigma}}_{\mathrm{obs}}^{-1} \mathbf{x}_{\mathrm{obs}}\|_2 \\ &\leq \alpha_t^2 \|\boldsymbol{\Sigma}_{\mathrm{cor}}^\top\|_2 \|\boldsymbol{\Sigma}_{\mathrm{obs}}^{-1} \boldsymbol{\Sigma}_{\mathrm{cor}} \mathbf{s}^{(k)} - \widehat{\boldsymbol{\Sigma}}_{\mathrm{obs}}^{-1} \boldsymbol{\Sigma}_{\mathrm{cor}} \mathbf{s}^{(k)}\|_2 \\ &\quad + \|\boldsymbol{\Sigma}_{\mathrm{cor}}^\top\|_2 \|\boldsymbol{\Sigma}_{\mathrm{obs}}^{-1} \mathbf{x}_{\mathrm{obs}} - \widehat{\boldsymbol{\Sigma}}_{\mathrm{obs}}^{-1} \mathbf{x}_{\mathrm{obs}}\|_2.\end{aligned}$$

Here, the latter term arises from approximating $\boldsymbol{\mu}_{\mathrm{cond}}(\mathbf{x}_{\mathrm{obs}})$, and $\widehat{\boldsymbol{\Sigma}}_{\mathrm{obs}}^{-1}(\mathbf{x}_{\mathrm{obs}} - \boldsymbol{\mu}_{\mathrm{obs}})$ represents the $K_{\mathrm{aux}}$-iteration auxiliary GD approximation of the matrix-vector product.

We provide a useful lemma to help control the error above.

**Lemma 4.** *For an arbitrarily fixed time $t \in (0, T]$, we have*

$$\|\mathbf{v}_t - \boldsymbol{\mu}_{\mathrm{cond}}(\mathbf{x}_{\mathrm{obs}})\|_2 \leq \left(1 + \frac{\|\boldsymbol{\Gamma}_{\mathrm{cor}}\|_2 \kappa(\boldsymbol{\Lambda})}{\lambda_{\min}(\boldsymbol{\Gamma}_{\mathrm{obs}})}\right) C_{\mathrm{data}}^\delta,$$

*and*

$$\|\mathbf{s}_t\|_2 \leq \sigma_t^{-2} \|\mathbf{v}_t - \boldsymbol{\mu}_{\mathrm{cond}}(\mathbf{x}_{\mathrm{obs}})\|_2.$$

Invoking Lemma 2 and Lemma 4, letting $\epsilon_0 = \left((\alpha_t^2 + 1)\|\boldsymbol{\Sigma}_{\mathrm{cor}}\|_2\right)^{-1}\epsilon$, to ensure that $\|\xi^{(k)}\|_2 \leq \epsilon$., we can bound the required auxiliary iteration steps by:

$$\begin{aligned}K_{\mathrm{aux}} &= \frac{\kappa(\boldsymbol{\Sigma}_{\mathrm{obs}}) + 1}{2} \log\left(\frac{\sqrt{Hd}(C_{\mathrm{data}}^\delta + \|\mathbf{s}\|_\infty)(\alpha_t^2 + 1)\|\boldsymbol{\Sigma}_{\mathrm{cor}}\|_2^2}{\lambda_{\min}(\boldsymbol{\Gamma}_{\mathrm{obs}})\lambda_{\min}(\boldsymbol{\Lambda})\epsilon}\right) \\ &= \mathcal{O}\left(\kappa(\boldsymbol{\Lambda})\kappa(\boldsymbol{\Gamma}_{\mathrm{obs}}) \log\left(\frac{Hd\kappa(\boldsymbol{\Lambda})\kappa(\boldsymbol{\Gamma}_{\mathrm{obs}})}{\sigma_t \epsilon}\right)\right).\end{aligned}$$

Lastly, we invoke Lemma 3, substitute $\epsilon$ with $\epsilon = \sigma_t^{-1}\left(\frac{2}{\kappa_t + 2}\right)\epsilon$, we have

$$K = \mathcal{O}\left(\kappa_t \log\left(\frac{Hd\kappa_t\kappa(\boldsymbol{\Gamma}_{\mathrm{obs}})\kappa(\boldsymbol{\Lambda})}{\epsilon}\right)\right).$$

Finally, notice that

$$\kappa_t = \frac{\lambda_{\max}(\alpha_t^2 \mathbf{\Sigma}_{\text{cond}} + \sigma_t^2 \mathbf{I})}{\lambda_{\min}(\alpha_t^2 \mathbf{\Sigma}_{\text{cond}} + \sigma_t^2 \mathbf{I})} \leq \kappa(\mathbf{\Sigma}_{\text{cond}}),$$

and leveraging

$$\kappa(\mathbf{\Sigma}_{\text{obs}}) = \kappa(\mathbf{\Gamma}_{\text{obs}})\kappa(\mathbf{\Lambda}),$$

we have

$$K = \mathcal{O}\Big(\kappa(\mathbf{\Sigma}_{\text{cond}}) \log \Big(\frac{Hd\kappa(\mathbf{\Sigma}_{\text{cond}})\kappa(\mathbf{\Sigma}_{\text{obs}})}{\epsilon}\Big)\Big), K_{\text{aux}} = \mathcal{O}\Big(\kappa(\mathbf{\Sigma}_{\text{obs}}) \log \Big(\frac{Hd\kappa(\mathbf{\Sigma}_{\text{obs}})}{\sigma_t \epsilon}\Big)\Big).$$

This completes the proof of Lemma 1. □

### B.3  Proofs of Lemma 2 and Lemma 3

To prove the lemmas, we first state a standard result in convex optimization.

**Lemma 5** (Theorem 3.12 in [Bubeck et al., 2015]). *Let $f$ be $\beta$-smooth and $\alpha$-strongly convex on $\mathbb{R}^d$ and $\mathbf{x}^*$ be the global minimizer. Then gradient descent with $\eta = \frac{2}{\alpha+\beta}$ satisfies*

$$\left\|\mathbf{x}^{(k+1)} - \mathbf{x}^*\right\|_2 \leq \left(\frac{\kappa - 1}{\kappa + 1}\right) \left\|\mathbf{x}^{(k)} - \mathbf{x}^*\right\|_2, \quad k = 0, 1, \ldots,$$

*where $\mathbf{x}^{(k+1)} = \mathbf{x}^{(k)} - \eta \nabla f(\mathbf{x}^{(k)})$ is the outcome at the $(k+1)$-th iteration of gradient descent, and $\kappa = \frac{\beta}{\alpha}$.*

Equipped with this lemma, the proof process is straightforward.

#### B.3.1  Proof of Lemma 2

*Proof.* Referring to (10) (expression of auxiliary GD step), the update steps are

$$\mathbf{u}^+ = \mathbf{u} - \theta \nabla \mathcal{L}_{\text{inner}}(\mathbf{u})$$
$$= \mathbf{u} - \theta \left(\mathbf{\Sigma}_{\text{obs}} \mathbf{u} - \mathbf{b}\right)$$

We should notice that $\mathcal{L}_{\text{aux}}$ is $\lambda_{\max}(\mathbf{\Sigma}_{\text{obs}})$-smooth and $\lambda_{\min}(\mathbf{\Sigma}_{\text{obs}})$-strongly convex. Then by Lemma 5, we have

$$\|\mathbf{u}^{(k_{\text{aux}}+1)} - \mathbf{u}\|_2 \leq \left(\frac{\kappa(\mathbf{\Sigma}_{\text{obs}}) - 1}{\kappa(\mathbf{\Sigma}_{\text{obs}}) + 1}\right) \|\mathbf{u}^{(k_{\text{aux}})} - \mathbf{u}\|_2$$

$$= (1 - \frac{2}{\kappa(\mathbf{\Sigma}_{\text{obs}}) + 1})^{k_{\text{aux}}+1} \|\mathbf{u}^{(0)} - \mathbf{u}\|_2$$

$$\leq \exp\left\{\frac{-2(k_{\text{aux}} + 1)}{\kappa(\mathbf{\Sigma}_{\text{obs}}) + 1}\right\} \|\mathbf{u}\|_2.$$

We also have

$$\|\mathbf{u}\|_2 = \|\mathbf{\Sigma}_{\text{obs}}^{-1} \mathbf{b}\|_2 \leq \|\mathbf{\Sigma}_{\text{obs}}\|_2^{-1} \|\mathbf{b}\|_2 \leq \lambda_{\min}(\mathbf{\Gamma}_{\text{obs}})\lambda_{\min}(\mathbf{\Lambda})^{-1} \|\mathbf{b}\|_2.$$

With preset error $\epsilon_0 > 0$, taking number of iterations $K_{\text{aux}} \geq \lceil \frac{\kappa(\mathbf{\Sigma}_{\text{obs}})+1}{2} \log(\frac{\|\mathbf{b}\|_2}{\lambda_{\min}(\mathbf{\Gamma}_{\text{obs}})\lambda_{\min}(\mathbf{\Lambda})\epsilon_0}) \rceil$, we obtain

$$\|\mathbf{u}^{(K_{\text{aux}})} - \mathbf{u}\|_2 \leq \epsilon_0.$$

□

#### B.3.2  Proof of Lemma 3

*Proof.* In each step, we incorporate an error term and represent our gradient update as in (14):

$$\mathbf{s}^{(k+1)} = \mathbf{s}^{(k)} - \eta \nabla \mathcal{L}_t(\mathbf{s}^{(k)}) + \xi^{(k)}.$$

We should also notice that $\mathcal{L}_t$ is $\lambda_{\max}\left(\alpha_t^2 \mathbf{\Sigma}_{\text{cond}} + \sigma_t^2 \mathbf{I}\right)$-smooth and $\lambda_{\min}\left(\alpha_t^2 \mathbf{\Sigma}_{\text{cond}} + \sigma_t^2 \mathbf{I}\right)$-strongly convex. Then with $\|\xi^{(k)}\|_2 \leq \epsilon$, by Lemma 5,

$$\|\mathbf{s}^{(k+1)} - \mathbf{s}\|_2 \leq \|\mathbf{s}^{(k)} - \eta\nabla\mathcal{L}_t(\mathbf{s}^{(k)})\|_2 + \|\xi^{(k)}\|_2,$$
$$\leq \|\mathbf{s}^{(k)} - \eta\nabla\mathcal{L}_t(\mathbf{s}^{(k)})\|_2 + \epsilon,$$
$$\leq \left(\frac{\kappa_t - 1}{\kappa_t + 1}\right)\|\mathbf{s}^{(k)} - \mathbf{s}\|_2 + \epsilon.$$

Then we have

$$\|\mathbf{s}^{(k+1)} - \mathbf{s}\|_2 - \frac{\kappa_t + 1}{2}\epsilon \leq \left(\frac{\kappa_t - 1}{\kappa_t + 1}\right)^{n+1}\left(\|\mathbf{s}^{(0)} - \mathbf{s}\|_2 - \frac{\kappa_t + 1}{2}\epsilon\right).$$

Similar to the proof of Lemma 2, we obtain

$$\|\mathbf{s}^{(K)} - \mathbf{s}\|_2 - \frac{\kappa_t + 1}{2}\epsilon \leq \exp\left\{\frac{-2N}{\kappa_t + 1}\right\}\left(\|\mathbf{s}\|_2 - \frac{\kappa_t + 1}{2}\epsilon\right) \leq \exp\left\{\frac{-2N}{\kappa_t + 1}\right\}(\|\mathbf{s}\|_2).$$

By invoking Lemma 4, we also have

$$\|\mathbf{s}\|_2 = \|(\alpha_t^2\boldsymbol{\Sigma}_{\text{cond}} + \sigma_t^2\mathbf{I})\|_2^{-1}\|(\mathbf{v}_t - \alpha_t\boldsymbol{\mu}_{\text{cond}})\|_2 \leq \sigma_t^{-2}\left[\left(1 + \frac{\kappa(\boldsymbol{\Lambda})}{\lambda_{\min}(\boldsymbol{\Gamma}_{\text{obs}})}\right)C_{\text{data}}^\delta\right].$$

Lastly, taking $K = \mathcal{O}\left(\kappa_t \log\left(\frac{Hd\kappa(\boldsymbol{\Lambda})\kappa(\boldsymbol{\Gamma}_{\text{obs}})}{\sigma_t\epsilon}\right)\right)$, we obtain

$$\|\mathbf{s}^{(K)} - \mathbf{s}\|_2 \leq \left(\frac{\kappa_t}{2} + 1\right)\epsilon.$$

This finishes the proof of Lemma 3. □

# C  Proof of Theorem 1

We provide the detailed proof of Theorem 1 in this section by explicitly construct a transformer architecture to unroll Algorithm 2. Firstly, we assume that the mean function $\boldsymbol{\mu}_{\text{obs}}, \boldsymbol{\mu}_{\text{miss}}$ can be constructed by an additional preprocessing network. Thus, the analysis in this section can also be conducted under the condition of event $\mathcal{C}_\delta$ and still assuming $\boldsymbol{\mu} = \mathbf{0}$ as stated in Appendix B.

## C.1  Key Steps for Proving Theorem 1

The proof of Theorem 1 is presented in a constructive framework. Revisiting the architecture in (5), we observe that it comprises the encoder $f_{\text{in}}$, which transforms the original input into a form compatible with the unrolling of Algorithm 2; the raw transformer blocks, which perform the algorithm unrolling; and the decoder $f_{\text{out}}$, which extracts and truncates the output to provide the final score approximation.

We define the major GD step with $k = 1$ as the first major GD step and those with $k > 1$ as the later major GD steps. Similarly, we categorize the auxiliary GD steps. Notably, the first major GD step is relatively simpler, while the later major GD steps are analogous to it. Accordingly, we separate our analysis into the first and later major GD steps. To establish Theorem 1, the proof proceeds through the following steps:

**Step 1.** Construct the encoder, decoder, and essential components that are critical for constructing the subsequent raw transformer architectures.

**Step 2.** Construct the raw transformer architecture for the first major GD step.

**Step 3.** Construct the raw transformer architecture for the later major GD steps analogously.

**Step 4.** Analyze the error and configuration of the raw transformer architectures constructed in the previous steps.

**Step 5.** Summarize the constructions and analyses to establish the result in Theorem 1.

## C.2  Constructing Encoder, Decoder and Some Crucial Transformer Components

For sake of simplicity, given a time step $t \in (t_0, T]$, we denote $(\mathbf{v}_t)_j = \mathbf{x}_j \in \mathbb{R}^d$ in the following analysis. Additionally, we define each $(\text{FFN}_l \circ \text{Attn}_l)$ as a transformer block. The architecture composed solely of transformer blocks, excluding the encoder $f_{\text{in}}$ and decoder $f_{\text{out}}$, is referred to as the raw transformer, denoted by $\mathcal{T}_{\text{raw}}(D, L, M, B)$.

**Encoder**  The encoder we need is to mapping our input $\mathbf{x}$ to higher dimensions in an attepmpt to include some useful values (e.g. time embeddings) and also some buffer spaces to finish the gradient descent process. For simplicity, at a specific time $t$, we suppose the encoder converts the initial input into $\mathbf{Y} = f_{\text{in}}([\mathbf{x}_1, \mathbf{x}_2, \ldots, \mathbf{x}_H, t]) = [\mathbf{y}_1^\top, \ldots, \mathbf{y}_N^\top]^\top \in \mathbb{R}^{D \times H}$, which satisfies

$$\mathbf{y}_i = \left[ \mathbf{x}_i^\top, \mathbf{e}_i^\top, \phi(t)^\top, \mathbf{0}_{6d}^\top, 1, 0, 1, \mathbf{x}_i^\top, \mathbf{0}_{4d}^\top \right],$$
$$\mathbf{y}_j = \left[ \mathbf{x}_j^\top, \mathbf{e}_i^\top, \phi(t)^\top, \mathbf{0}_{6d}^\top, 1, 0, 1, \mathbf{0}_{5d}^\top \right].$$

where $\phi(t) = [\eta_t, \alpha_t, \sigma_t^2, \alpha_t^2]^\top \in \mathbb{R}^{d_t}$ with $d_t = 4$. Specifically, we use different subscriptions for observed indices and missing indices, i.e. $i \in \mathcal{I}_{\text{obs}}, j \in \mathcal{I}_{\text{miss}}$. For simplicity, we omitted the subscript $t$ here, and $\mathbf{0}_{6d}^\top, \mathbf{0}_{5d}^\top, \mathbf{0}_{4d}^\top$ serve as the buffer space for storing the components necessary for unrolling the algorithm.

**Decoder**  Suppose the output tokens from the transformer blocks has produced a conditional score approximator in matrix shape, and the stability for the computation inside the network, we design the decoder $f_{\text{out}} = f_{\text{norm}} \circ f_{\text{linear}}$. Where $f_{\text{linear}} : \mathbb{R}^{D \times H} \to \mathbb{R}^{d|\mathcal{I}_{\text{miss}}|}$ extracts and flattens the input into a vector that aligns with the dimension of the conditional score function, and $f_{\text{norm}} : \mathbb{R}^{d|\mathcal{I}_{\text{miss}}|} \to \mathbb{R}^{d|\mathcal{I}_{\text{miss}}|}$ controls the output range of the network by the upper bound of the score function. By Lemma 4, denote $\sigma_t^{-2} \left( 1 + \frac{\kappa(\boldsymbol{\Lambda})}{\lambda_{\min}(\boldsymbol{\Gamma}_{\text{obs}})} \right) C_{\text{data}}^\delta$ as $R_t$, we can set

$$f_{\text{norm}}(\mathbf{s}) = \begin{cases} \mathbf{s}, & \text{if } \|\mathbf{s}\|_2 \leq R_t, \\ \frac{R_t}{\|\mathbf{s}\|_2} \mathbf{s}, & \text{otherwise.} \end{cases}$$

**$\mathbf{f}_{\text{mult}}$ Module**  At the end of this part, we also provide the construction of the multiplication module, which approximates the product between scalars and vectors. This is a crucial component in constructing $f_{\text{GD}}$ later. We introduce a lemma, which is a modified version of Corollary 3 in [Fu et al., 2024b]:

**Lemma 6.** *Suppose input to be* $\mathbf{Y} = [\mathbf{y}_1, \mathbf{y}_2, \cdots, \mathbf{y}_H] \in \mathbb{R}^{D \times H}$ *with* $\mathbf{y}_i = [\mathbf{x}_i^\top, \mathbf{0}_{3d}^\top, w_i, \mathbf{z}_i^\top]$, *where* $\mathbf{x}_i \in [-B, B]^d, w_i \in [-B, B]$ *and* $\mathbf{z}_i \in \mathbb{R}^{d_z}$. *Given any* $\epsilon_{\mathrm{mult}} > 0$, *there exists a (FFN-only) transformer architecture such that*

$$\mathbf{f}_{\mathrm{mult}} = \mathrm{FFN}_L \circ \mathrm{FFN}_{L-1} \circ \cdots \circ \mathrm{FFN}_1$$

*with* $L = \mathcal{O}(\log(B/\epsilon_{\mathrm{mult}}))$ *layers that approximately multiply each component* $\mathbf{x}_i$ *with the weight* $w_i$ *and put it into a buffer, keeping other dimensions the same. This can be formally written as*

$$\mathbf{f}_{\mathrm{mult}}(\mathbf{Y}) = \begin{bmatrix} \mathbf{x}_1, & \cdots, & \mathbf{x}_H \\ f_{\mathrm{mult}}(w_1, \mathbf{x}_1), & \cdots, & f_{\mathrm{mult}}(w_H, \mathbf{x}_H) \\ \mathbf{0}_{2d}, & \cdots, & \mathbf{0}_{2d} \\ w_1, & \cdots, & w_H \\ \mathbf{z}_1, & \cdots, & \mathbf{z}_H \end{bmatrix}, \textit{where } \|f_{\mathrm{mult}}(w_i, \mathbf{x}_i) - w_i\mathbf{x}_i\|_\infty \le \epsilon_{\mathrm{mult}}.$$

*The number of nonzero coefficients in each weight matrices or bias vectors is at most* $\mathcal{O}(d)$, *and the norm of the matrices and bias are all bounded by* $\mathcal{O}(Bd)$.

### C.3 First Major GD Step

In this section, we construct the transformer architecture unrolling the first step of major GD procedure, and the result can be summarized as:

**Lemma 7** (Construct first major GD step)**.** *There exists a raw transformer architecture* $f_{\mathrm{GD},1} \in \mathcal{T}_{\mathrm{raw}}(D, L, M, B)$, *which can construct an approximate first step major GD result* $\widetilde{\mathbf{s}}^{(1)}$ *from the output of the encoder.*

*Given an error level* $\epsilon > 0$ *and learning rate* $\eta_t > 0$, *the approximated first step GD result satisfies*
$$\widetilde{\mathbf{s}}^{(1)} - \mathbf{s}^{(1)} = \xi^{(1)}, \textit{where } \|\xi^{(1)}\|_2 \le \epsilon,$$
*where* $\mathbf{s}^{(1)}$ *is the groundtruth gradient step.*

*The configuration of the raw transformer architecture satisfies*
$$D = 12d + d_e + d_t + 3, L = \mathcal{O}\left(\kappa(\mathbf{\Lambda})\kappa(\mathbf{\Gamma}_{\mathrm{obs}})\log^2\left(\frac{Hd\kappa(\mathbf{\Lambda})\kappa(\mathbf{\Sigma}_{\mathrm{obs}})}{\epsilon}\right)\right),$$
$$M = 4H, B = \mathcal{O}\left(\sqrt{H d^3}(r^2 + \kappa(\mathbf{\Lambda})\kappa(\mathbf{\Gamma}_{\mathrm{obs}})\sigma_t^{-1})\right).$$

We defer the analysis of transformer configuration to C.5.

Following the encoder network construction above, we obtain the following input:
$$\mathbf{Y} = \begin{bmatrix} \mathbf{x}_i & \cdots & \mathbf{x}_j \\ \mathbf{e}_i & \cdots & \mathbf{e}_j \\ \phi(t) & \cdots & \phi(t) \\ \mathbf{0}_{6d} & \cdots & \mathbf{0}_{6d} \\ 1 & \cdots & 1 \\ 1 & \cdots & 0 \\ 0 & \cdots & 1 \\ \mathbf{x}_i & \cdots & \mathbf{0}_d \\ \mathbf{0}_{4d} & \cdots & \mathbf{0}_{4d} \end{bmatrix}.$$

Revisiting the major GD gradient step in (8), our goal is to form the following at major GD first step:
$$\mathbf{s}_j^{(1)} = \mathbf{s}_j^{(0)} + \eta_t\alpha_t\boldsymbol{\mu}_{\mathrm{cond}} - \eta_t\mathbf{x}_j \approx \mathbf{0}^d + \underbrace{(\mathbf{\Sigma}_{\mathrm{cor}}^\top\widehat{\mathbf{\Sigma}}_{\mathrm{obs}}^{-1}(f_{\mathrm{mult}}(\eta_t\alpha_t, \mathbf{x}_{\mathrm{obs}})))_j}_{\eta_t\alpha_t\boldsymbol{\mu}_{j,\mathrm{cond}}} - f_{\mathrm{mult}}(\eta_t, \mathbf{x}_j). \quad (15)$$

Initially, we apply a multiplication module to construct:
$$\mathbf{f}_{\mathrm{mult}}(\mathbf{Y}) = \begin{bmatrix} f_{\mathrm{mult}}(\eta_t, \mathbf{x}_i) & \cdots & f_{\mathrm{mult}}(\eta_t, \mathbf{x}_j) \\ \mathbf{e}_i & \cdots & \mathbf{e}_j \\ \phi(t) & \cdots & \phi(t) \\ f_{\mathrm{mult}}(\eta_t\alpha_t, \mathbf{x}_i - \boldsymbol{\mu}_{\mathrm{i,obs}}) & \cdots & \mathbf{0}_d \\ \mathbf{0}_{5d} & \cdots & \mathbf{0}_{5d} \\ 1 & \cdots & 1 \\ 1 & \cdots & 0 \\ 0 & \cdots & 1 \\ \mathbf{x}_i & \cdots & \mathbf{0}_d \\ \mathbf{0}_{3d} & \cdots & \mathbf{0}_{4d} \end{bmatrix}.$$

This encapsulates the fundamental operations required for constructing the first major GD iteration.

### C.3.1 Auxiliary GD for First Major GD Step

In this section, we approximate the term $\boldsymbol{\Sigma}_{\mathrm{obs}}^{-1}(\eta_t\alpha_t\mathbf{x}_{\mathrm{obs}})$ using an iterative auxiliary GD procedure.

**First Auxiliary GD Step**  Starting from the initialization $\mathbf{u}_i^{(0)} = \mathbf{0}_d$ for $i \in \mathcal{I}_{\mathrm{obs}}$, we want the first auxiliary GD iteration finish the update:

$$\mathbf{u}_i^{(1)} = \mathbf{u}_i^{(0)} + \theta\left(\eta_t\alpha_t\mathbf{x}_{\mathrm{obs}}\right).$$

Using $\mathbf{f}_{\mathrm{mult}}$, we can easily obtain

$$\mathbf{f}_{\mathrm{mult}} \circ \mathbf{f}_{\mathrm{mult}}(\mathbf{Y}) = \begin{bmatrix} f_{\mathrm{mult}}(\eta_t, \mathbf{x}_i) & \cdots & f_{\mathrm{mult}}(\eta_t, \mathbf{x}_j) \\ \mathbf{e}_i & \cdots & \mathbf{e}_j \\ \phi(t) & \cdots & \phi(t) \\ f_{\mathrm{mult}}(\theta, f_{\mathrm{mult}}(\eta_t\alpha_t, \mathbf{x}_i))(= \mathbf{u}_i^{(1)}) & \cdots & \mathbf{0}_d \\ \mathbf{0}_{5d} & \cdots & \mathbf{0}_{5d} \\ 1 & \cdots & 1 \\ 1 & \cdots & 0 \\ 0 & \cdots & 1 \\ \mathbf{x}_i & \cdots & \mathbf{0}_d \\ \mathbf{0}_{4d} & \cdots & \mathbf{0}_{4d} \end{bmatrix}.$$

This finished the first step of auxiliary GD.

**Auxiliary GD Later Steps**  For subsequent iterations, the updated rule for the auxiliary GD becomes:

$$\mathbf{u}_i^{(k_{\mathrm{aux}}+1)} = \mathbf{u}_i^{(k_{\mathrm{aux}})} + \theta(\eta_t\alpha_t\mathbf{x}_{\mathrm{obs}}) - \theta\sum_{k\in\mathcal{I}_{\mathrm{obs}}}\boldsymbol{\Gamma}_{i,k}\boldsymbol{\Lambda}\mathbf{u}_k,$$

$$= \mathbf{u}_i^{(k_{\mathrm{aux}})} + \theta(\eta_t\alpha_t\mathbf{x}_{\mathrm{obs}}) - \theta\sum_{m=0}^{H-1}\sum_{k\in\mathcal{I}_{\mathrm{obs}}}\gamma_m\mathbb{1}\{|i-k|=m\}\boldsymbol{\Lambda}\mathbf{u}_k, \qquad (16)$$

for $k_{\mathrm{aux}} = 1, 2, \cdots, K_{\mathrm{aux}} - 1$.

Similar to the first auxiliary GD step performed above, we firstly use $\mathbf{f}_{\mathrm{mult}}$ to obtain

$$\mathbf{f}_{\mathrm{mult}} \circ \mathbf{f}_{\mathrm{mult}} \circ \mathbf{f}_{\mathrm{mult}}(\mathbf{Y}) = \begin{bmatrix} f_{\mathrm{mult}}(\eta_t, \mathbf{x}_i) & \cdots & f_{\mathrm{mult}}(\eta_t, \mathbf{x}_j) \\ \mathbf{e}_i & \cdots & \mathbf{e}_j \\ \phi(t) & \cdots & \phi(t) \\ \mathbf{u}_i^{(k_{\mathrm{aux}})} & \cdots & \mathbf{0}_d \\ \mathbf{0}_d & \cdots & \mathbf{0}_d \\ f_{\mathrm{mult}}(\theta\eta_t\alpha_t, \mathbf{x}_i) & \cdots & \mathbf{0}_d \\ f_{\mathrm{mult}}(\theta, \mathbf{u}_i^{(k_{\mathrm{aux}})}) & \cdots & \mathbf{0}_d \\ \mathbf{0}_{2d} & \cdots & \mathbf{0}_{2d} \\ 1 & \cdots & 1 \\ 1 & \cdots & 0 \\ 0 & \cdots & 1 \\ \mathbf{x}_i & \cdots & \mathbf{0}_d \\ \mathbf{0}_{4d} & \cdots & \mathbf{0}_{4d} \end{bmatrix}.$$

Then, by constructing a $4H$-head attention block as described in [C.6.1](#), we obtain

$$\begin{bmatrix} f_{\mathrm{mult}}(\eta_t, \mathbf{x}_i) & \cdots & f_{\mathrm{mult}}(\eta_t, \mathbf{x}_j) \\ \mathbf{e}_i & \cdots & \mathbf{e}_j \\ \phi(t) & \cdots & \phi(t) \\ \mathbf{u}_i^{(k_{\mathrm{aux}})} & \cdots & \mathbf{0}_d \\ \sum_{m=0}^{H-1}\sum_{k\in\mathcal{I}_{\mathrm{obs}}}\gamma_m\mathbb{1}\{|i-k|=m\}\boldsymbol{\Lambda} f_{\mathrm{mult}}(\theta, \mathbf{u}_k^{(k_{\mathrm{aux}})}) & \cdots & \mathbf{0}_d \\ f_{\mathrm{mult}}(\theta\eta_t\alpha_t, \mathbf{x}_i) & \cdots & \mathbf{0}_d \\ f_{\mathrm{mult}}(\theta, \mathbf{u}_i^{(k_{\mathrm{aux}})}) & \cdots & \mathbf{0}_d \\ \mathbf{0}_{2d} & \cdots & \mathbf{0}_{2d} \\ 1 & \cdots & 1 \\ 1 & \cdots & 0 \\ 0 & \cdots & 1 \\ \mathbf{x}_i & \cdots & \mathbf{0}_d \\ \mathbf{0}_{4d} & \cdots & \mathbf{0}_{4d} \end{bmatrix}.$$

Lastly, after combining an linear transformation FFN block with the attention block above to build up the basic transformer block $\mathcal{TB}_{\mathrm{obs}}$, we will have

$$
\mathcal{TB}_{\mathrm{obs}} \circ \mathbf{f}_{\mathrm{mult}} \circ \mathbf{f}_{\mathrm{mult}} \circ \mathbf{f}_{\mathrm{mult}}(\mathbf{Y}) =
\begin{bmatrix}
f_{\mathrm{mult}}(\eta_t, \mathbf{x}_i) & \cdots & f_{\mathrm{mult}}(\eta_t, \mathbf{x}_j) \\
\mathbf{e}_i & \cdots & \mathbf{e}_j \\
\phi(t) & \cdots & \phi(t) \\
\mathbf{u}_i^{(k_{\mathrm{aux}}+1)} & \cdots & \mathbf{0}_d \\
\mathbf{0}_{5d} & \cdots & \mathbf{0}_{5d} \\
1 & \cdots & 1 \\
1 & \cdots & 0 \\
0 & \cdots & 1 \\
\mathbf{x}_i & \cdots & \mathbf{0}_d \\
\mathbf{0}_{4d} & \cdots & \mathbf{0}_{4d}
\end{bmatrix},
$$

where

$$
\widetilde{\mathbf{u}}_i^{(k_{\mathrm{aux}}+1)} = \mathbf{u}_i^{(k_{\mathrm{aux}})} - \sum_{m=0}^{H-1} \sum_{k \in \mathcal{I}_{\mathrm{obs}}} \gamma_m \mathbb{1}\{|i-k| = m\} \mathbf{\Lambda} f_{\mathrm{mult}}(\theta, \mathbf{u}_k^{(k_{\mathrm{aux}})}) + f_{\mathrm{mult}}(\theta \eta_t \alpha_t, \mathbf{x}_i - \boldsymbol{\mu}_{\mathrm{i,obs}}).
$$

This completes a later step auxiliary GD update.

**Final result for Auxiliary GD**  We denote the iterative blocks $(\mathcal{TB}_{\mathrm{obs}} \circ \mathbf{f}_{\mathrm{mult}} \circ \mathbf{f}_{\mathrm{mult}})^{K_{\mathrm{aux}}}$ as $f_{\mathrm{inner}}$. The result of the auxiliary GD for approximating $(\mathbf{\Sigma}_{\mathrm{obs}}^{-1}(\eta_t \alpha_t(\mathbf{x}_{\mathrm{obs}} - \boldsymbol{\mu}_{\mathrm{obs}})))_i$ is expressed as:

$$
\widetilde{\mathbf{u}}_i = \left( \widehat{\mathbf{\Sigma}}_{\mathrm{obs}}^{-1} f_{\mathrm{mult}}(\eta_t \alpha_t, (\mathbf{x}_{\mathrm{obs}})) \right)_i.
$$

After completing $K_{\mathrm{aux}}$ auxiliary GD iterations, we obtain at the following transformation:

$$
f_{\mathrm{inner}} \circ \mathbf{f}_{\mathrm{mult}}(\mathbf{Y}) =
\begin{bmatrix}
f_{\mathrm{mult}}(\eta_t, \mathbf{x}_i) & \cdots & f_{\mathrm{mult}}(\eta_t, \mathbf{x}_j) \\
\mathbf{e}_i & \cdots & \mathbf{e}_j \\
\phi(t) & \cdots & \phi(t) \\
\widetilde{\mathbf{u}}_i & \cdots & \mathbf{0}_d \\
\mathbf{0}_{5d} & \cdots & \mathbf{0}_{5d} \\
1 & \cdots & 1 \\
1 & \cdots & 0 \\
0 & \cdots & 1 \\
\mathbf{x}_i & \cdots & \mathbf{0}_d \\
\mathbf{0}_{4d} & \cdots & \mathbf{0}_{4d}
\end{bmatrix},
$$

which incorporates the iterative updates. The resulting output includes $\widetilde{\mathbf{u}}_i$ for each observation entry, alongside the original data. We therefore finish the auxiliary GD procedure for the first major GD step.

### C.3.2  Matrix Multiplication

After $K_{\mathrm{aux}}$ steps of auxiliary GD iterations, we proceed with an additional matrix multiplication step to compute $\mathbf{\Sigma}_{\mathrm{cor}}^{\top} \widetilde{\mathbf{u}}$.

The multiplication can be expressed as:

$$
(\mathbf{\Sigma}_{\mathrm{cor}}^{\top} \widetilde{\mathbf{u}})_j = \sum_{m=0}^{H-1} \sum_{i \in \mathcal{I}_{\mathrm{obs}}} \gamma_m \mathbb{1}\{|i-j| = m\} \mathbf{\Lambda} \widetilde{\mathbf{u}}_i.
$$

Referring to the construction in C.6.2, this computation can be implemented using a $4H$-head attention block:

$$
\begin{bmatrix}
f_{\mathrm{mult}}(\eta_t, \mathbf{x}_i) & \cdots & f_{\mathrm{mult}}(\eta_t, \mathbf{x}_j) \\
\mathbf{e}_i & \cdots & \mathbf{e}_j \\
\phi(t) & \cdots & \phi(t) \\
\widetilde{\mathbf{u}}_i & \cdots & \mathbf{0}_d \\
\mathbf{0}_d & \cdots & \sum_{m=0}^{H-1} \sum_{i \in \mathcal{I}_{\mathrm{obs}}} \gamma_m \mathbb{1}\{|i-j| = m\} \mathbf{\Lambda} \widetilde{\mathbf{u}}_i (\approx \eta_t \alpha_t(\widehat{\boldsymbol{\mu}}_{\mathrm{j,cond}} \mathbf{x}_{\mathrm{obs}})) \\
\mathbf{0}_{4d} & \cdots & \mathbf{0}_{4d} \\
1 & \cdots & 1 \\
1 & \cdots & 0 \\
0 & \cdots & 1 \\
\mathbf{x}_i & \cdots & \mathbf{0}_d \\
\mathbf{0}_{4d} & \cdots & \mathbf{0}_{4d}
\end{bmatrix}.
$$

Combining the attention block described above with a linear transformation through a FFN block, which we denote as $\mathcal{TB}_{\mathrm{cort}}$, we obtain:

$$
f_{\mathrm{GD},1} = \mathcal{TB}_{\mathrm{cort}} \circ f_{\mathrm{inner}} \circ \mathbf{f}_{\mathrm{mult}}(\mathbf{Y}) =
\begin{bmatrix}
f_{\mathrm{mult}}(\eta_t, \mathbf{x}_i) & \cdots & f_{\mathrm{mult}}(\eta_t, \mathbf{x}_j) \\
\mathbf{e}_i & \cdots & \mathbf{e}_j \\
\phi(t) & \cdots & \phi(t) \\
\widetilde{\mathbf{u}}_i & \cdots & \mathbf{0}_d \\
\mathbf{0}_d & \cdots & \widetilde{\mathbf{s}}_j^{(1)} \\
\mathbf{0}_{4d} & \cdots & \mathbf{0}_{4d} \\
1 & \cdots & 1 \\
1 & \cdots & 0 \\
0 & \cdots & 1 \\
\mathbf{x}_i & \cdots & \mathbf{0}_d \\
\mathbf{0}_d & \cdots & (\mathbf{\Sigma}_{\mathrm{cor}}^\top(\widehat{\mathbf{\Sigma}}_{\mathrm{obs}}^{-1} f_{\mathrm{mult}}(\eta_t \alpha_t, \mathbf{x}_{\mathrm{obs}})))_j \\
\mathbf{0}_{3d} & \cdots & \mathbf{0}_{3d}
\end{bmatrix}.
$$

We now define $\widetilde{\mathbf{s}}_j^{(1)}$ as:

$$
\widetilde{\mathbf{s}}_j^{(1)} = f_{\mathrm{mult}}(\eta_t \alpha_t, \boldsymbol{\mu}_{\mathrm{j,miss}}) + \left( \mathbf{\Sigma}_{\mathrm{cor}}^\top \left( \widehat{\mathbf{\Sigma}}_{\mathrm{obs}}^{-1} f_{\mathrm{mult}}(\eta_t \alpha_t, \mathbf{x}_{\mathrm{obs}}) \right) \right)_j - f_{\mathrm{mult}}(\eta_t, \mathbf{x}_j).
$$

Comparing this result with (15), we observe that the first step of the major gradient descent is now complete. We represent this step as $f_{\mathrm{GD},1}$.

For simplicity, we introduce the notation:

$$
\eta_t \widehat{\alpha_t \boldsymbol{\mu}_{j,\mathrm{cond}}} = \left( \mathbf{\Sigma}_{\mathrm{cor}}^\top \left( \widehat{\mathbf{\Sigma}}_{\mathrm{obs}}^{-1} f_{\mathrm{mult}}(\eta_t \alpha_t, \mathbf{x}_{\mathrm{obs}}) \right) \right)_j.
$$

### C.4 Major GD Later Steps

In this section, we construct the transformer architecture unrolling the later steps of major GD procedure, and the result can be summarized as

**Lemma 8** (Construct major GD later steps). *There exists a raw transformer architecture $f_{\mathrm{GD}} \in \mathcal{T}_{\mathrm{raw}}(D, L, M, B)$ (i.e. without encoder and decoder), which can construct a new approximate later step GD result $\widetilde{\mathbf{s}}^+$ from the output of the latest step of major GD. Given an error level $\epsilon \in (0, 1)$ and learning rate $\eta_t > 0$, the approximated first step GD result satisfies*

$$
\widetilde{\mathbf{s}}^{(k+1)} - \mathbf{s}^{(k+1)} = \xi^+, \text{where } \|\xi^+\|_2 \le \epsilon,
$$

*where $\mathbf{s}^{(k+1)}$ is the groundtruth gradient step.*

*Furthermore, the configuration of the raw transformer architecture satisfies*

$$
D = 12d + d_e + d_t + 3, L = \mathcal{O}\left( \kappa_t \kappa(\mathbf{\Lambda}) \kappa(\mathbf{\Gamma}_{\mathrm{obs}}) \log^2 \left( \frac{Hd\kappa(\mathbf{\Lambda})\kappa(\mathbf{\Gamma}_{\mathrm{obs}})}{\sigma_t \epsilon} \right) \right),
$$

$$
M = 4H, B = \mathcal{O}\left( \sqrt{Hd^3}(r^2 + \kappa(\mathbf{\Lambda})\kappa(\mathbf{\Gamma}_{\mathrm{obs}})\sigma_t^{-1}) \right).
$$

We defer the analysis of transformer configuration to C.5.

In the later steps of the major GD, we need to compute the following update:

$$
\begin{aligned}
\mathbf{s}^{(k+1)} &= \mathbf{s}^{(k)} - \eta \left[ -\alpha_t^2 \mathbf{\Sigma}_{\mathrm{cor}}^\top \mathbf{\Sigma}_{\mathrm{obs}}^{-1} \mathbf{\Sigma}_{\mathrm{cor}} \mathbf{s}^{(k)} - \alpha_t \boldsymbol{\mu}_{\mathrm{cond}}(\mathbf{x}_{\mathrm{obs}}) + \alpha_t^2 \mathbf{\Sigma}_{\mathrm{miss}} \mathbf{s}^{(k)} + \sigma_t^2 \mathbf{s}^{(k)} + \mathbf{v}_t \right] \\
&\approx \mathbf{s}^{(k)} + \mathbf{\Sigma}_{\mathrm{cor}}^\top \widehat{\mathbf{\Sigma}}_{\mathrm{obs}}^{-1} \mathbf{\Sigma}_{\mathrm{cor}} f_{\mathrm{mult}}(\eta_t \alpha_t^2, \mathbf{s}^{(k)}) + \eta_t \alpha_t \widehat{\boldsymbol{\mu}_{\mathrm{cond}}}(\mathbf{x}_{\mathrm{obs}}) + \mathbf{\Sigma}_{\mathrm{miss}} f_{\mathrm{mult}}(\eta_t \alpha_t^2, \mathbf{s}^{(k)}) \\
&\quad - f_{\mathrm{mult}}(\eta_t \sigma_t^2, \mathbf{s}^{(k)}) - f_{\mathrm{mult}}(\eta_t, \mathbf{v}_t).
\end{aligned}
\tag{17}
$$

In the following proof, for the sake of simplicity, we use $\mathbf{s}^+, \mathbf{s}$ as abbreviation for $\mathbf{s}^{(k+1)}, \mathbf{s}^{(k)}$, respectively.

In each new major GD step, the input to the iteration is the output of the most recent GD step. For simplicity, we continue to represent this input using $\mathbf{Y}$.

Similar to the construction in the first step, we first apply a multiplication module to obtain:

$$
\mathbf{f}_{\mathrm{mult}}(\mathbf{Y}) =
\begin{bmatrix}
f_{\mathrm{mult}}(\eta_t, \mathbf{x}_i) & \cdots & f_{\mathrm{mult}}(\eta_t, \mathbf{x}_j) \\
\mathbf{e}_i & \cdots & \mathbf{e}_j \\
\phi(t) & \cdots & \phi(t) \\
\mathbf{0}_d & \cdots & \mathbf{s}_j \\
\mathbf{0}_{3d} & \cdots & \mathbf{0}_{3d} \\
\mathbf{0}_d & \cdots & f_{\mathrm{mult}}(\eta_t \alpha_t^2, \mathbf{s}_j) \\
\mathbf{0}_d & \cdots & f_{\mathrm{mult}}(\eta_t \sigma_t^2, \mathbf{s}_j) \\
1 & \cdots & 1 \\
1 & \cdots & 0 \\
0 & \cdots & 1 \\
\mathbf{x}_i & \cdots & \mathbf{0}_d \\
\mathbf{0}_d & \cdots & \eta_t \widehat{\alpha_t \boldsymbol{\mu}}_{j,cond} \\
\mathbf{0}_{3d} & \cdots & \mathbf{0}_{3d}
\end{bmatrix} .
$$

Next, we proceed to the matrix multiplication step, followed by the auxiliary GD process.

### C.4.1  Matrix Multiplication 1

To begin, we compute $\boldsymbol{\Sigma}_{\mathrm{cor}}\mathbf{s}$, which can be expressed as:

$$
(\boldsymbol{\Gamma}_{\mathrm{cor}} \otimes \boldsymbol{\Lambda})\mathbf{s}.
$$

For each $i$ corresponding to the observations, this can be further rewritten as:

$$
\sum_{m=0}^{H-1} \sum_{j \in \mathcal{I}_{\mathrm{miss}}} \gamma_m \mathbb{1}\{|i - j| = m\} \boldsymbol{\Lambda} \mathbf{s}_j.
$$

To perform this computation, similar to the construction in the first step major GD, we employ a $4H$-head attention block combined with an identical FFN block to form $\mathcal{TB}_{\mathrm{cor}}$:

$$
\mathcal{TB}_{\mathrm{cor}}(\mathbf{Y})
$$

$$
=
\begin{bmatrix}
f_{\mathrm{mult}}(\eta_t, \mathbf{x}_i) & \cdots & f_{\mathrm{mult}}(\eta_t, \mathbf{x}_j) \\
\mathbf{e}_i & \cdots & \mathbf{e}_j \\
\phi(t) & \cdots & \phi(t) \\
\mathbf{0}_d & \cdots & \mathbf{s}_j \\
\sum_{m=0}^{H-1} \sum_{j \in \mathcal{I}_{\mathrm{miss}}} \gamma_m \mathbb{1}\{|i - j| = m\} \boldsymbol{\Lambda} f_{\mathrm{mult}}(\eta_t \sigma_t^2, \mathbf{s}_j) & \cdots & \mathbf{0}_d \\
\mathbf{0}_{2d} & \cdots & \mathbf{0}_{2d} \\
\mathbf{0}_d & \cdots & f_{\mathrm{mult}}(\eta_t \alpha_t^2, \mathbf{s}_j) \\
\mathbf{0}_d & \cdots & f_{\mathrm{mult}}(\eta_t \sigma_t^2, \mathbf{s}_j) \\
1 & \cdots & 1 \\
1 & \cdots & 0 \\
0 & \cdots & 1 \\
\mathbf{x}_i & \cdots & \mathbf{0}_d \\
\mathbf{0}_d & \cdots & \eta_t \widehat{\alpha_t \boldsymbol{\mu}}_{j,cond} \\
\mathbf{0}_{3d} & \cdots & \mathbf{0}_{3d}
\end{bmatrix} .
$$

### C.4.2  Auxiliary GD

In this step, we compute $\boldsymbol{\Sigma}_{\mathrm{obs}}^{-1}\boldsymbol{\Sigma}_{\mathrm{cor}}\mathbf{s}$.

Following the auxiliary GD procedure described in C.3.1, we employ a similar iterative approach using $f_{\mathrm{inner}}$ to approximate the multiplication between a matrix inverse and vectors:

$$
f_{\text{inner}} \circ \mathcal{TB}_{\text{cor}}(\mathbf{Y}) =
\begin{bmatrix}
f_{\text{mult}}(\eta_t, \mathbf{x}_i) & \cdots & f_{\text{mult}}(\eta_t, \mathbf{x}_j) \\
\mathbf{e}_i & \cdots & \mathbf{e}_j \\
\phi(t) & \cdots & \phi(t) \\
\mathbf{0}_d & \cdots & \mathbf{s}_j \\
(\widehat{\boldsymbol{\Sigma}}_{\text{obs}}^{-1} \boldsymbol{\Sigma}_{\text{cor}} f_{\text{mult}}(\eta_t \alpha_t^2, \mathbf{s}))_i & \cdots & \mathbf{0}_d \\
\mathbf{0}_{4d} & \cdots & \mathbf{0}_{4d} \\
\mathbf{0}_d & \cdots & f_{\text{mult}}(\eta_t \alpha_t^2, \mathbf{s}_j) \\
\mathbf{0}_d & \cdots & f_{\text{mult}}(\eta_t \sigma_t^2, \mathbf{s}_j) \\
1 & \cdots & 1 \\
1 & \cdots & 0 \\
0 & \cdots & 1 \\
\mathbf{x}_i & \cdots & \mathbf{0}_d \\
\mathbf{0}_d & \cdots & \eta_t \widehat{\alpha_t \boldsymbol{\mu}_{j,cond}} \\
\mathbf{0}_{3d} & \cdots & \mathbf{0}_{3d}
\end{bmatrix}.
$$

### C.4.3 Matrix Multiplication 2

In this step, we compute $\boldsymbol{\Sigma}_{\text{cor}}^{\top} \boldsymbol{\Sigma}_{\text{obs}}^{-1} \boldsymbol{\Sigma}_{\text{cor}} \mathbf{s}$.

Following a similar procedure as in C.3.2, we construct a transformer block $\mathcal{TB}_{\text{cort}}$. This block similarly employs a $4H$-head attention mechanism combined with an identity FFN to perform the matrix multiplication:

$$
\begin{aligned}
&\mathcal{TB}_{\text{cort}} \circ f_{\text{inner}} \circ \mathcal{TB}_{\text{cor}}(\mathbf{Y}) \\
&=
\begin{bmatrix}
f_{\text{mult}}(\eta_t, \mathbf{x}_i) & \cdots & f_{\text{mult}}(\eta_t, \mathbf{x}_j) \\
\mathbf{e}_i & \cdots & \mathbf{e}_j \\
\phi(t) & \cdots & \phi(t) \\
\mathbf{0}_d & \cdots & \mathbf{s}_j \\
(\widehat{\boldsymbol{\Sigma}}_{\text{obs}}^{-1} \boldsymbol{\Sigma}_{\text{cor}} f_{\text{mult}}(\eta_t \alpha_t^2, \mathbf{s}))_i & \cdots & \mathbf{0}_d \\
\mathbf{0}_d & \cdots & (\boldsymbol{\Sigma}_{\text{cor}}^{\top} \widehat{\boldsymbol{\Sigma}}_{\text{obs}}^{-1} \boldsymbol{\Sigma}_{\text{cor}} f_{\text{mult}}(\eta_t \alpha_t^2, \mathbf{s}))_j \\
\mathbf{0}_d & \cdots & \mathbf{0}_d \\
\mathbf{0}_d & \cdots & f_{\text{mult}}(\eta_t \alpha_t^2, \mathbf{s}_j) \\
\mathbf{0}_d & \cdots & f_{\text{mult}}(\eta_t \sigma_t^2, \mathbf{s}_j) \\
1 & \cdots & 1 \\
1 & \cdots & 0 \\
0 & \cdots & 1 \\
\mathbf{x}_i & \cdots & \mathbf{0}_d \\
\mathbf{0}_d & \cdots & \eta_t \widehat{\alpha_t \boldsymbol{\mu}_{j,cond}} \\
\mathbf{0}_{3d} & \cdots & \mathbf{0}_{3d}
\end{bmatrix}.
\end{aligned}
$$

### C.4.4 Matrix Multiplication 3

In this step, we compute $\boldsymbol{\Sigma}_{\text{miss}} \mathbf{s}$.

Following a similar procedure as in C.6.1, we construct a transformer block $\mathcal{TB}_{\text{miss}}$. This attention layer utilizes a $4H$-head attention block to obtain:

$$\mathcal{TB}_{\text{miss}} \circ \mathcal{TB}_{\text{cort}} \circ f_{\text{inner}} \circ \mathcal{TB}_{\text{cor}}(\mathbf{Y})$$

$$= \begin{bmatrix}
f_{\text{mult}}(\eta_t, \mathbf{x}_i) & \cdots & f_{\text{mult}}(\eta_t, \mathbf{x}_j) \\
\mathbf{e}_i & \cdots & \mathbf{e}_j \\
\phi(t) & \cdots & \phi(t) \\
\mathbf{0}_d & \cdots & \mathbf{s}_j \\
(\widehat{\boldsymbol{\Sigma}}_{\text{obs}}^{-1}\boldsymbol{\Sigma}_{\text{cor}}f_{\text{mult}}(\eta_t\alpha_t^2,\mathbf{s}))_i & \cdots & \mathbf{0}_d \\
\mathbf{0}_d & \cdots & (\boldsymbol{\Sigma}_{\text{cor}}^\top\widehat{\boldsymbol{\Sigma}}_{\text{obs}}^{-1}\boldsymbol{\Sigma}_{\text{cor}}f_{\text{mult}}(\eta_t\alpha_t^2,\mathbf{s}))_j \\
\mathbf{0}_d & \cdots & (\boldsymbol{\Sigma}_{\text{miss}}f_{\text{mult}}(\eta_t\sigma_t^2,\mathbf{s}))_j \\
\mathbf{0}_d & \cdots & f_{\text{mult}}(\eta_t\alpha_t^2,\mathbf{s}_j) \\
\mathbf{0}_d & \cdots & f_{\text{mult}}(\eta_t\sigma_t^2,\mathbf{s}_j) \\
1 & \cdots & 1 \\
1 & \cdots & 0 \\
0 & \cdots & 1 \\
\mathbf{x}_i & \cdots & \mathbf{0}_d \\
\mathbf{0}_d & \cdots & \eta_t\widehat{\alpha_t\boldsymbol{\mu}_{j,cond}} \\
\mathbf{0}_{3d} & \cdots & \mathbf{0}_{3d}
\end{bmatrix}.$$

Then, with a linear transformation FFN layer, we get the final output

$$\mathcal{TB}_{\text{miss}} \circ \mathcal{TB}_{\text{cort}} \circ f_{\text{inner}} \circ \mathcal{TB}_{\text{cor}}(\mathbf{Y}) = \begin{bmatrix}
f_{\text{mult}}(\eta_t, \mathbf{x}_i) & \cdots & f_{\text{mult}}(\eta_t, \mathbf{x}_j) \\
\mathbf{e}_i & \cdots & \mathbf{e}_j \\
\phi(t) & \cdots & \phi(t) \\
\mathbf{0}_d & \cdots & \widetilde{\mathbf{s}}_j^+ \\
\mathbf{0}_{5d} & \cdots & \mathbf{0}_{5d} \\
1 & \cdots & 1 \\
1 & \cdots & 0 \\
0 & \cdots & 1 \\
\mathbf{x}_i & \cdots & \mathbf{0}_d \\
\mathbf{0}_d & \cdots & \eta_t\widehat{\alpha_t\boldsymbol{\mu}_{j,cond}} \\
\mathbf{0}_{3d} & \cdots & \mathbf{0}_{3d}
\end{bmatrix}.$$

where

$$\widetilde{\mathbf{s}}_j^+ = \mathbf{s}_j + \left(\boldsymbol{\Sigma}_{\text{cor}}^\top\widehat{\boldsymbol{\Sigma}}_{\text{obs}}^{-1}f_{\text{mult}}(\eta_t\alpha_t^2,\mathbf{s})\right)_j + \eta_t\alpha_t\widehat{\boldsymbol{\mu}_{j,\text{cond}}}(\mathbf{x}_{\text{obs}})$$
$$+ \left(\boldsymbol{\Sigma}_{\text{miss}}f_{\text{mult}}(\eta_t\alpha_t^2,\mathbf{s})\right)_j - f_{\text{mult}}(\eta_t\sigma_t^2,\mathbf{s}_j) - f_{\text{mult}}(\eta_t,\mathbf{x}_j).$$

Comparing this expression with (17), we conclude that one major GD update has been completed.

We can represent the later major GD steps compactly as:
$$f_{\text{GD}} = \mathcal{TB}_{\text{miss}} \circ \mathcal{TB}_{\text{cort}} \circ f_{\text{inner}} \circ \mathcal{TB}_{\text{cor}}.$$

## C.5 Error Analysis and Transformer Configurations

In this section, we analyze the error induced by using transformer architectures to unroll the gradient descent procedure, and derive the corresponding transformer configurations to formally establish the result in Lemma 7 and 8. Lastly, we combine these results to finish the proof of Theorem 1.

Above all, we should notice that, by our construction above, leveraging transformer to approximate each step of Auxiliary GD also induces noise. Consequently, similar to (14), in each auxiliary GD step, we incorporate an error term and represent the update as:

$$\mathbf{u}^{(k_{\text{aux}}+1)} = \mathbf{u}^{(k_{\text{aux}})} - \theta\nabla\mathcal{L}_{\text{t,aux}}(\mathbf{u}^{(k_{\text{aux}})}) + \xi_0^{(k_{\text{aux}})}, \tag{18}$$

where $\xi_0^{(k_{\text{aux}})}$ represents the approximation error term in each auxiliary GD step. We can also state a corresponding Lemma that sharing the same proof strategy with its counterpart in major GD (Lemma 3):

**Lemma 9.** *For an arbitrarily fixed time $t \in (0, T]$ and given an error tolerance $\epsilon_0 \in (0, 1)$, if we can control $\|\xi_0^{(k_{\text{aux}})}\|_2 \le \epsilon_0$, then running the auxiliary GD in (14) with a suitable step size $\theta$ for*

$$K_{\text{aux}} = \left\lceil \frac{\kappa(\boldsymbol{\Sigma}_{\text{obs}})+1}{2}\log\left(\frac{\|\mathbf{b}\|_2}{\lambda_{\min}(\boldsymbol{\Gamma}_{\text{obs}})\lambda_{\min}(\boldsymbol{\Lambda})\epsilon_0}\right)\right\rceil$$

*iterations gives:*

$$\|\mathbf{u}^{(K_{\text{aux}})} - \mathbf{u}\|_2 \le \frac{\kappa(\boldsymbol{\Sigma}_{\text{obs}})+3}{2}\epsilon_0.$$

### C.5.1 First Major GD Step

In this part, we analyze the noise introduced by the construction in C.3, and corresponding transformer architecture configuration (i.e. $D, L, M, B$).

**Bounding Approximation Error** Let $\mathbf{s}^{(1)}$ denote the exact major GD update, and $\widetilde{\mathbf{s}}^{(1)}$ represent our approximation. According to (15), in first major GD step, transformer blocks are utilized to compute:

$$\widetilde{\mathbf{s}}_j^{(1)} = \mathbf{\Sigma}_{\mathrm{cor}}^\top (\widehat{\mathbf{\Sigma}}_{\mathrm{obs}}^{-1} f_{\mathrm{mult}}(\eta_t \alpha_t, \mathbf{x}_{\mathrm{obs}}))_j - f_{\mathrm{mult}}(\eta_t, \mathbf{x}_j),$$

for $j \in |\mathcal{I}_{\mathrm{miss}}|$, as an approximation to:

$$\mathbf{s}^{(1)} = \eta_t \left( \alpha_t \left( \boldsymbol{\mu}_{\mathrm{miss}} + \mathbf{\Sigma}_{\mathrm{cor}}^\top \mathbf{\Sigma}_{\mathrm{obs}}^{-1} \mathbf{x}_{\mathrm{obs}} - \mathbf{v}_t \right) \right).$$

In first major GD step, auxiliary GD is responsible for computing $\widehat{\mathbf{\Sigma}}_{\mathrm{obs}}^{-1}(\eta_t \alpha_t (\mathbf{x}_{\mathrm{obs}} - \boldsymbol{\mu}_{\mathrm{obs}}))$.

From (16), each auxiliary GD step updates as (here we only analyze the later auxiliary GD step, and we use $\mathbf{u}^+, \mathbf{u}$ for $\widetilde{\mathbf{u}}^{(k_{\mathrm{aux}}+1)}, \widetilde{\mathbf{u}}^{(k_{\mathrm{aux}})}$ for the sake of simplicity):

$$\mathbf{u}_i^+ = \mathbf{u}_i + \theta(\eta_t \alpha_t \mathbf{x}_{\mathrm{obs}}) - \theta \sum_{m=0}^{H-1} \sum_{k \in \mathcal{I}_{\mathrm{obs}}} \gamma_m \mathbb{1}\{|i - k| = m\} \mathbf{\Lambda} \mathbf{u}_k,$$

and we approximate it using:

$$\widetilde{\mathbf{u}}_i^+ = \mathbf{u}_i - \sum_{m=0}^{H-1} \sum_{k \in \mathcal{I}_{\mathrm{obs}}} \gamma_m \mathbb{1}\{|i - k| = m\} \mathbf{\Lambda} f_{\mathrm{mult}}(\theta, \mathbf{u}_k) + f_{\mathrm{mult}}(\theta \eta_t \alpha_t, \mathbf{x}_i).$$

To ensure control over the error $\|\widetilde{\mathbf{u}}^+ - \mathbf{u}^+\|_2 = \|\xi_0\|_2 \leq \epsilon_0$, Lemma 6 indicates that setting $\epsilon_{\mathrm{mult,aux,1}} = \frac{\epsilon_0}{(H^{3/2}\|\mathbf{\Lambda}\|_{\mathrm{F}} + H^{1/2})\sqrt{d}}$ suffices. This requires $L_{\mathrm{mult,aux,1}} = \mathcal{O}\left(\log\left(\frac{dN\|\mathbf{\Lambda}\|_{\mathrm{F}}}{\epsilon_0}\right)\right)$ iterations.

With each step noise controlled, the entire auxiliary GD procedure, combined with the subsequent matrix product blocks, yields $\mathbf{\Sigma}_{\mathrm{cor}} \widehat{\mathbf{\Sigma}}_{\mathrm{obs}}^{-1} f_{\mathrm{mult}}(\eta_t \alpha_t, \mathbf{x}_{\mathrm{obs}} - \boldsymbol{\mu}_{\mathrm{obs}})$. According to Lemma 9, setting

$$K_{\mathrm{aux}} = \left\lceil \frac{\kappa(\mathbf{\Sigma}_{\mathrm{obs}}) + 1}{2} \log\left(\frac{\eta_t \alpha_t \|\mathbf{x}_{\mathrm{obs}}\|_2}{\lambda_{\min}(\mathbf{\Gamma}_{\mathrm{obs}})\lambda_{\min}(\mathbf{\Lambda})\epsilon_0}\right) \right\rceil,$$

ensures that:

$$\|\mathbf{\Sigma}_{\mathrm{cor}}^\top \widehat{\mathbf{\Sigma}}_{\mathrm{obs}}^{-1} f_{\mathrm{mult}}(\eta_t \alpha_t, \mathbf{x}_{\mathrm{obs}} - \boldsymbol{\mu}_{\mathrm{obs}}) - \mathbf{\Sigma}_{\mathrm{cor}}^\top \mathbf{\Sigma}_{\mathrm{obs}}^{-1} \eta_t \alpha_t \mathbf{x}_{\mathrm{obs}}\|_2 \leq \|\mathbf{\Sigma}_{\mathrm{cor}}\|_2 \frac{\kappa(\mathbf{\Sigma}_{\mathrm{obs}}) + 3}{2} \epsilon_0. \quad (19)$$

This provides an approximation of $\mathbf{\Sigma}_{\mathrm{cor}}^\top \mathbf{\Sigma}_{\mathrm{obs}}^{-1}$ with controlled error bounds.

Finally, the $\mathbf{f}_{\mathrm{mult}}$ module approximates scalar and vector multiplications to complete the first step of gradient descent. The overall error for each $j$ is computed as:

$$
\begin{aligned}
\|\widetilde{\mathbf{s}}_j^{(1)} - \mathbf{s}_j^{(1)}\|_2 &\leq \|f_{\mathrm{mult}}(\eta_t \alpha_t, \boldsymbol{\mu}_{\mathrm{j,miss}}) - \eta_t \alpha_t \boldsymbol{\mu}_{\mathrm{j,miss}}\|_2 \\
&\quad + \|\mathbf{\Sigma}_{\mathrm{cor}}^\top \widehat{\mathbf{\Sigma}}_{\mathrm{obs}}^{-1} f_{\mathrm{mult}}(\eta_t \alpha_t, \mathbf{x}_{\mathrm{obs}}) - \mathbf{\Sigma}_{\mathrm{cor}}^\top \mathbf{\Sigma}_{\mathrm{obs}}^{-1} \eta_t \alpha_t (\mathbf{x}_{\mathrm{obs}})\|_2 \\
&\quad + \|f_{\mathrm{mult}}(\eta_t, \mathbf{x}_j) - \eta_t \mathbf{x}_j\|_2 \\
&\leq \|\mathbf{\Sigma}_{\mathrm{cor}}\|_2 \frac{\kappa(\mathbf{\Sigma}_{\mathrm{obs}}) + 3}{2} \epsilon_0 + 2\sqrt{d} \epsilon_{\mathrm{mult}}.
\end{aligned}
$$

By setting $\epsilon_0 = \left(\|\mathbf{\Sigma}_{\mathrm{cor}}\|_{\mathrm{F}}(\kappa(\mathbf{\Sigma}_{\mathrm{obs}}) + 3)\sqrt{H}\right)^{-1} \epsilon$, which leads to an auxiliary gradient descent step count of

$$K_{\mathrm{aux,1}} = \left\lceil (\kappa(\mathbf{\Sigma}_{\mathrm{obs}}) + 1) \log\left(\frac{\left(\eta_t \alpha_t \|\mathbf{\Sigma}_{\mathrm{cor}}\|_2 (\kappa(\mathbf{\Sigma}_{\mathrm{obs}}) + 3)\sqrt{H}\right) \sqrt{H} d C_{\mathrm{data}}^\delta}{\lambda_{\min}(\mathbf{\Gamma}_{\mathrm{obs}})\lambda_{\min}(\mathbf{\Lambda})\epsilon}\right) \right\rceil,$$

and by Lemma 6, setting $\epsilon_{\mathrm{mult}} = \left(8\sqrt{dN}\right)^{-1} \epsilon$, which leads to

$$L_{\mathrm{mult,1}} = \mathcal{O}\left(\log\left(\frac{dN}{\epsilon}\right)\right),$$

we successfully control the error $\|\xi^{(1)}\|_2 = \|\widetilde{\mathbf{s}}^{(1)} - \mathbf{s}^{(1)}\|_2 \leq \frac{\epsilon}{2} < \epsilon$.

**Configuration of Transformer Architecture for Approximating the First Major GD Step** We finally summarize our construction by characterizing the configuration of the architecture:

- The input to the transformer is of dimension $D \times H$ with $D = 12d + d_e + d_t + 3$.

- In each auxiliary GD step, we use 1 transformer block to form the matrix product and some $\mathbf{f}_{\text{mult}}$ modules, requiring a total of $L_{\text{mult,aux}}$ transformer blocks. We need to perform $P_1$ auxiliary GD steps. After completing the auxiliary GD, additional $\mathbf{f}_{\text{mult}}$ modules are used to compute scalar and vector products, which require $L_{\text{mult,1}}$ blocks. Thus, the number of the transformer blocks is bounded by

$$L = K_{\text{aux,1}}(1 + L_{\text{mult,aux,1}}) + L_{\text{mult,1}}$$
$$= \mathcal{O}\left(\kappa(\mathbf{\Lambda})\kappa(\mathbf{\Gamma}_{\text{obs}}) \log^2\left(\frac{Hd\kappa(\mathbf{\Lambda})\kappa(\mathbf{\Sigma}_{\text{obs}}))}{\epsilon}\right)\right).$$

- The number of transformer blocks is bounded by $M = 4H$.

- With the constructions above, referring to Lemma 6, the norm of the multiplication module is bounded by $\mathcal{O}(d(\|\mathbf{x}\|_\infty + \|\mathbf{s}\|_\infty);$, and Lemma 4 helps us bound $\mathbf{x}_\infty$ and $\|v\|_\infty$. Referring to the attention module constructed in C.6.2 and C.6.1, the norm of the attention matrices are bounded by $\mathcal{O}(d(r^2 + \lambda_{\max}(\mathbf{\Lambda})))$; and considering the weight matrices in the FFN, since they only do linear transformations and only have at most $\mathcal{O}(d)$ nonzero weights, their norm are bounded by $\mathcal{O}(d)$. To sum up, we have the norm of the transformer parameters bounded by

$$\mathcal{O}\left(\sqrt{Hd^3}(r^2 + \kappa(\mathbf{\Lambda})\kappa(\mathbf{\Gamma}_{\text{obs}})\sigma_t^{-1})\right).$$

And this finishes the proof of Lemma 7.

### C.5.2 Major GD Later Steps

In this part, we analyze the noise introduced by the construction in C.4, and corresponding transformer architecture configuration.

**Bounding Approximation Error** In later steps, according to 17, we use

$$\widetilde{\mathbf{s}}^+ = \mathbf{s} + \mathbf{\Sigma}_{\text{cor}}^\top \widehat{\mathbf{\Sigma}}_{\text{obs}}^{-1} \mathbf{\Sigma}_{\text{cor}} f_{\text{mult}}(\eta_t \alpha_t^2, \mathbf{s}) + \eta_t \widehat{\alpha_t \boldsymbol{\mu}_{\text{cond}}}$$
$$+ \mathbf{\Sigma}_{\text{miss}} f_{\text{mult}}(\eta_t \alpha_t^2, \mathbf{s}) - f_{\text{mult}}(\eta_t \sigma_t^2, \mathbf{s}) - f_{\text{mult}}(\eta_t, \mathbf{v}_t)$$

to approximate

$$\mathbf{s}^+ = \mathbf{s} - \eta_t[-\alpha_t^2 \mathbf{\Sigma}_{\text{cor}}^\top \mathbf{\Sigma}_{\text{obs}}^{-1} \mathbf{\Sigma}_{\text{cor}} \mathbf{s} - \alpha_t \boldsymbol{\mu}_{\text{cond}} + \alpha_t^2 \mathbf{\Sigma}_{\text{miss}} \mathbf{s} + \sigma_t^2 \mathbf{s} + \mathbf{v}_t].$$

We first consider the auxiliary gradient descent which computes $\widehat{\mathbf{\Sigma}}_{\text{obs}}^{-1} \mathbf{\Sigma}_{\text{cor}} f_{\text{mult}}(\eta_t \alpha_t^2, \mathbf{s})$. Compared to the first step analysis, we simply replace $\mathbf{x}_{\text{obs}}$ with $\mathbf{s}$. So we can completely follow the procedure in (19). To control $\|\widetilde{\mathbf{u}}^+ - \mathbf{u}^+\|_2 = \|\xi\|_2 \leq \epsilon_0$, we set the corresponding inside multiplication module error as

$$\epsilon_{\text{mult,aux}} = \frac{\epsilon_0}{(H^{3/2}\|\mathbf{\Lambda}\|_F + H^{1/2})\sqrt{d}},$$

which requires

$$L_{\text{mult,aux,+}} = \mathcal{O}\left(\log\left(\frac{\|\mathbf{\Sigma}_{\text{cor}}\|_2\|\mathbf{s}\|_\infty dN\|\mathbf{\Lambda}\|_F}{\epsilon_0}\right)\right).$$

Combining the auxiliary GD output with the following matrix multiplication blocks, we obtain $\mathbf{\Sigma}_{\text{cor}}^\top \widehat{\mathbf{\Sigma}}_{\text{obs}}^{-1} \mathbf{\Sigma}_{\text{cor}} f_{\text{mult}}(\eta_t \alpha_t^2, \mathbf{s})$. By Lemma 9, with $K_{\text{aux}} = \lceil \frac{\kappa(\mathbf{\Sigma}_{\text{obs}})+1}{2} \log(\frac{\eta_t \alpha_t \|\mathbf{\Sigma}_{\text{cor}}\|_F\|s\|_2}{\lambda_{\min}(\mathbf{\Gamma}_{\text{obs}})\lambda_{\min}(\mathbf{\Lambda})\epsilon_0})\rceil$, we have

$$\|\mathbf{\Sigma}_{\text{cor}}^\top \widehat{\mathbf{\Sigma}}_{\text{obs}}^{-1} \mathbf{\Sigma}_{\text{cor}} f_{\text{mult}}(\eta_t \alpha_t^2, \mathbf{s}) - \eta_t \alpha_t^2 \mathbf{\Sigma}_{\text{cor}}^\top \mathbf{\Sigma}_{\text{obs}}^{-1} \mathbf{\Sigma}_{\text{cor}} s\|_2 \leq \|\mathbf{\Sigma}_{\text{cor}}\|_F \frac{\kappa(\mathbf{\Sigma}_{\text{obs}}) + 3}{2}\epsilon_0.$$

Next, we decompose the overall error term. Recall that

$$\eta_t \widehat{\alpha_t \boldsymbol{\mu}_{j,\text{cond}}} = f_{\text{mult}}(\eta_t \alpha_t, \boldsymbol{\mu}_{j,\text{miss}}) + \left(\mathbf{\Sigma}_{\text{cor}}^\top \left(\widehat{\mathbf{\Sigma}}_{\text{obs}}^{-1} f_{\text{mult}}(\eta_t \alpha_t, (\mathbf{x}_{\text{obs}}))\right)\right)_j,$$

similar to the analysis in the first iteration, we can derive the error bound for approximating $\mathbf{s}_j^+$ as:

$$\|\widetilde{\mathbf{s}}_j^+ - \mathbf{s}_j^+\|_2 \leq \|\boldsymbol{\Sigma}_{\mathrm{cor}}^\top \widehat{\boldsymbol{\Sigma}}_{\mathrm{obs}}^{-1} \boldsymbol{\Sigma}_{\mathrm{cor}} f_{\mathrm{mult}}(\eta_t \alpha_t^2, \mathbf{s}) - \eta_t \alpha_t^2 \boldsymbol{\Sigma}_{\mathrm{cor}}^\top \boldsymbol{\Sigma}_{\mathrm{obs}}^{-1} \boldsymbol{\Sigma}_{\mathrm{cor}} \mathbf{s}\|_2$$
$$+ \|\eta_t \alpha_t \widehat{\boldsymbol{\mu}_{\mathrm{cond}}}(\mathbf{x}_{\mathrm{obs}}) - \eta_t \alpha_t \boldsymbol{\mu}_{\mathrm{cond}}(\mathbf{x}_{\mathrm{obs}})\|_2$$
$$+ \|\boldsymbol{\Sigma}_{\mathrm{miss}} f_{\mathrm{mult}}(\eta_t \alpha_t^2, \mathbf{s}) - \eta_t \alpha_t^2 \boldsymbol{\Sigma}_{\mathrm{miss}} \mathbf{s}\|_2$$
$$+ \|f_{\mathrm{mult}}(\eta_t \sigma_t^2, \mathbf{s}) - \eta_t \sigma_t^2 \mathbf{s}\|_2$$
$$+ \|f_{\mathrm{mult}}(\eta_t, \mathbf{x}_j) - \eta_t \mathbf{x}_j\|_2$$
$$\leq \|\boldsymbol{\Sigma}_{\mathrm{cor}}\|_2 \frac{\kappa(\boldsymbol{\Sigma}_{\mathrm{obs}}) + 3}{2} \epsilon_0 + 2\sqrt{d}\epsilon_{\mathrm{mult}} + \frac{\epsilon}{2} + \|\boldsymbol{\Sigma}_{\mathrm{miss}}\|_2 \sqrt{d}\epsilon_{\mathrm{mult}},$$

where the second term, $\eta_t \alpha_t \widehat{\boldsymbol{\mu}_{\mathrm{cond}}}(\mathbf{x}_{\mathrm{obs}})$, was computed in the first iteration bound and is thus bounded by $\frac{\epsilon}{2}$.

By setting $\epsilon_0 = \left(2\|\boldsymbol{\Sigma}_{\mathrm{cor}}\|_{\mathrm{F}}(\kappa(\boldsymbol{\Sigma}_{\mathrm{obs}}) + 3)\sqrt{H}\right)^{-1} \epsilon$, which leads to the auxiliary gradient descent step count:

$$K_{\mathrm{aux},+} = \left\lceil (\kappa(\boldsymbol{\Sigma}_{\mathrm{obs}}) + 1) \log \left( \frac{\left(\eta_t \alpha_t \|\boldsymbol{\Sigma}_{\mathrm{cor}}\|_2^2 (\kappa(\boldsymbol{\Sigma}_{\mathrm{obs}}) + 3)\sqrt{H}\right) \|\mathbf{s}\|_2}{\lambda_{\min}(\boldsymbol{\Gamma}_{\mathrm{obs}})\lambda_{\min}(\boldsymbol{\Lambda})\epsilon} \right) \right\rceil,$$

and setting $\epsilon_{\mathrm{mult}} = \left(8\sqrt{dN}(\|\boldsymbol{\Sigma}_{\mathrm{miss}}\|_{\mathrm{F}} + 2)\right)^{-1} \epsilon$, which leads to

$$L_{\mathrm{mult},+} = \mathcal{O}\left( \log\left( \frac{dH(\|\mathbf{s}\|_\infty + C_{\mathrm{data}}^\delta)\|\boldsymbol{\Sigma}_{\mathrm{miss}}\|_\infty}{\epsilon} \right) \right),$$

we successfully control the error $\|\xi^+\|_2 = \|\widetilde{\mathbf{s}}^+ - \mathbf{s}^+\|_2 \leq \epsilon$.

**Size of Transformer Architecture for Approximating the Later Steps Major GD** We finally summarize our construction by characterizing the size of the architecture:

- The input to the transformer is of dimension $D \times H$ with $D = 12d + d_e + d_t + 3$.
- In the later major GD steps, , we use $1 + L_{\mathrm{mult,aux},+}$ transformer blocks for each auxiliary GD step, and a total of $K_{\mathrm{aux},+}$ auxiliary GD steps are required. After completing the auxiliary GD, we perform additional matrix multiplications (e.g., multiplying vectors by $\boldsymbol{\Sigma}_{\mathrm{cor}}$, $\boldsymbol{\Sigma}_{\mathrm{cor}}^\top$, and $\boldsymbol{\Sigma}_{\mathrm{miss}}$), which require 3 transformer blocks for attention. Subsequently, $\mathbf{f}_{\mathrm{mult}}$ modules are used to complete the major GD, requiring $L_{\mathrm{mult},+}$ blocks. Thus, the total number of transformer blocks required for each subsequent major GD step is bounded by:

$$L = K_{\mathrm{aux},+}(1 + L_{\mathrm{mult,aux},+}) + 3 + L_{\mathrm{mult},+}$$
$$= \mathcal{O}\left( \kappa_t \kappa(\boldsymbol{\Lambda})\kappa(\boldsymbol{\Gamma}_{\mathrm{obs}}) \log^2\left( \frac{Hd\kappa(\boldsymbol{\Lambda})\kappa(\boldsymbol{\Gamma}_{\mathrm{obs}})}{\sigma_t \epsilon} \right) \right).$$

- The number of transformer blocks is bounded by $M = 4H$.
- Same as the analysis in the first step of major GD, we have the norm of the transformer parameters bounded by

$$\mathcal{O}\left( \sqrt{H d^3}(r^2 + \kappa(\boldsymbol{\Lambda})\kappa(\boldsymbol{\Gamma}_{\mathrm{obs}})\sigma_t^{-1}) \right).$$

And this finishes the proof of Lemma 8.

### C.5.3 Proof of Theorem 1

*Proof.* We formally construct the conditional score approximation transformer as follows:

$$\widetilde{\mathbf{s}}(\mathbf{v}_t, \mathbf{x}_{\mathrm{obs}}) = f_{\mathrm{out}} \circ \underbrace{f_{\mathrm{GD}} \circ \cdots \circ f_{\mathrm{GD}}}_{(K-1)\times f_{\mathrm{GD}}} \circ f_{\mathrm{GD},1} \circ f_{\mathrm{in}}(\mathbf{v}_t, \mathbf{x}_{\mathrm{obs}}).$$

Recalling that $\kappa_t = \kappa(\alpha_t^2 \boldsymbol{\Sigma}_{\mathrm{cond}} + \sigma_t^2 \mathbf{I})$, by the major GD convergence result in Lemma 3, to ensure $\|\widetilde{\mathbf{s}} - \mathbf{s}\|_2 \leq \sigma_t^{-1}\epsilon$, the total major GD iteration number required is upper bounded by $K = \mathcal{O}\left( \kappa_t \log\left( \frac{Hd\kappa_t \kappa(\boldsymbol{\Lambda})\kappa(\boldsymbol{\Gamma}_{\mathrm{obs}})}{\epsilon} \right) \right)$, which is obtained by substituting $\epsilon$ with $\sigma_t^{-1}\left( \frac{2}{\kappa_t + 2} \right)\epsilon$.

Utilizing Lemma 7 and 8, and substituting $\epsilon$ with $\sigma_t^{-1}\left(\frac{2}{\kappa_t+2}\right)\epsilon$, the following transformer configuration can control the error in each major GD step:

$$D = 12d + d_e + d_t + 3, \quad L = \mathcal{O}\left(\kappa_t^2 \kappa(\mathbf{\Lambda})\kappa(\mathbf{\Gamma}_{\mathrm{obs}})\log^3\left(\frac{Hd\kappa_t\kappa(\mathbf{\Lambda})\kappa(\mathbf{\Gamma}_{\mathrm{obs}})}{\epsilon}\right)\right),$$

$$M = 4H, \quad B = \mathcal{O}\left(\sqrt{Hd^3}(r^2 + \kappa(\mathbf{\Lambda})\kappa(\mathbf{\Gamma}_{\mathrm{obs}})\sigma_t^{-1})\right),$$

where $\ell$ is computed by $K$ times the transformer block required in each major GD step.

Finally, by substituting $\sigma_t^{-1}, \kappa_t$ with $\sigma_{t_0}^{-1}, \kappa_{t_0}$, and considering the truncation range which is induced by the decoder ($R = \mathcal{O}(\sigma_{t_0}^{-2}\sqrt{Hd}\kappa(\mathbf{\Lambda})\kappa(\mathbf{\Gamma}_{\mathrm{obs}}))$), we obtain a uniform bound for any $t \in [t_0, T]$. Taking supremum over all admissible $\mathcal{I}_{\mathrm{obs}} \subset \mathcal{I}$, and leverage the relationship that $\kappa(\mathbf{\Sigma}_{\mathrm{cond}}) = \kappa(\mathbf{\Lambda})\kappa(\mathbf{\Gamma}_{\mathrm{obs}})$, we finish the proof of Theorem 1.

$\square$

### C.6 Construction of Attention Layers

In this section, we construct the attention layers used in the transformer architectures built up in C.3 and C.4.

We utilize the added

$$\begin{bmatrix} \mathbf{x}_i & \cdots & \mathbf{x}_j \\ \cdots & \cdots & \cdots \\ 1 & \cdots & 1 \\ 1 & \cdots & 0 \\ 0 & \cdots & 1 \\ \cdots & \cdots & \cdots \end{bmatrix}$$

to construct different types of interaction between different types of samples. (i.e. When we want attention exclusively among observed samples or missing samples, we use the construction method as described in $\mathcal{TB}_{obs}$ below; When we want attention between observed samples and missing samples, while setting all other interactions to zero, we use the construction method as described in $\mathcal{TB}_{cort}$ below. )

The intuition of (16) suggests a construction of a multi-head attention layer. Formally, for an arbitrary value of $m$, we construct four attention heads with ReLU activation. The indicator function $\mathbb{1}\{|i-j|=m\}$ can be realized by calculating the auxiliary product $\mathbf{e}_i^\top \mathbf{e}_j$ of time embedding. To see this, we observe

$$\mathbf{e}_i^\top \mathbf{e}_j = \frac{1}{2}\left(2r^2 - \|\mathbf{e}_i - \mathbf{e}_j\|_2^2\right) = \frac{1}{2}\left(2r^2 - f^2(|i-j|)\right).$$

Therefore, it holds that

$$\mathbb{1}\{|i-j|=m\} = \mathbb{1}\left\{\mathbf{e}_i^\top \mathbf{e}_j = r^2 - \frac{1}{2}f^2(m)\right\},$$

since $f$ Assumption 1 ensures that the time embeddings uniquely identify discrete time gaps through their pairwise distances. Directly approximating an indicator function using a ReLU network can be difficult. Yet we note that $|i-j|$ can only take integer values. Therefore, we can slightly widen the decision band for the indicator function. Specifically, we denote a minimum gap $\Delta = \min_{i=1,\ldots,H-1}\{f^2(i+1) - f^2(i)\}$. Thus, we deduce

$$\mathbb{1}\{|i-j|=m\} = \mathbb{1}\left\{\mathbf{e}_i^\top \mathbf{e}_j \in \left[r^2 - \frac{1}{2}f^2(m) - \frac{1}{4}\Delta, \ r^2 - \frac{1}{2}f^2(m) + \frac{1}{4}\Delta\right]\right\}.$$

We can use four ReLU functions to approximate the right-hand side of the last display, and simultaneously take different type of interaction types into account. We use another indicator function (which can be realized by the 0s and 1s added above) to represent what types of interaction we want in this specific transformer block.

We construct a trapezoid function as follows:

$$
\begin{aligned}
\mathbb{1}\{|i - j| = m\} = {} & \frac{8}{\Delta} \, \mathrm{ReLU} \left( \mathbf{e}_i^\top \mathbf{e}_j - r^2 + \frac{1}{2} f^2(m) + \frac{1}{4}\Delta \right) \\
& - \frac{8}{\Delta} \, \mathrm{ReLU} \left( \mathbf{e}_i^\top \mathbf{e}_j - r^2 + \frac{1}{2} f^2(m) + \frac{1}{8}\Delta \right) \\
& - \frac{8}{\Delta} \, \mathrm{ReLU} \left( \mathbf{e}_i^\top \mathbf{e}_j - r^2 + \frac{1}{2} f^2(m) - \frac{1}{8}\Delta \right) \\
& + \frac{8}{\Delta} \, \mathrm{ReLU} \left( \mathbf{e}_i^\top \mathbf{e}_j - r^2 + \frac{1}{2} f^2(m) - \frac{1}{4}\Delta \right).
\end{aligned}
$$

### C.6.1 Construction of attention matrices related to the observed part

We construct the attention matrices for $\mathcal{TB}_{obs}$ here.

For particular $m$, we utilize

$$
(\mathbf{Q}^1)^\top \mathbf{K}^1 = \mathrm{diag}\left( \left[ \mathbf{0}_{d \times d}, \mathbf{I}_{d_e}, \mathbf{0}_{d_t \times d_t}, \mathbf{0}_{(6d) \times (6d)}, 0, -r^2 + \frac{1}{2} f^2(m) + \frac{1}{4}\Delta, 0, \mathbf{0}_{(3d) \times (3d)} \right] \right),
$$

$$
\mathbf{V}^1 = \begin{bmatrix} \mathbf{0}_{(4d+d_e+d_t) \times (2d+d_e+d_t)} & \mathbf{0}_{(4d+d_e+d_t) \times d} & \mathbf{0}_{(4d+d_e+d_t) \times (5d+3)} \\ \mathbf{0}_{d \times (d+d_e+d_t)} & \frac{8}{\Delta}\gamma_m \mathbf{\Lambda} & \mathbf{0}_{d \times (6d+3)} \\ \mathbf{0}_{(6d+1) \times (2d+d_e+d_t)} & \mathbf{0}_{(6d+1) \times d} & \mathbf{0}_{(6d+1) \times (5d+3)} \end{bmatrix}.
$$

and

$$
(\mathbf{Q}^2)^\top \mathbf{K}^2 = \mathrm{diag}\left( \left[ \mathbf{0}_{d \times d}, \mathbf{I}_{d_e}, \mathbf{0}_{d_t \times d_t}, \mathbf{0}_{(6d) \times (6d)}, 0, -r^2 + \frac{1}{2} f^2(m) + \frac{1}{8}\Delta, 0, \mathbf{0}_{(5d) \times (5d)} \right] \right),
$$

$$
\mathbf{V}^2 = -\mathbf{V}^1
$$

and

$$
(\mathbf{Q}^3)^\top \mathbf{K}^3 = \mathrm{diag}\left( \left[ \mathbf{0}_{d \times d}, \mathbf{I}_{d_e}, \mathbf{0}_{d_t \times d_t}, \mathbf{0}_{(6d) \times (6d)}, 0, -r^2 + \frac{1}{2} f^2(m) - \frac{1}{8}\Delta, 0, \mathbf{0}_{(5d) \times (5d)} \right] \right),
$$

$$
\mathbf{V}^3 = -\mathbf{V}^1
$$

and

$$
(\mathbf{Q}^4)^\top \mathbf{K}^4 = \mathrm{diag}\left( \left[ \mathbf{0}_{d \times d}, \mathbf{I}_{d_e}, \mathbf{0}_{d_t \times d_t}, \mathbf{0}_{(6d) \times (6d)}, 0, -r^2 + \frac{1}{2} f^2(m) - \frac{1}{4}\Delta, 0, \mathbf{0}_{(5d) \times (5d)} \right] \right),
$$

$$
\mathbf{V}^4 = \mathbf{V}^1.
$$

It is easy to verify that

$$
\sum_{a=1}^{4} y_i^T (\mathbf{Q}^a)^\top \mathbf{K}^a y_j = \mathbb{1}\{|i - j| = m\} \mathbb{1}\{i, j \in \mathcal{I}_{\mathrm{obs}}\}.
$$

We can claim that $4H$ attention heads and identity FFN are enough for constructing this block.

### C.6.2 Construction of attention matrices related to the correlation part

We only need to do some small changes to $\mathcal{TB}_{obs}$.

For particular $m$, let

$$(\mathbf{Q}^1)^\top \mathbf{K}^1$$
$$= \mathrm{diag}\left(\left[\mathbf{0}_{d \times d}, \mathbf{I}_{d_e}, \mathbf{0}_{d_t \times d_t}, \mathbf{0}_{(6d) \times (6d)}, -r^2 + \frac{1}{2}f^2(m) + \frac{1}{4}\Delta, \frac{1}{2}\Delta, \frac{1}{2}\Delta, \mathbf{0}_{(5d) \times (5d)}\right]\right),$$

$$\mathbf{V}^1 = \begin{bmatrix} \mathbf{0}_{(2d+d_e+d_t) \times (d+d_e+d_t)} & \mathbf{0}_{(2d+d_e+d_t) \times d} & \mathbf{0}_{(2d+d_e+d_t) \times (6d+3)} \\ \mathbf{0}_{d \times (d+d_e+d_t)} & \frac{8}{\Delta}\gamma_m\Lambda & \mathbf{0}_{d \times (6d+3)} \\ \mathbf{0}_{(8d+1) \times (d+d_e+d_t)} & \mathbf{0}_{(8d+1) \times d} & \mathbf{0}_{(8d+1) \times (6d+3)} \end{bmatrix}.$$

and

$$(\mathbf{Q}^2)^\top \mathbf{K}^2$$
$$= \mathrm{diag}\left(\left[\mathbf{0}_{d \times d}, \mathbf{I}_{d_e}, \mathbf{0}_{d_t \times d_t}, \mathbf{0}_{(6d) \times (6d)}, -r^2 + \frac{1}{2}f^2(m) + \frac{1}{8}\Delta, \frac{1}{2}\Delta, \frac{1}{2}\Delta, \mathbf{0}_{(5d) \times (5d)}\right]\right),$$

$$\mathbf{V}^2 = -\mathbf{V}^1$$

and

$$(\mathbf{Q}^3)^\top \mathbf{K}^3$$
$$= \mathrm{diag}\left(\left[\mathbf{0}_{d \times d}, \mathbf{I}_{d_e}, \mathbf{0}_{d_t \times d_t}, \mathbf{0}_{(6d) \times (6d)}, -r^2 + \frac{1}{2}f^2(m) - \frac{1}{8}\Delta, \frac{1}{2}\Delta, \frac{1}{2}\Delta, \mathbf{0}_{(5d) \times (5d)}\right]\right),$$

$$\mathbf{V}^3 = -\mathbf{V}^1$$

and

$$(\mathbf{Q}^4)^\top \mathbf{K}^4$$
$$= \mathrm{diag}\left(\left[\mathbf{0}_{d \times d}, \mathbf{I}_{d_e}, \mathbf{0}_{d_t \times d_t}, \mathbf{0}_{(6d) \times (6d)}, -r^2 + \frac{1}{2}f^2(m) - \frac{1}{4}\Delta, \frac{1}{2}\Delta, \frac{1}{2}\Delta, \mathbf{0}_{(5d) \times (5d)}\right]\right).$$

$$\mathbf{V}^4 = \mathbf{V}^1$$

It is easy to verify that

$$\sum_{a=1}^{4} y_i^T (\mathbf{Q}^a)^\top \mathbf{K}^a y_j = \mathbb{1}\{|i - j| = m\}\mathbb{1}\{\{i \in \mathcal{I}_{\mathrm{obs}}, j \in \mathcal{I}_{\mathrm{miss}}\} \cup \{i \in \mathcal{I}_{\mathrm{miss}}, j \in \mathcal{I}_{\mathrm{obs}}\}\}.$$

We can thus state that $4H$ attention heads and identity FFN are enough for constructing this block.

# D  Proofs of Theorem 2 and Corollary 1

In this section, we provide the detailed proof of Theorem 2 and Corollary 1.

Firstly, we introduce some notations specifically for this part for sake of simplicity. We denote our training set with $n$ i.i.d. samples as

$$\mathcal{D}^{(n)} = \{\mathbf{x}^{(i)}\}_{i=1}^n = \{(\mathbf{x}_{\text{miss}}^{(i)}, \mathbf{x}_{\text{obs}}^{(i)})\}_{i=1}^n = \{(\mathbf{x}^{(i)}, \mathbf{y}^{(i)})\}_{i=1}^n.$$

We introduce the corollary below which will act as an significant role in our later proof:

**Corollary 2.** *By choosing the transformer architecture $\mathcal{T}(D, L, M, B, R)$ as in Theorem 1, the early-stopping time $t_0 < 1$ and the terminal time $T = \mathcal{O}(\log n)$, it holds that*

$$\mathbb{E}_{\{(\mathbf{x}^{(i)}, \mathbf{y}^{(i)})\}_{i=1}^n} [\mathcal{R}(\widehat{\mathbf{s}})] \lesssim \frac{H d^2 \kappa_{t_0}^4 \kappa^2(\mathbf{\Lambda}) \kappa^2(\mathbf{\Gamma}_{\text{obs}}) t_0^{-1}}{n} \log(H d \kappa(\mathbf{\Lambda}) \kappa(\mathbf{\Gamma}_{\text{obs}}) n t_0^{-1}),$$

*where $\kappa_{t_0} := \kappa(\alpha_t^2 \mathbf{\Sigma}_{\text{cond}} + \sigma_t^2 \mathbf{I})$.*

The proof of Corollary 2 is deferred to Appendix E.

## D.1  Proof of Theorem 2

Although our assumption on Gaussian processes does not ensure the Novikov's condition to hold, according to [Chen et al., 2022], as long as we have bounded the second moment for the score estimation error and finite KL divergence w.r.t the standard Gaussian, we could still adopt Girsanov's Theorem and bound the KL divergence between the two distribution. We restate the Lemma as follows:

**Lemma 10** (Corollary D.1 in [Oko et al., 2023], see also Theorem 2 in [Chen et al., 2022]). *Let $p_0$ be a probability distribution, and let $Y = \{Y_t\}_{t \in [0,T]}$ and $Y' = \{Y'_t\}_{t \in [0,T]}$ be two stochastic processes that satisfy the following SDEs:*

$$dY_t = s(Y_t, t)dt + dW_t, \quad Y_0 \sim p_0,$$
$$dY'_t = s'(Y'_t, t)dt + dW_t, \quad Y'_0 \sim p_0.$$

*We further define the distributions of $Y_t$ and $Y'_t$ by $p_t$ and $p'_t$. Suppose that*

$$\int_x p_t(x) \|(s - s')(x, t)\|^2 dx \le C \tag{20}$$

*for any $t \in [0, T]$. Then we have*

$$\text{KL}\,(p_T \| p'_T) \le \int_0^T \frac{1}{2} \int_x p_t(x) \|(s - s')(x, t)\|^2 dx\, dt.$$

Equipped with Corollary 2 and Lemma 10, we are ready to prove Theorem 2.

*Proof of Theorem 2.* Firstly, following the proof of Lemma 12, we can easily verify that for any $\mathbf{s} \in \mathcal{T}(D, L, M, B, R)$,

$$\int_x p_t(\mathbf{v}_t \mid \mathbf{y}) \|s(\mathbf{v}_t, \mathbf{y}, t) - \nabla \log p_t(\mathbf{v}_t \mid \mathbf{y})\|_2^2 d\mathbf{x} \lesssim \frac{1}{\sigma_t^4}.$$

Thus, the condition (10) holds for all $t \in [t_0, T]$, which means that we could apply Girsanov's theorem in this time range.

To further distinguish the SDE defined in (1), (2), and (3), we denote the distribution of $\mathbf{x}_t, \mathbf{v}_t, \widehat{\mathbf{v}}_t$ as $P_t, P_t^{\leftarrow}, \widehat{P}_t^{\leftarrow}$, respectively. Additionally, we need to introduce another intermediate backward process between $P_t^{\leftarrow}, \widehat{P}_t^{\leftarrow}$ as follows

$$d\mathbf{v}_t'^{\leftarrow} = \left[\frac{1}{2}\mathbf{v}_t'^{\leftarrow} + \nabla \log p_{T-t}(\mathbf{v}_t'^{\leftarrow}|\mathbf{y})\right] dt + d\bar{\mathbf{w}}_t \quad \text{with} \quad \mathbf{v}_0'^{\leftarrow} \sim \mathcal{N}(\mathbf{0}, \mathbf{I}_{d|\mathcal{I}_{\text{miss}}|}),$$

and we denote the marginal distribution of $\mathbf{v}_t'^{\leftarrow}$ (conditioned on $\mathbf{y}$) as $P'_{T-t}(\cdot|\mathbf{y})$.

Equipped with these notations, we can decompose the total variation between $P$ and $\widehat{P}_{t_0}^{\leftarrow}$ as

$$\mathbb{E}_{\mathbf{y}}\left[\mathrm{TV}(P,\widehat{P}_{t_0}^{\leftarrow})\right] \leq \mathbb{E}_{\mathbf{y}}\left[\mathrm{TV}(P,P_{t_0})+\mathrm{TV}(P_{t_0},P_{t_0}^{\leftarrow})+\mathrm{TV}(P_{t_0}^{\leftarrow},P_{t_0}'^{\leftarrow})+\mathrm{TV}(P_{t_0}'^{\leftarrow},\widehat{P}_{t_0}^{\leftarrow})\right]$$

$$= \mathbb{E}_{\mathbf{y}}\left[\mathrm{TV}(P,P_{t_0})+\mathrm{TV}(P_{t_0}^{\leftarrow},P_{t_0}'^{\leftarrow})+\mathrm{TV}(P_{t_0}'^{\leftarrow},\widehat{P}_{t_0}^{\leftarrow})\right]. \tag{21}$$

We denote $\{\xi_i\}_{i=1}^{d|\mathcal{I}_{\mathrm{miss}}|}$ as the eigenvalues of $\boldsymbol{\Sigma}_{\mathrm{cond}}$, and we can do eigenvalue decompositions to $\boldsymbol{\Sigma}_{\mathrm{cond}}$ as $\boldsymbol{\Sigma}_{\mathrm{cond}}=\boldsymbol{Q}\boldsymbol{\Xi}\boldsymbol{Q}^{\top}$, where $(\boldsymbol{\Xi})_{ii}=\xi_i$.

Considering the second last term,, by Data Processing Inequality and Pinsker's Inequality (see e.g. Lemma 2 in [Canonne, 2022]), we have

$\mathbb{E}_{\mathbf{y}}[\mathrm{TV}(P_{t_0}^{\leftarrow},P_{t_0}'^{\leftarrow})]$

$$\lesssim \sqrt{\mathbb{E}_{\mathbf{y}}[\mathrm{KL}(P_{t_0}^{\leftarrow}||P_{t_0}'^{\leftarrow})]} \quad \text{(Pinsker's Inequality)}$$

$$\lesssim \sqrt{\mathbb{E}_{\mathbf{y}}[\mathrm{KL}(P_T||\mathcal{N}(\mathbf{0},\mathbf{I}_{d|\mathcal{I}_{\mathrm{miss}}|}))]} \quad \text{(Data Processing Inequality)}$$

$$\lesssim \sqrt{\mathbb{E}_{\mathbf{y}}[\mathrm{KL}(P||\mathcal{N}(\mathbf{0},\mathbf{I}_{d|\mathcal{I}_{\mathrm{miss}}|}))]}\exp(-T)$$

$$= \sqrt{\mathbb{E}_{\mathbf{y}}[-\log(|\boldsymbol{\Sigma}_{\mathrm{cond}}|)+\mathrm{tr}(\boldsymbol{\Sigma}_{\mathrm{cond}})+\mathbf{y}^{\top}\boldsymbol{\Sigma}_{\mathrm{obs}}^{-1}\boldsymbol{\Sigma}_{\mathrm{cor}}\boldsymbol{\Sigma}_{\mathrm{cor}}^{\top}\boldsymbol{\Sigma}_{\mathrm{obs}}^{-1}\mathbf{y}]-d|\mathcal{I}_{\mathrm{miss}}|}\exp(-T)$$

$$\leq \sqrt{-\log(|\boldsymbol{\Sigma}_{\mathrm{cond}}|)+\mathrm{tr}(\boldsymbol{\Sigma}_{\mathrm{cond}})+\mathbb{E}_{\mathbf{z}\sim\mathcal{N}(\mathbf{0},\mathbf{I})}[\mathbf{z}^{\top}\boldsymbol{\Sigma}_{\mathrm{obs}}^{-\frac{1}{2}}\boldsymbol{\Sigma}_{\mathrm{cor}}\boldsymbol{\Sigma}_{\mathrm{cor}}^{\top}\boldsymbol{\Sigma}_{\mathrm{obs}}^{-\frac{1}{2}}\mathbf{z}]-d|\mathcal{I}_{\mathrm{miss}}|}\exp(-T)$$

$$= \sqrt{-\log(|\boldsymbol{\Sigma}_{\mathrm{cond}}|)+\mathrm{tr}(\boldsymbol{\Sigma}_{\mathrm{cond}}+\boldsymbol{\Sigma}_{\mathrm{cor}}^{\top}\boldsymbol{\Sigma}_{\mathrm{obs}}^{-1}\boldsymbol{\Sigma}_{\mathrm{cor}})-d|\mathcal{I}_{\mathrm{miss}}|}\exp(-T)$$

$$\lesssim \sqrt{-\log(|\boldsymbol{\Sigma}_{\mathrm{cond}}|)+\mathrm{tr}(\boldsymbol{\Sigma}_{\mathrm{cond}}+\boldsymbol{\Sigma}_{\mathrm{cor}}^{\top}\boldsymbol{\Sigma}_{\mathrm{obs}}^{-1}\boldsymbol{\Sigma}_{\mathrm{cor}})-d|\mathcal{I}_{\mathrm{miss}}|}\exp(-T)$$

$$\lesssim \sqrt{-Hd\log(\xi_{\min})+\mathrm{tr}(\boldsymbol{\Sigma}_{\mathrm{miss}})-d|\mathcal{I}_{\mathrm{miss}}|}\exp(-T)$$

$$\lesssim \sqrt{Hd\xi_{\min}^{-1}}\exp(-T), \tag{22}$$

where we leverage the close-form solution of the KL-divergence between two gaussian distributions in the first equality.

Regarding the first term, by Pinsker's Inequality and the close-form solution of the KL-divergence between Gaussian distributions [Pardo, 2018], we have

$\mathbb{E}_{\mathbf{y}}\left[\mathrm{TV}(P(\cdot|\mathbf{y}),P_{t_0}(\cdot|\mathbf{y}))\right]$

$$= \mathbb{E}_{\mathbf{y}}\left[\mathrm{TV}(P_{t_0}(\cdot|\mathbf{y}),P(\cdot|\mathbf{y}))\right]$$

$$\lesssim \mathbb{E}_{\mathbf{y}}\left[\sqrt{\mathrm{KL}(P_{t_0}(\cdot|\mathbf{y})||P(\cdot|\mathbf{y}))}\right]$$

$$= \mathbb{E}_{\mathbf{y}}\left[\sqrt{\mathrm{KL}(\mathcal{N}(\alpha_{t_0}\boldsymbol{\Sigma}_{\mathrm{cor}}^{\top}\boldsymbol{\Sigma}_{\mathrm{obs}}^{-1}\mathbf{y},\alpha_{t_0}^2\boldsymbol{\Sigma}_{\mathrm{cond}}+\sigma_{t_0}^2\mathbf{I}_{d|\mathcal{I}_{\mathrm{miss}}|})||\mathcal{N}(\boldsymbol{\Sigma}_{\mathrm{cor}}^{\top}\boldsymbol{\Sigma}_{\mathrm{obs}}^{-1}\mathbf{y},\boldsymbol{\Sigma}_{\mathrm{cond}}))}\right]$$

$$\leq \sqrt{\mathbb{E}_{\mathbf{y}}\left[\mathrm{KL}(\mathcal{N}(\alpha_{t_0}\boldsymbol{\Sigma}_{\mathrm{cor}}^{\top}\boldsymbol{\Sigma}_{\mathrm{obs}}^{-1}\mathbf{y},\alpha_{t_0}^2\boldsymbol{\Sigma}_{\mathrm{cond}}+\sigma_{t_0}^2\mathbf{I}_{d|\mathcal{I}_{\mathrm{miss}}|})||\mathcal{N}(\boldsymbol{\Sigma}_{\mathrm{cor}}^{\top}\boldsymbol{\Sigma}_{\mathrm{obs}}^{-1}\mathbf{y},\boldsymbol{\Sigma}_{\mathrm{cond}}))\right]},$$

Leveraging the close-form solution of the KL-divergence between two gaussian distributions, we further have

$$\mathbb{E}_{\mathbf{y}}\left[\mathrm{KL}(\mathcal{N}(\alpha_{t_0}\boldsymbol{\Sigma}_{\mathrm{cor}}^{\top}\boldsymbol{\Sigma}_{\mathrm{obs}}^{-1}\mathbf{y},\alpha_{t_0}^2\boldsymbol{\Sigma}_{\mathrm{cond}}+\sigma_{t_0}^2\mathbf{I}_{d|\mathcal{I}_{\mathrm{miss}}|})||\mathcal{N}(\boldsymbol{\Sigma}_{\mathrm{cor}}^{\top}\boldsymbol{\Sigma}_{\mathrm{obs}}^{-1}\mathbf{y},\boldsymbol{\Sigma}_{\mathrm{cond}}))\right]$$

$$\lesssim \underbrace{-\log\left(\frac{|\alpha_{t_0}^2\boldsymbol{\Sigma}_{\mathrm{cond}}+\sigma_{t_0}^2\mathbf{I}|}{|\boldsymbol{\Sigma}_{\mathrm{cond}}|}\right)}_{A}+\underbrace{\mathrm{tr}(\boldsymbol{\Sigma}_{\mathrm{cond}}^{-1}(\alpha_{t_0}^2\boldsymbol{\Sigma}_{\mathrm{cond}}+\sigma_{t_0}^2\mathbf{I}))}_{B}$$

$$+\underbrace{(1-\alpha_{t_0})^2\mathbb{E}_{\mathbf{y}}\left[\mathbf{y}^{\top}\boldsymbol{\Sigma}_{\mathrm{obs}}^{-1}\boldsymbol{\Sigma}_{\mathrm{cor}}\boldsymbol{\Sigma}_{\mathrm{cond}}^{-1}\boldsymbol{\Sigma}_{\mathrm{cor}}^{\top}\boldsymbol{\Sigma}_{\mathrm{obs}}^{-1}\mathbf{y}\right]}_{C}-d|\mathcal{I}_{\mathrm{miss}}|.$$

Considering term $A$, we have

$$|\alpha_{t_0}^2\boldsymbol{\Sigma}_{\mathrm{cond}}+\sigma_{t_0}^2\mathbf{I}|=|\boldsymbol{\Sigma}_{\mathrm{cond}}|\cdot|\alpha_{t_0}^2\mathbf{I}+\sigma_{t_0}^2\boldsymbol{\Sigma}_{\mathrm{cond}}^{-1}|$$

$$=|\boldsymbol{\Sigma}_{\mathrm{cond}}|\cdot\prod_{i=1}^{d|\mathcal{I}_{\mathrm{miss}}|}(\alpha_{t_0}^2+\sigma_{t_0}^2\xi_i^{-1}).$$

Then we obtain

$$A = - \sum_{i=1}^{d|\mathcal{I}_{\text{miss}}|} \log(\alpha_{t_0}^2 + \sigma_{t_0}^2 \xi_i^{-1}).$$

Regarding term $B$, we have

$$
\begin{aligned}
B = \text{tr}(\boldsymbol{\Sigma}_{\text{cond}}^{-1}(\alpha_{t_0}^2 \boldsymbol{\Sigma}_{\text{cond}} + \sigma_{t_0}^2 \mathbf{I})) &= \text{tr}(\boldsymbol{Q}(\alpha_{t_0}^2 \boldsymbol{\Xi} + \sigma_{t_0}^2 \mathbf{I})\boldsymbol{Q}^\top (\boldsymbol{Q}\boldsymbol{\Xi}^{-1}\boldsymbol{Q}^\top)) \\
&= \text{tr}((\alpha_{t_0}^2 \boldsymbol{\Xi} + \sigma_{t_0}^2 \mathbf{I})\boldsymbol{\Xi}^{-1}) \\
&= \sum_{i=1}^{d|\mathcal{I}_{\text{miss}}|} (\alpha_{t_0}^2 + \sigma_{t_0}^2 \xi_i^{-1}) \\
&\leq d|\mathcal{I}_{\text{miss}}| + d|\mathcal{I}_{\text{miss}}|\xi_{\min}^{-1}\sigma_t^2
\end{aligned}
$$

Considering term $C$, we have

$$
\begin{aligned}
C &= (1 - \alpha_{t_0})^2 \mathbb{E}_{\mathbf{y}} \left[ \mathbf{y}^\top \boldsymbol{\Sigma}_{\text{obs}}^{-1} \boldsymbol{\Sigma}_{\text{cor}} \boldsymbol{\Sigma}_{\text{cond}}^{-1} \boldsymbol{\Sigma}_{\text{cor}}^\top \boldsymbol{\Sigma}_{\text{obs}}^{-1} \mathbf{y} \right] \\
&= (1 - \alpha_{t_0})^2 \mathbb{E}_{\mathbf{z} \sim \mathcal{N}(0, \mathbf{I}_{d|\mathcal{I}_{\text{obs}}|})} \left[ \mathbf{z}^\top \boldsymbol{\Sigma}_{\text{obs}}^{-\frac{1}{2}} \boldsymbol{\Sigma}_{\text{cor}} \boldsymbol{\Sigma}_{\text{cond}}^{-1} \boldsymbol{\Sigma}_{\text{cor}}^\top \boldsymbol{\Sigma}_{\text{obs}}^{-\frac{1}{2}} \mathbf{z} \right] \\
&= (1 - \alpha_{t_0})^2 \text{tr}(\boldsymbol{\Sigma}_{\text{cor}}^\top \boldsymbol{\Sigma}_{\text{obs}}^{-1} \boldsymbol{\Sigma}_{\text{cor}} \boldsymbol{\Sigma}_{\text{cond}}^{-1}).
\end{aligned}
$$

Thus, with $\alpha_{t_0} = e^{-\frac{t_0}{2}}, \sigma_{t_0} = \sqrt{1 - e^{-t_0}}$, we can take $t_0 = \mathcal{O}(\xi_{\min} n^{-\frac{1}{2}})$, and

$$\mathbb{E}_{\mathbf{y}} \left[ \text{TV}(P(\cdot|\mathbf{y}), P_{t_0}(\cdot|\mathbf{y})) \right] \lesssim n^{-\frac{1}{2}}(Hd)^{\frac{1}{2}}. \tag{23}$$

Combining (21), (23), (22), and invoking Lemma 10, we have:

$$
\begin{aligned}
\mathbb{E}_{\mathbf{y}} \left[ \text{TV}(P, \widehat{P}_{t_0}^{\leftarrow}) \right] &\lesssim n^{-\frac{1}{2}}(Hd)^{\frac{1}{2}} + \sqrt{Hd(\xi_{\min}^{-1} + \kappa(\boldsymbol{\Lambda}))} \exp(-T) + \\
&\quad + \mathbb{E}_{\mathbf{y}} \left[ \sqrt{\int_{t_0}^T \frac{1}{2} \int_{\mathbf{v}_t} p_t(\mathbf{v}_t|\mathbf{y})\|\widehat{\mathbf{s}}(\mathbf{v}_t, \mathbf{y}, t) - \nabla \log p_t(\mathbf{v}_t|\mathbf{y})\|_2^2 d\mathbf{v}_t dt} \right] \quad (24) \\
&\lesssim n^{-\frac{1}{2}}(Hd)^{\frac{1}{2}} + \sqrt{Hd(\xi_{\min}^{-1} + \kappa(\boldsymbol{\Lambda}))} \exp(-T) + \sqrt{\mathcal{R}(\widehat{\mathbf{s}})}.
\end{aligned}
$$

Plugging in the result in Corollary 2 (taking $T = \mathcal{O}(\log n)$, and $t_0 = \mathcal{O}(\xi_{\min} n^{-\frac{1}{2}})$), we finally obtain

$$
\begin{aligned}
\mathbb{E}_{\mathcal{D}} \left[ \mathbb{E}_{\mathbf{y}} \left[ \text{TV}(P, \widehat{P}_{t_0}^{\leftarrow}) \right] \right] &\lesssim \frac{H^{\frac{1}{2}} d\kappa^{\frac{5}{2}}(\boldsymbol{\Gamma}_{\text{cond}})\kappa(\boldsymbol{\Lambda})\kappa(\boldsymbol{\Gamma}_{\text{obs}})}{n^{\frac{1}{2}}} \log^{\frac{1}{2}}(Hd\kappa(\boldsymbol{\Lambda})\kappa(\boldsymbol{\Gamma}_{\text{obs}})n), \\
&= \widetilde{O}(\sqrt{Hd^2\kappa^5(\boldsymbol{\Sigma}_{\text{cond}})\kappa^2(\boldsymbol{\Sigma}_{\text{obs}})}/\sqrt{n}).
\end{aligned}
$$

where $\boldsymbol{\Gamma}_{\text{cond}} = \boldsymbol{\Gamma}_{\text{miss}} - \boldsymbol{\Gamma}_{\text{cor}}^\top \boldsymbol{\Gamma}_{\text{obs}}^{-1} \boldsymbol{\Gamma}_{\text{cor}}$. We complete our proof.

$\square$

### D.2 Proof of Corollary 1

With Theorem 2 established, the conclusion in Corollary 1 goes straightforward.

*Proof.* Firstly, by the definition of total variation distance, we have the relationship

$$|\widehat{P}_{t_0}(\mathbf{x}_{\text{miss}}^* \in \widehat{\mathcal{CR}}_{1-\alpha}^*) - P(\mathbf{x}_{\text{miss}}^* \in \widehat{\mathcal{CR}}_{1-\alpha}^*)| \leq \text{TV}(\widehat{P}_{t_0}(\cdot|\mathbf{x}_{\text{obs}}^*), P(\cdot|\mathbf{x}_{\text{obs}}^*)).$$

Following the decomposition in (21), we can obtain

$$
\begin{aligned}
&\text{TV}(\widehat{P}_{t_0}(\cdot|\mathbf{x}_{\text{obs}}^*), P(\cdot|\mathbf{x}_{\text{obs}}^*)) \\
&\quad \leq \text{TV}(P(\cdot|\mathbf{x}_{\text{obs}}^*), P_{t_0}(\cdot|\mathbf{x}_{\text{obs}}^*)) + \text{TV}(P_{t_0}(\cdot|\mathbf{x}_{\text{obs}}^*), P_{t_0}^{\leftarrow}(\cdot|\mathbf{x}_{\text{obs}}^*)) \\
&\quad\quad + \text{TV}(P_{t_0}^{\leftarrow}(\cdot|\mathbf{x}_{\text{obs}}^*, \widehat{P}_{t_0}(\cdot|\mathbf{x}_{\text{obs}}^*))).
\end{aligned}
$$

Regarding the right hand side, following the derivation in (24), we can bound each term similarly by taking $t_0 = \mathcal{O}(\lambda_{\min}(\mathbf{\Sigma}_{\text{cond}})n^{-\frac{1}{2}})$ and $T = \mathcal{O}(\log n)$.

For the second term, we leverage the close-form solution of the KL-divergence between two gaussian distributions:

$$\text{TV}(P_{t_0}(\cdot|\mathbf{x}^*_{\text{obs}}), P^{\leftarrow}_{t_0}(\cdot|\mathbf{x}^*_{\text{obs}}))$$

$$\lesssim \sqrt{-\log(|\mathbf{\Sigma}_{\text{cond}}|) + \text{tr}(\mathbf{\Sigma}_{\text{cond}}) + \mathbf{x}^*_{\text{obs}}{}^\top \mathbf{\Sigma}^{-1}_{\text{obs}} \mathbf{\Sigma}_{\text{cor}} \mathbf{\Sigma}^\top_{\text{cor}} \mathbf{\Sigma}^{-1}_{\text{obs}} \mathbf{x}^*_{\text{obs}} - d|\mathcal{I}_{\text{miss}}|} \exp(-T)$$

$$\leq \frac{1}{n}\sqrt{-\log(|\mathbf{\Sigma}_{\text{cond}}|) + \text{tr}(\mathbf{\Sigma}_{\text{cond}}) + \mathbf{x}^*_{\text{obs}}{}^\top \mathbf{\Sigma}^{-1}_{\text{obs}} \mathbf{\Sigma}_{\text{cor}} \mathbf{\Sigma}^\top_{\text{cor}} \mathbf{\Sigma}^{-1}_{\text{obs}} \mathbf{x}^*_{\text{obs}} - d|\mathcal{I}_{\text{miss}}|}$$

$$\lesssim \frac{1}{n}\sqrt{-Hd\log(\lambda_{\min}(\mathbf{\Sigma}_{\text{cond}})) + \mathbf{x}^*_{\text{obs}}{}^\top \mathbf{\Sigma}^{-1}_{\text{obs}} \mathbf{\Sigma}_{\text{cor}} \mathbf{\Sigma}^\top_{\text{cor}} \mathbf{\Sigma}^{-1}_{\text{obs}} \mathbf{x}^*_{\text{obs}}}$$

$$\lesssim \epsilon^{(n)}_{\text{diff}} + \frac{1}{n}\sqrt{\mathbf{x}^*_{\text{obs}}{}^\top \mathbf{\Sigma}^{-1}_{\text{obs}} \mathbf{\Sigma}_{\text{cor}} \mathbf{\Sigma}^\top_{\text{cor}} \mathbf{\Sigma}^{-1}_{\text{obs}} \mathbf{x}^*_{\text{obs}}}.$$

For the last term, recalling the definition of $\text{DS}(P_{\mathbf{x}_{\text{obs}}}, P_{\mathbf{x}^*_{\text{obs}}}; \mathcal{G})$, we have

$$\mathbb{E}_{\mathcal{D}^{(n)}}\left[\text{TV}(P^{\leftarrow}_{t_0}(\cdot|\mathbf{x}^*_{\text{obs}}, \widehat{P}_{t_0}(\cdot|\mathbf{x}^*_{\text{obs}})))\right]$$

$$\lesssim \mathbb{E}_{\mathcal{D}^{(n)}}\left[\sqrt{\int_{t_0}^T \frac{1}{2} \int_{\mathbf{v}_t} p_t(\mathbf{v}_t|\mathbf{x}^*_{\text{obs}})||\widehat{\mathbf{s}}(\mathbf{v}_t, \mathbf{x}^*_{\text{obs}}, t) - \nabla \log p_t(\mathbf{v}_t|\mathbf{x}^*_{\text{obs}})||_2^2 \mathbf{dv}_t \mathbf{d}t}\right]$$

$$\lesssim \mathbb{E}_{\mathcal{D}^{(n)}}\left[\sqrt{\int_{t_0}^T \mathbb{E}_{\mathbf{v}_t}[||\widehat{\mathbf{s}}(\mathbf{v}_t, \mathbf{x}^*_{\text{obs}}, t) - \nabla \log p_t(\mathbf{v}_t|\mathbf{x}^*_{\text{obs}})||_2^2]\mathbf{d}t}\right]$$

$$\lesssim \mathbb{E}_{\mathcal{D}^{(n)}}\left[\sqrt{\int_{t_0}^T \mathbb{E}_{\mathbf{v}_t, \mathbf{y} \sim P_{\mathbf{x}_{\text{obs}}}}[||\widehat{\mathbf{s}}(\mathbf{v}_t, \mathbf{y}, t) - \nabla \log p_t(\mathbf{v}_t|\mathbf{y})||_2^2]\mathbf{d}t \cdot \sup_{\ell \in \mathcal{G}} \frac{\mathbb{E}_{\mathbf{y}=\mathbf{x}^*_{\text{obs}}}[\ell(\mathbf{y})]}{\mathbb{E}_{\mathbf{y} \sim P_{\mathbf{x}_{\text{obs}}}}[\ell(\mathbf{y})]}}\right]$$

$$\lesssim \sqrt{\mathbb{E}_{\mathcal{D}^{(n)}}[\mathcal{R}(\widehat{\mathbf{s}})]} \cdot \sqrt{\text{DS}(P_{\mathbf{x}^*_{\text{obs}}}, P_{\mathbf{x}^*_{\text{obs}}}; \mathcal{G})}$$

$$\lesssim \epsilon^{(n)}_{\text{diff}} \cdot \sqrt{\text{DS}(P_{\mathbf{x}^*_{\text{obs}}}, P_{\mathbf{x}^*_{\text{obs}}}; \mathcal{G})}.$$

Let

$$\psi(\mathbf{x}^*_{\text{obs}}) := \max\left\{\sqrt{\lambda_{\min}(\mathbf{\Sigma}_{\text{cond}})\mathbf{x}^*_{\text{obs}}{}^\top \mathbf{\Sigma}^{-\frac{1}{2}}_{\text{obs}} \mathbf{\Sigma}_{\text{cor}} \mathbf{\Sigma}^{-1}_{\text{cond}} \mathbf{\Sigma}^\top_{\text{cor}} \mathbf{\Sigma}^{-\frac{1}{2}}_{\text{obs}} \mathbf{x}^*_{\text{obs}}},\right.$$

$$\left.\sqrt{\mathbf{x}^*_{\text{obs}}{}^\top \mathbf{\Sigma}^{-1}_{\text{obs}} \mathbf{\Sigma}_{\text{cor}} \mathbf{\Sigma}^\top_{\text{cor}} \mathbf{\Sigma}^{-1}_{\text{obs}} \mathbf{x}^*_{\text{obs}}}\right\}.$$

For the first term, leveraging the result in (23), and the decomposition of term C in the proof Theorem 2, we have

$$\text{TV}(P(\cdot|\mathbf{x}^*_{\text{obs}}), P_{t_0}(\cdot|\mathbf{x}^*_{\text{obs}})) \lesssim n^{-\frac{1}{2}}(Hd)^{\frac{1}{2}} + n^{-\frac{1}{2}}\psi(\mathbf{x}^*_{\text{obs}}) \lesssim \epsilon^{(n)}_{\text{diff}} + n^{-\frac{1}{2}}\psi(\mathbf{x}^*_{\text{obs}}).$$

Finally, we can combine all the bounds above to obtain

$$\mathbb{E}_{\mathcal{D}^{(n)}}[|\widehat{P}_{t_0}(\mathbf{x}^*_{\text{miss}} \in \widehat{\mathcal{CR}}^*_{1-\alpha}) - P(\mathbf{x}^*_{\text{miss}} \in \widehat{\mathcal{CR}}^*_{1-\alpha})|]$$

$$\lesssim \mathbb{E}_{\mathcal{D}^{(n)}}[\text{TV}(P(\cdot|\mathbf{x}^*_{\text{obs}}), P_{t_0}(\cdot|\mathbf{x}^*_{\text{obs}})) + \text{TV}(P_{t_0}(\cdot|\mathbf{x}^*_{\text{obs}}), P^{\leftarrow}_{t_0}(\cdot|\mathbf{x}^*_{\text{obs}}))$$

$$+ \text{TV}(P^{\leftarrow}_{t_0}(\cdot|\mathbf{x}^*_{\text{obs}}, \widehat{P}_{t_0}(\cdot|\mathbf{x}^*_{\text{obs}})))]$$

$$\lesssim n^{-\frac{1}{2}}\psi(\mathbf{x}^*_{\text{obs}}) + \epsilon^{(n)}_{\text{diff}} \cdot \sqrt{\text{DS}(P_{\mathbf{x}^*_{\text{obs}}}, P_{\mathbf{x}^*_{\text{obs}}}; \mathcal{G})}$$

$$\lesssim \epsilon^{(n)}_{\text{diff}} \cdot \sqrt{\text{DS}(P_{\mathbf{x}^*_{\text{obs}}}, P_{\mathbf{x}^*_{\text{obs}}}; \mathcal{G})} + n^{-\frac{1}{2}}\psi(\mathbf{x}^*_{\text{obs}}),$$

and the corollary follows.

$\square$

# E   Proof of Corollary 2

In this section, we provide the detailed proof of Corollary 2.

**Training Loss**   During training, given a state $\mathbf{v}_t = \alpha_t \mathbf{v}_0 + \sigma_t \mathbf{z}, \mathbf{z} \sim \mathcal{N}(\mathbf{0}, \mathbf{I}_{d|\mathcal{I}_{\mathrm{miss}}|}), \mathbf{v}_0 = \mathbf{x}_{\mathrm{miss}}$, we aim to minimize the ideal risk function:

$$\mathcal{R}(\mathbf{s}) = \int_{t_0}^{T} \mathbb{E}_{\mathbf{v}_t, \mathbf{x}_{\mathrm{obs}}} \left[ \|\mathbf{s}(\mathbf{v}_t, \mathbf{x}_{\mathrm{obs}}, t) - \nabla_{\mathbf{v}_t} \log p_t(\mathbf{v}_t | \mathbf{x}_{\mathrm{obs}}) \|_2^2 \right] dt. \tag{25}$$

However, in practice, the objective (25) is not directly accessible. According to Lemma C.3 in Vincent [2011], an equivalent objective function $\mathcal{L}(\mathbf{s})$, which differs from $\mathcal{R}(\mathbf{s})$ only by a constant, can be used for optimization:

$$\mathcal{L}(\mathbf{s}) = \int_{t_0}^{T} \mathbb{E}_{(\mathbf{x}_{\mathrm{miss}}, \mathbf{x}_{\mathrm{obs}})} \left[ \mathbb{E}_{\mathbf{v}_t | \mathbf{v}_0 = \mathbf{x}_{\mathrm{miss}}} \left[ \|\mathbf{s}(\mathbf{v}_t, \mathbf{x}_{\mathrm{obs}}, t) - \nabla_{\mathbf{v}_t} \log \phi_t(\mathbf{v}_t | \mathbf{v}_0) \|_2^2 \right] \right] dt. \tag{26}$$

Here, $\phi_t$ is the Gaussian transition kernel of the forward process, satisfying $\nabla \log \phi_t(\mathbf{v}_t | \mathbf{v}_0) = \frac{-(\mathbf{v}_t - \alpha_t \mathbf{v}_0)}{\sigma_t^2}$.

Thus, we can leverage the corresponding empirical loss (11):

$$\widehat{\mathbf{s}} \in \arg \min_{\mathbf{s} \in \mathcal{T}} \widehat{\mathcal{L}}(\mathbf{s}), where \ \widehat{\mathcal{L}}(\mathbf{s}) = \frac{1}{n} \sum_{i=1}^{n} \ell(\mathbf{x}^{(i)}, \mathbf{y}^{(i)}; \mathbf{s}),$$

where the loss function is defined in (12)

$$\ell(\mathbf{x}^{(i)}, \mathbf{y}^{(i)}; \mathbf{s}) = \int_{t_0}^{T} \mathbb{E}_{\mathbf{v}_t | \mathbf{v}_0 = \mathbf{x}^{(i)}} \left[ \|\mathbf{s}(\mathbf{v}_t, \mathbf{y}, t) - \nabla_{\mathbf{v}_t} \log \phi_t(\mathbf{v}_t | \mathbf{v}_0) \|_2^2 \right] dt.$$

## E.1   Steps for Proving Corollary 2

### E.1.1   Risk Decomposition

The proof procedure is analogous to the proof of Theorem 4.1 in [Fu et al., 2024c], provided in Appendix D of the same work. Our goal is to derive a bound on $\mathbb{E}_{\{(\mathbf{x}^{(i)}, \mathbf{y}^{(i)})\}_{i=1}^{n}} [\mathcal{R}(\widehat{\mathbf{s}})]$. We denote the ground truth score function as $\mathbf{s}^*$ and set $\mathcal{R}(\mathbf{s}^*) = 0$.

Following the setup, the risk can be decomposed as:

$$\mathcal{R}(\widehat{\mathbf{s}}) = \mathcal{R}(\widehat{\mathbf{s}}) - \mathcal{R}(\mathbf{s}^\star) = \mathcal{L}(\widehat{\mathbf{s}}) - \mathcal{L}(\mathbf{s}^\star),$$

where $\widehat{\mathbf{s}}$ is the score function trained on dataset $\mathcal{D}^{(\backslash)}$ using the empirical risk. By creating $n$ i.i.d. ghost samples

$$(\mathcal{D}')^{(n)} = \{(\mathbf{x}^{(i')}, \mathbf{y}^{(i')})\}_{i'=1}^{n} \sim \mathcal{P}_{\mathbf{x}_{\mathrm{miss}}, \mathbf{x}_{\mathrm{obs}}},$$

the population risk of $\widehat{\mathbf{s}}$ can be rewritten as:

$$\mathcal{R}(\widehat{\mathbf{s}}) - \mathcal{R}(\mathbf{s}^\star) = \mathbb{E}_{(\mathcal{D}')^{(n)}} \left[ \frac{1}{n} \sum_{i=1}^{n} \left( \ell(\mathbf{x}^{(i)}, \mathbf{y}^{(i)}; \widehat{\mathbf{s}}) - \ell(\mathbf{x}^{(i)}, \mathbf{y}^{(i)}; \mathbf{s}^\star) \right) \right].$$

To bound the rewritten population risk, we can further decompose it by analyzing its behavior in a truncated area (aligning with our score approximation analysis in Theorem 1, we analyze the error conditioning on the event $\mathcal{C}_\delta = \{\|\mathbf{x}\|_2, \|\mathbf{v}\|_2 \le C_{\mathrm{data}}^\delta\}$), and the error induced by truncation.

The truncated loss function is defined as

$$\ell^{\mathrm{trunc}}(\mathbf{x}, \mathbf{y}; \mathbf{s}) = \int_{t_0}^{T} \mathbb{E}_{\mathbf{v}_t | \mathbf{v}_0 = \mathbf{x}} \left[ \|\mathbf{s}(\mathbf{v}_t, \mathbf{y}, t) - \nabla_{\mathbf{v}_t} \log \phi_t(\mathbf{v}_t | \mathbf{v}_0) \|_2^2 \mathbb{1}_{\mathcal{C}_\delta} \right] dt. \tag{27}$$

Accordingly, we denote the truncated domain of the score function by $\mathcal{X} = [-C_{\mathrm{data}}^\delta, C_{\mathrm{data}}^\delta]_{d|\mathcal{I}_{\mathrm{miss}}|} \times [-C_{\mathrm{data}}^\delta, C_{\mathrm{data}}^\delta]_{d|\mathcal{I}_{\mathrm{obs}}|}$, and the truncated loss function class defined as

$$\mathcal{S}(C_{\mathrm{data}}^\delta) = \left\{ \ell^{\mathrm{trunc}}(\cdot, \cdot, \mathbf{s}) : \mathcal{X} \to \mathbb{R} | \mathbf{s} \in \mathcal{T} \right\}. \tag{28}$$

Define the following intermediate terms ($\widehat{\mathbf{s}}$ depends on $\mathcal{D}^{(\backslash)}$):

$$\mathcal{L}_1 = \frac{1}{n} \sum_{i=1}^{n} \left( \ell(\mathbf{x}^{(i)}, \mathbf{y}^{(i)}; \widehat{\mathbf{s}}) - \ell(\mathbf{x}^{(i)}, \mathbf{y}^{(i)}; \mathbf{s}^\star) \right),$$

$$\mathcal{L}_1^{\text{trunc}} = \frac{1}{n} \sum_{i=1}^{n} \left( \ell^{\text{trunc}}(\mathbf{x}^{(i)}, \mathbf{y}^{(i)}; \widehat{\mathbf{s}}) - \ell^{\text{trunc}}(\mathbf{x}^{(i)}, \mathbf{y}^{(i)}; \mathbf{s}^\star) \right),$$

$$\mathcal{L}_2 = \frac{1}{n} \sum_{i=1}^{n} \left( \ell(\mathbf{x}'^{(i)}; \mathbf{y}'^{(i)}, \widehat{\mathbf{s}}) - \ell(\mathbf{x}'^{(i)}, \mathbf{y}'^{(i)}; \mathbf{s}^\star) \right),$$

and

$$\mathcal{L}_2^{\text{trunc}} = \frac{1}{n} \sum_{i=1}^{n} \left( \ell^{\text{trunc}}(\mathbf{x}'^{(i)}, \mathbf{y}'^{(i)}; \widehat{\mathbf{s}}) - \ell^{\text{trunc}}(\mathbf{x}'^{(i)}, \mathbf{y}'^{(i)}; \mathbf{s}^\star) \right).$$

The decomposition for the expected empirical risk over $\mathcal{D}^{(\backslash)}$ then becomes:

$$\mathbb{E}_{\mathcal{D}^{(n)}}[\mathcal{R}(\widehat{\mathbf{s}})] = \underbrace{\mathbb{E}_{\mathcal{D}^{(n)}}\left[ \mathbb{E}_{(\mathcal{D}')^{(n)}}\left[ \mathcal{L}_2 - \mathcal{L}_2^{\text{trunc}} \right] \right] + \mathbb{E}_{\mathcal{D}^{(n)}}\left[ \mathcal{L}_1^{\text{trunc}} - \mathcal{L}_1 \right]}_{A}$$

$$+ \underbrace{\mathbb{E}_{\mathcal{D}^{(n)}}\left[ \mathbb{E}_{(\mathcal{D}')^{(n)}}\left[ \mathcal{L}_2^{\text{trunc}} \right] - \mathcal{L}_1^{\text{trunc}} \right]}_{B} + \underbrace{\mathbb{E}_{\mathcal{D}^{(n)}}\left[ \mathcal{L}_1 \right]}_{C}. \tag{29}$$

The terms $A$, $B$, and $C$ respectively represent the error incurred due to truncation, approximation in truncation, and the in-sample empirical risk expectation.

### E.1.2 Bound of Each Component

We first bound the data range with high probability. The proof of the lemmas stated in this section are deferred to E.3.

**Lemma 11** (Range of the data). *Given a sufficiently large data truncation range $C_{\text{data}}^\delta > 0$, we have*

$$\mathbb{P}\left( \cup_{i=1}^{n} \left\{ \|\mathbf{x}^{(i)}\|_2, \|\mathbf{v}^{(i)}\|_2 \leq C_{\text{data}}^\delta \right\} \right) \geq 1 - \delta_{n,d},$$

*where $\delta_{n,d} = 2n \exp\left\{ \frac{-C_2 C_{\text{data}}^2}{8(Hd+1)(\|\mathbf{\Lambda}\|_{\text{F}}+1)} \right\}$, and $C_2$ is the absolute constant defined in Lemma 16.*

*We also have $\mathbb{P}[E] \geq 1 - \delta_d$, where $\delta_d = \frac{1}{n} \delta_{n,d}$.*

In the following analysis, for the sake of simplicity, we denote $C_\Sigma = 1 + \frac{\|\mathbf{\Gamma}_{\text{cor}}\|_2 \kappa(\mathbf{\Lambda})}{\lambda_{\min}(\mathbf{\Gamma}_{\text{obs}})}$, which origins from Lemma 4. Then we state Lemma 12 to bound term $A$ in (29), which is the counterpart of $(D.12)$ in [Fu et al., 2024c].

**Lemma 12.** *For any $\mathbf{s} \in \mathcal{T}$,*

$$\mathbb{E}_{\mathbf{x}, \mathbf{y}}\left[ |\ell(\mathbf{x}, \mathbf{y}; \mathbf{s}) - \ell^{\text{trunc}}(\mathbf{x}, \mathbf{y}; \mathbf{s})| \right] \lesssim \sqrt{\delta_d}\left[ \left( C_\Sigma C_{\text{data}}^\delta \right)^2 + Hd \right] \left( T + \frac{1}{t_0} \right).$$

It is straightforward to conclude that

$$A \lesssim \sqrt{\delta_d}\left[ \left( C_\Sigma C_{\text{data}}^\delta \right)^2 + Hd \right] \left( T + \frac{1}{t_0} \right). \tag{30}$$

Then we proceed to the term $C$ in (29). For any $\mathbf{s} \in \mathcal{T}$, we have the following relationship

$$
\begin{aligned}
C &= \mathbb{E}_{\mathcal{D}^{(n)}}[\mathcal{L}_1] \\
&= \mathbb{E}_{\mathcal{D}^{(n)}}\left[ \frac{1}{n} \sum_{i=1}^{n} \left( \ell(\mathbf{x}^{(i)}, \mathbf{y}^{(i)}; \widehat{\mathbf{s}}) - \ell(\mathbf{x}^{(i)}, \mathbf{y}^{(i)}; \mathbf{s}^\star) \right) \right] \\
&\leq \mathbb{E}_{\mathcal{D}^{(n)}}\left[ \frac{1}{n} \sum_{i=1}^{n} \left( \ell(\mathbf{x}^{(i)}, \mathbf{y}^{(i)}; \mathbf{s}) - \ell(\mathbf{x}^{(i)}, \mathbf{y}^{(i)}; \mathbf{s}^\star) \right) \right] \\
&\leq \mathcal{R}(\mathbf{s}),
\end{aligned}
$$

the inequality holds due to $\widehat{\mathbf{s}}$ minimizes $\widehat{\mathcal{L}}$.

Taking minimum $w.r.t.$ $\mathbf{s} \in \mathcal{T}$, we have

$$C \leq \min_{\mathbf{s} \in \mathcal{T}} \mathcal{R}(\mathbf{s}) = \min_{\mathbf{s} \in \mathcal{T}} \int_{t_0}^{T} \mathbb{E}_{\mathbf{v}_t, \mathbf{y}} \left[ \|\mathbf{s}(\mathbf{v}_t, \mathbf{y}, t) - \nabla_{\mathbf{v}_t} \log p_t(\mathbf{v}_t|\mathbf{y})\|_2^2 \right] dt. \tag{31}$$

**Lemma 13.** *Given an error level $\epsilon \in (0, 1)$,*

$$\min_{\mathbf{s} \in \mathcal{T}} \mathcal{R}(\mathbf{s}) \lesssim \epsilon^2 \left( T + \log\left(\frac{1}{t_0}\right) \right) + \sqrt{\delta_d} \left[ (C_\Sigma C_{\text{data}}^\delta)^2 + Hd \right] \left( T + \frac{1}{t_0} \right).$$

Finally, we can proceed to the bound of term $B$. In an attempt to providing the bound, we first need to calculate the covering number of the loss function class $\mathcal{S}(C_{\text{data}}^\delta)$, and correspondingly, the covering number of our transformer architecture function class. The covering number is defined as follows:

**Definition 2.** *We denote $\mathcal{N}(\delta, \mathcal{F}, \|\cdot\|)$ to be the $\delta-$covering number of any function class $\mathcal{F}$ w.r.t the norm $\|\cdot\|$, i.e.,*

$$\mathcal{N}(\delta; \mathcal{F}, \|\cdot\|) = \arg\min_H \{\exists \{f_i\}_{i=1}^N \subseteq \mathcal{F}, s.t. \ \forall f \in \mathcal{F}, \exists i \in [N], \|f_i - f\| \leq \delta \}.$$

A modified version of Lemma 23 in [Fu et al., 2024b] provides the following result on transformer covering numbers:

**Lemma 14.** *Consider the entire transformer architecture $\mathcal{F} = \mathcal{T}(D, L, M, B, R)$ (i.e. with encoder and decoder). If the input to the transformer satisfy $\|\mathbf{v}_t\|_2, \|\mathbf{y}\|_2 \leq C_{\text{data}}^\delta$, the time embedding $\mathbf{e}$ and the diffusion time-step embedding $\phi(t)$ satisfy $\|\mathbf{e}\|_2 = r, \|\phi(t)\|_2 \leq C_{\text{diff}}$ and $r, C_{\text{diff}} \leq \mathcal{O}(\sqrt{Hd})$, then the log-covering number of the transformer architecture is bounded by*

$$\log \mathcal{N}(\delta_s; \mathcal{F}, \|\cdot\|_{F,\infty}) \lesssim D^2 M \left( L^2 \log\left(BMNRC_{\text{data}}^\delta C_\Sigma\right) + \log\left(\frac{BMLHdC_{\text{data}}^\delta C_\Sigma}{\delta_s}\right) \right).$$

Then, we can leverage the following lemma to calculate the covering number of the corresponding truncated loss function class.

**Lemma 15.** *Suppose $\widehat{\mathbf{s}}^{(1)}, \widehat{\mathbf{s}}^{(2)} \in \mathcal{T}(D, L, M, B, R)$ such that $\|\widehat{\mathbf{s}}^{(1)}(\mathbf{v}_t, \mathbf{y}; t) - \widehat{\mathbf{s}}^{(2)}(\mathbf{v}_t, \mathbf{y}; t)\|_2 \leq \delta_s$ for any $\|\mathbf{v}_t\|_2, \|\mathbf{y}\|_2, \|\mathbf{x}\|_2 \leq C_{\text{data}}^\delta$ and $t \geq t_0$, then we have*

$$|\ell^{trunc}(\widehat{\mathbf{s}}^{(1)}) - \ell^{trunc}(\widehat{\mathbf{s}}^{(2)})| \leq 4\delta_s \left( T + \log\left(\frac{1}{t_0}\right) \right) (C_\Sigma C_{\text{data}}^\delta + \sqrt{Hd}).$$

Equipped with this lemma, it is straight forward to derive that

$$\log \mathcal{N}(\delta_l; \mathcal{S}(C_{\text{data}}^\delta), \|\cdot\|_\infty) \lesssim D^2 M \left( L^2 \log\left(BMNRC_{\text{data}}^\delta C_\Sigma\right) + \log\left(\frac{BMLHdC_{\text{data}}^\delta C_\Sigma}{\delta_s}\right) \right),$$

and $\delta_s$ satisfies

$$\delta_l = 4\delta_s \left( T + \log\left(\frac{1}{t_0}\right) \right) (C_\Sigma C_{\text{data}}^\delta + \sqrt{Hd}).$$

Invoking the bound provided in (D.16) of [Fu et al., 2024c], we have

$$B \lesssim C + A + \frac{1}{n} \log \mathcal{N}(\delta_l; \mathcal{S}(C_{\text{data}}^\delta), \|\cdot\|_\infty) \int_{t_0}^T \sigma_t^{-4} \mathrm{d}t + 7\delta_l$$

$$\lesssim C + A + \frac{(T + \frac{1}{t_0})}{n} D^2 M \left( L^2 \log\left(BMNRC_{\text{data}}^\delta C_\Sigma\right) + \log\left(\frac{BMLHdC_{\text{data}}^\delta C_\Sigma}{\delta_s}\right) \right)$$

$$+ 28\delta_s \left( T + \log\left(\frac{1}{t_0}\right) \right) (C_\Sigma C_{\text{data}}^\delta + \sqrt{Hd}). \tag{32}$$

Combining the bound of A, B and C ((30),(32), (31)), we can leverage the empirical risk decomposition (29) to finalize the proof of Corollary 2.

### E.2 Proof of Corollary 2

*Proof of Corollary 2.* By (29), (30),(32) and (31), we have

$$\mathbb{E}_{\mathcal{D}^{(n)}}[\mathcal{R}(\widehat{\mathbf{s}})] \leq A + B + C$$

$$\lesssim 2\sqrt{\delta_d} \left[ (C_\Sigma C_{\text{data}}^\delta)^2 + Hd \right] \left( T + \frac{1}{t_0} \right)$$

$$+ \frac{(T + \frac{1}{t_0})}{n} D^2 M \left( L^2 \log\left(BMNRC_{\text{data}}^\delta C_\Sigma\right) + \log\left(\frac{BMLHdC_{\text{data}}^\delta C_\Sigma}{\delta_s}\right) \right)$$

$$+ 28\delta_s \left( T + \log\left(\frac{1}{t_0}\right) \right) (C_\Sigma C_{\text{data}}^\delta + \sqrt{Hd}) \quad + \epsilon^2 \left( T + \log\left(\frac{1}{t_0}\right) \right).$$

Plugging in the configuration of our transformer architecture in Theorem 1, and take

$$C_{\text{data}}^{\delta} = \mathcal{O}(\sqrt{Hd}\kappa(\mathbf{\Lambda})\kappa(\mathbf{\Gamma}_{\text{obs}})\|\mathbf{\Gamma}_{\text{cor}}\|_2 \log(Hdn)), \epsilon = \frac{1}{\sqrt{n}}, \delta_s = \frac{1}{nC_{\text{data}}^{\delta}C_{\Sigma}}, T = \mathcal{O}(\log(n)),$$

the inequality above gives rise to

$$\mathbb{E}_{\mathcal{D}^{(n)}}[\mathcal{R}(\widehat{\mathbf{s}})] \lesssim \frac{Hd^2\kappa_{t_0}^4\kappa^2(\mathbf{\Lambda})\kappa^2(\mathbf{\Gamma}_{\text{obs}})t_0^{-1}}{n} \log(Hd\kappa(\mathbf{\Lambda})\kappa(\mathbf{\Gamma}_{\text{obs}})nt_0^{-1}).$$

$\square$

## E.3 Proof of Supporting Lemmas in E.1.2

*Proof of Lemma 11.* We first state the polynomial concentration lemma for Gaussian random variables.

**Lemma 16** (Lemma 24 in [Fu et al., 2024b]). *Let $g$ be a polynomial of degree $p$ and $x \sim \mathcal{N}(0, I_d)$. Then there exists an absolute positive constant $C_p$, depending only on $p$, such that for any $\delta < 1$,*

$$\mathbb{P}\left[|g(x) - \mathbb{E}[g(x)]| \geq \delta\sqrt{\text{Var}(g(x))}\right] \leq 2\exp\left(-C_p\delta^{2/p}\right).$$

For a random variable $\mathbf{r} \sim \mathcal{N}(\mathbf{0}, \mathbf{\Sigma}_0)$, consider $g(\cdot) = \|\cdot\|_2^2$, we have

$$\mathbb{E}[g(\mathbf{u})] = \text{tr}(\mathbf{\Sigma}_0), \mathbb{E}[g(\mathbf{u})^2] \leq 3\|\mathbf{\Sigma}_0\|_{\text{F}}^2.$$

Applying Lemma 16, we can conclude that with high probability at least $1 - 2\exp(-C_2\delta)$,

$$|\|\mathbf{u}\|_2^2 - \mathbb{E}[\|\mathbf{u}\|_2^2]| \leq \delta\sqrt{\text{Var}(\|g(\mathbf{u})\|_2^2)} \leq \sqrt{3}\delta\|\mathbf{\Sigma}_0\|_{\text{F}}.$$

Considering $\mathbf{v}_t$, we have $\mathbf{\Sigma}_1 = \alpha_t^2(\mathbf{\Gamma}_{\text{miss}} \otimes \mathbf{\Lambda}) + \sigma_t^2\mathbf{I}$; and for $\mathbf{x}$, we have $\mathbf{\Sigma}_2 = \mathbf{\Gamma} \otimes \mathbf{\Lambda}$. Therefore,

$$\begin{aligned}
\|\mathbf{v}^{(m)}\|_2 &\leq \sqrt{\text{tr}(\mathbf{\Sigma}_1) + \sqrt{3}\delta\|\mathbf{\Sigma}_1\|_{\text{F}}} \\
&\leq \sqrt{\text{tr}(\mathbf{\Gamma}_{\text{miss}})\text{tr}(\mathbf{\Lambda}) + d|\mathcal{I}_{\text{miss}}| + \sqrt{3}\delta(\|\mathbf{\Gamma}_{\text{miss}}\|_{\text{F}}\|\mathbf{\Lambda}\|_{\text{F}} + d|\mathcal{I}_{\text{miss}}|)} \\
&\leq \sqrt{d\|\mathbf{\Lambda}\|_{\text{F}} + d|\mathcal{I}_{\text{miss}}| + \sqrt{3}\delta(\|\mathbf{\Gamma}_{\text{miss}}\|_{\text{F}}\|\mathbf{\Lambda}\|_{\text{F}} + d|\mathcal{I}_{\text{miss}}|)} \\
&\leq \sqrt{(Hd + 1)(\|\mathbf{\Lambda}\|_{\text{F}} + 1)}(1 + \sqrt{2}\delta),
\end{aligned}$$

the last inequality holds for $\delta < 1$. Similar inequalities hold for $\|\mathbf{x}^{(i)}\|_2$.

Consider $C_{\text{data}}^{\delta} \geq 2\sqrt{(Hd+1)(\|\mathbf{\Lambda}\|_{\text{F}}+1)}$ and let $\delta = \frac{(C_{\text{data}}^{\delta})^2}{8Hd+1)(\|\mathbf{\Lambda}\|_{\text{F}}+1)}$. We can then obtain a union bound. With probability at least $1 - 2n\exp\left\{\frac{-C_2(C_{\text{data}})^2}{8(Hd+1)(\|\mathbf{\Lambda}\|_{\text{F}}+1)}\right\}$,

$$\max\{\|\mathbf{x}^{(i)}\|_2, \|\mathbf{v}^{(m)}\|_2\}_{i=1}^n \leq C_{\text{data}}^{\delta}.$$

We finish the proof by setting $\delta_{n,d} = 2n\exp\left\{\frac{-C_2C_{\text{data}}^2}{8Hd+1)(\|\mathbf{\Lambda}\|_{\text{F}}+1)}\right\}$. $\square$

*Proof of Lemma 12.* For any $\mathbf{s} \in \mathcal{T}$ ($\mathbf{s}$ can depend on $\mathbf{x}, \mathbf{y}$),

$$\mathbb{E}_{\mathbf{x},\mathbf{y}}\big[|\ell(\mathbf{x},\mathbf{y};\mathbf{s}) - \ell^{\text{trunc}}(\mathbf{x},\mathbf{y};\mathbf{s})|\big]$$

$$= \int_{t_0}^{T} \int_{\mathbf{x},\mathbf{y}} \mathbb{E}_{\mathbf{v}_t|\mathbf{v}_0=\mathbf{x}}\big[\|\mathbf{s}(\mathbf{v},\mathbf{y},t) - \nabla \log \phi_t(\mathbf{v}_t|\mathbf{v}_0)\|_2^2 \mathbb{1}_{\mathcal{C}_\delta}\big] p(\mathbf{x},\mathbf{y})\mathbf{dxdy}dt$$

$$\leq 2\int_{t_0}^{T} \int_{\mathbf{x},\mathbf{y}} \mathbb{E}_{\mathbf{v}_t|\mathbf{v}_0=\mathbf{x}}\left[\left(\|\mathbf{s}(\mathbf{v},\mathbf{y},t)\|_2^2 + \left\|\frac{\mathbf{v}_t - \alpha_t\mathbf{v}_0}{\sigma_t^2}\right\|_2^2\right)\mathbb{1}_{\mathcal{C}_\delta}\right] p(\mathbf{x},\mathbf{y})\mathbf{dxdy}dt$$

$$\lesssim \int_{t_0}^{T} \mathbb{P}(\mathcal{C}_\delta)R_t^2 dt + \int_{t_0}^{T} \mathbb{E}_{\mathbf{v}_t,\mathbf{x},\mathbf{y},\mathbf{z}\sim\mathcal{N}(\mathbf{0},\mathbf{I}_{d|\mathcal{I}_{\text{miss}}|})}\left[\left\|\frac{\sigma_t\mathbf{z}}{\sigma_t^2}\right\|_2^2 \mathbb{1}_{\mathcal{C}_\delta}\right]dt$$

$$\leq \delta_d (C_\Sigma C_{\text{data}}^\delta)^2 \int_{t_0}^{T} \sigma_t^{-4} dt + \int_{t_0}^{T} \mathbb{P}^{1/2}[\mathcal{C}_\delta]\mathbb{E}_{\mathbf{z}\sim\mathcal{N}(\mathbf{0},\mathbf{I}_{d|\mathcal{I}_{\text{miss}}|})}\left[\left\|\frac{\mathbf{z}}{\sigma_t}\right\|_2^2\right]dt$$

$$\leq \delta_d (C_\Sigma C_{\text{data}}^\delta)^2 \int_{t_0}^{T} \sigma_t^{-4} dt + \sqrt{\delta_d}\int_{t_0}^{T} \sigma_t^{-2} H d\, dt$$

$$\leq \sqrt{\delta_d}\big[(C_\Sigma C_{\text{data}}^\delta)^2 + Hd\big]\int_{t_0}^{T} \sigma_t^{-4} dt$$

$$= \sqrt{\delta_d}\big[(C_\Sigma C_{\text{data}}^\delta)^2 + Hd\big]\int_{t_0}^{T} \left(\frac{e^t}{e^t - 1}\right)^2 dt$$

$$\lesssim \sqrt{\delta_d}\big[(C_\Sigma C_{\text{data}}^\delta)^2 + Hd\big]\left(T + \frac{1}{t_0}\right),$$

where $R_t$ is the truncation range of the decoder, and we apply triangular inequality in the third line. $\qquad\square$

*Proof of Lemma 13.* Since $\widetilde{\mathbf{s}} \in \mathcal{T}$, we can invoke Theorem 1 and triangular inequality:

$$\min_{\mathbf{s}\in\mathcal{T}} \mathcal{R}(\mathbf{s}) = \min_{\mathbf{s}\in\mathcal{T}} \int_{t_0}^{T} \mathbb{E}_{\mathbf{v}_t,\mathbf{y}}\big[\|\mathbf{s}(\mathbf{v}_t,\mathbf{y},t) - \nabla_{\mathbf{v}_t}\log p_t(\mathbf{v}_t|\mathbf{y})\|_2^2\big]dt$$

$$\leq \int_{t_0}^{T} \mathbb{E}_{\mathbf{v}_t,\mathbf{y}}\big[\|\widetilde{\mathbf{s}}(\mathbf{v}_t,\mathbf{y},t) - \nabla_{\mathbf{v}_t}\log p_t(\mathbf{v}_t|\mathbf{y})\|_2^2 \mathbb{1}_{\mathcal{C}_\delta}\big]dt$$

$$\qquad + 2\int_{t_0}^{T} \mathbb{E}_{\mathbf{v}_t,\mathbf{y}}\big[(\|\widetilde{\mathbf{s}}(\mathbf{v}_t,\mathbf{y},t)\|_2^2 + \|\nabla_{\mathbf{v}_t}\log p_t(\mathbf{v}_t|\mathbf{y})\|_2^2)\mathbb{1}_{\mathcal{C}_\delta}\big]dt.$$

$$\leq \epsilon^2 \int_{t_0}^{T} \sigma_t^{-2} dt + 2\delta_d \int_{t_0}^{T} \|\widetilde{\mathbf{s}}(\mathbf{v}_t,\mathbf{y},t)\|_2^2 dt$$

$$\qquad + 2\sqrt{\delta_d}\int_{t_0}^{T} \mathbb{E}_{\mathbf{v}_t,\mathbf{y}}^{1/2}\big[\|\nabla_{\mathbf{v}_t}\log p_t(\mathbf{v}_t|\mathbf{y})\|_2^4\big]dt$$

$$\leq \epsilon^2 \int_{t_0}^{T} \sigma_t^{-2} dt + 2\delta_d \int_{t_0}^{T} \|\widetilde{\mathbf{s}}(\mathbf{v}_t,\mathbf{y},t)\|_2^2 dt$$

$$\qquad + 2\sqrt{\delta_d}\int_{t_0}^{T} \mathbb{E}_{\mathbf{v}_t,\mathbf{y}}\big[\|\nabla_{\mathbf{v}_t}\log p_t(\mathbf{v}_t|\mathbf{y})\|_2^2\big]dt,$$

where we apply Cauchy-Schwarz inequality in the second step, Jensen's inequality in the last step. For the second last term, similar to the proof of Lemma 12, we have

$$\int_{t_0}^{T} \mathbb{E}_{\mathbf{v}_t,\mathbf{y}}\big[\|\widetilde{\mathbf{s}}(\mathbf{v}_t,\mathbf{y},t)\|_2^2\big]dt \leq (C_{\text{data}}^\delta C_\Sigma)^2 \int_{t_0}^{T} \sigma_t^{-4} dt.$$

For the last term, we have

$$\int_{t_0}^{T} \mathbb{E}_{\mathbf{v}_t,\mathbf{y}} \left[ \|\nabla_{\mathbf{v}_t} \log p_t(\mathbf{v}_t|\mathbf{y})\|_2^2 \right] dt$$

$$\leq \int_{t_0}^{T} \sigma_t^{-4} \mathbb{E}_{\mathbf{v}_t,\mathbf{y}} \left[ \|\mathbf{v}_t - \alpha_t \boldsymbol{\Sigma}_{\mathrm{cor}} \boldsymbol{\Sigma}_{\mathrm{obs}}^{-1} \mathbf{y}\|_2^2 \right] dt$$

$$\leq 2 \int_{t_0}^{T} \sigma_t^{-4} \mathbb{E}_{\mathbf{x},\mathbf{y},\mathbf{z}\sim\mathcal{N}(0,\mathbf{I}_{d|\mathcal{I}_{\mathrm{miss}}|})} \left[ \|\mathbf{x} - \boldsymbol{\Sigma}_{\mathrm{cor}} \boldsymbol{\Sigma}_{\mathrm{obs}}^{-1} \mathbf{y}\|_2^2 + \|\sigma_t \mathbf{z}\|_2^2 \right] dt$$

$$\leq 2 \int_{t_0}^{T} \sigma_t^{-4} \left[ \mathbb{E}_{\mathbf{u}\sim\mathcal{N}(0,\boldsymbol{\Sigma}_{\mathrm{cond}})} \left[ \|\mathbf{u}\|_2^2 \right] + \sigma_t^2 Hd \right] dt$$

$$= 2 \left[ \sigma_t^2 Hd + \mathrm{tr}(\boldsymbol{\Sigma}_{\mathrm{miss}} - \boldsymbol{\Sigma}_{\mathrm{cor}}^{\top} \boldsymbol{\Sigma}_{\mathrm{obs}}^{-1} \boldsymbol{\Sigma}_{\mathrm{cor}}) \right] \int_{t_0}^{T} \sigma_t^{-4} dt$$

$$\leq 2 \left[ \sigma_t^2 Hd + \mathrm{tr}(\boldsymbol{\Gamma}_{\mathrm{miss}}) \mathrm{tr}(\boldsymbol{\Lambda}) \right] \int_{t_0}^{T} \sigma_t^{-4} dt,$$

where we utilize the positive definiteness of $\boldsymbol{\Sigma}_{\mathrm{cor}}^{\top} \boldsymbol{\Sigma}_{\mathrm{obs}}^{-1} \boldsymbol{\Sigma}_{\mathrm{cor}}$ in the last inequality.
Combining all the terms together, we have

$$\min_{\mathbf{s}\in\mathcal{T}} \mathcal{R}(\mathbf{s}) \leq \epsilon^2 \int_{t_0}^{T} \sigma_t^{-2} dt + 2\sqrt{\delta_d} \int_{t_0}^{T} \mathbb{E}_{\mathbf{v}_t,\mathbf{y}} \left[ \|\widetilde{\mathbf{s}}(\mathbf{v}_t,\mathbf{y},t)\|_2^2 + \|\nabla_{\mathbf{v}_t} \log p_t(\mathbf{v}_t|\mathbf{y})\|_2^2 \right] dt$$

$$\lesssim \epsilon^2 \left( T + \log\left(\frac{1}{t_0}\right) \right) + \sqrt{\delta_d} \left[ (C_{\Sigma} C_{\mathrm{data}}^{\delta})^2 + Hd \right] \left( T + \frac{1}{t_0} \right).$$

$\square$

*Proof of Lemma 15.* We have

$$\left| \mathbb{E}_{\mathbf{v}_t|\mathbf{v}_0=\mathbf{x}} \left[ \|\widehat{\mathbf{s}}^{(1)}(\mathbf{v}_t,\mathbf{y};t) - \nabla \log \phi_t(\mathbf{v}_t|\mathbf{v}_0)\|_2^2 - \|\widehat{\mathbf{s}}^{(2)}(\mathbf{v}_t,\mathbf{y};t) - \nabla \log \phi_t(\mathbf{v}_t|\mathbf{v}_0)\|_2^2 \right] \right|$$

$$= \left| \mathbb{E}_{\mathbf{z}\sim\mathcal{N}(0,\mathbf{I}_{d|\mathcal{I}_{\mathrm{miss}}|})} \left[ \|\widehat{\mathbf{s}}^{(1)}(\alpha_t\mathbf{x} + \sigma_t\mathbf{z},\mathbf{y};t) + \frac{\mathbf{z}}{\sigma_t}\|_2^2 - \|\widehat{\mathbf{s}}^{(2)}(\alpha_t\mathbf{x} + \sigma_t\mathbf{z},\mathbf{y};t) + \frac{\mathbf{z}}{\sigma_t}\|_2^2 \right] \right|$$

$$= \left| \mathbb{E}_{\mathbf{z}\sim\mathcal{N}(0,\mathbf{I}_{d|\mathcal{I}_{\mathrm{miss}}|})} \left[ \left( (\widehat{\mathbf{s}}^{(1)} - \widehat{\mathbf{s}}^{(2)})(\alpha_t\mathbf{x} + \sigma_t\mathbf{z},\mathbf{y};t) \right)^{\top} \left( (\widehat{\mathbf{s}}^{(1)} + \widehat{\mathbf{s}}^{(2)})(\alpha_t\mathbf{x} + \sigma_t\mathbf{z},\mathbf{y};t) + \frac{2\mathbf{z}}{\sigma_t} \right) \right] \right|$$

$$\leq \left| \mathbb{E}_{\mathbf{z}\sim\mathcal{N}(0,\mathbf{I}_{d|\mathcal{I}_{\mathrm{miss}}|})} \left[ \left\| (\widehat{\mathbf{s}}^{(1)} - \widehat{\mathbf{s}}^{(2)})(\alpha_t\mathbf{x} + \sigma_t\mathbf{z},\mathbf{y};t) \right\|_2 \left\| (\widehat{\mathbf{s}}^{(1)} + \widehat{\mathbf{s}}^{(2)})(\alpha_t\mathbf{x} + \sigma_t\mathbf{z},\mathbf{y};t) + \frac{2\mathbf{z}}{\sigma_t} \right\|_2 \right] \right|$$

$$\leq 2\delta_s \mathbb{E}_{\mathbf{z}\sim\mathcal{N}(0,\mathbf{I}_{d|\mathcal{I}_{\mathrm{miss}}|})} \left[ \left\| (\widehat{\mathbf{s}}^{(1)} + \widehat{\mathbf{s}}^{(2)})(\alpha_t\mathbf{x} + \sigma_t\mathbf{z},\mathbf{y};t) \right\|_2 + \left\| \frac{2\mathbf{z}}{\sigma_t} \right\|_2 \right]$$

$$\leq 2\delta_s \left[ 2\|\widehat{\mathbf{s}}\|_2 + 2\sigma_t^{-1} \mathbb{E}_{\mathbf{z}\sim\mathcal{N}(0,\mathbf{I}_{d|\mathcal{I}_{\mathrm{miss}}|})} \left[ \|\mathbf{z}\|_2 \right] \right]$$

$$\leq 4\delta_s \sigma_t^{-2} (C_{\Sigma} C_{\mathrm{data}}^{\delta} + \sqrt{Hd}).$$

Therefore, we obtain

$$|\ell^{\mathrm{trunc}}(\widehat{\mathbf{s}}^{(1)}) - \ell^{\mathrm{trunc}}(\widehat{\mathbf{s}}^{(2)})| \leq 4\delta_s (C_{\Sigma} C_{\mathrm{data}}^{\delta} + \sqrt{Hd}) \int_{t_0}^{T} \sigma_t^{-2} dt$$

$$\leq 4\delta_s \left( T + \log\left(\frac{1}{t_0}\right) \right) (C_{\Sigma} C_{\mathrm{data}}^{\delta} + \sqrt{Hd}).$$

$\square$

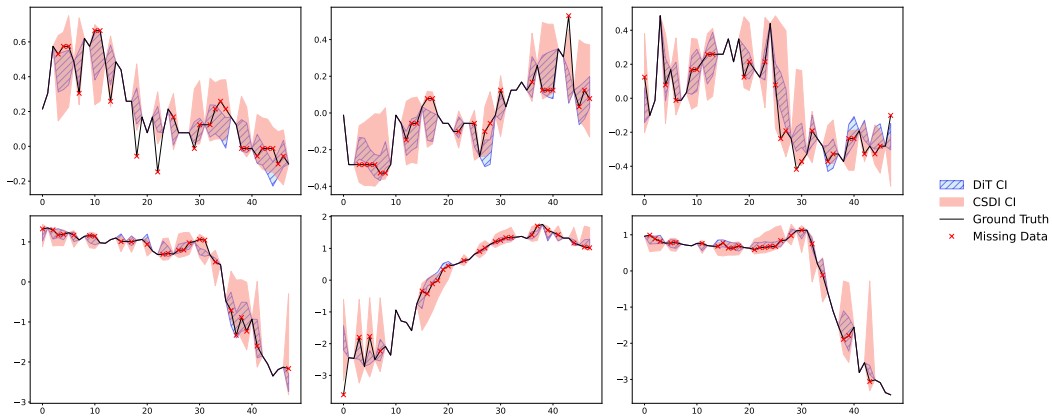

Figure 4: Comparison of imputation methods on the Electricity dataset, with 95% CR.

## F  Experiment Details

For our numerical experiments, we trained the models using a batch size of 64. Our adapted DiT model architecture used a hidden size of 256, 12 transformer layers, and 16 attention heads per layer. We utilized the PyPOTS [Du, 2023] framework to implement and handle hyperparameter tuning for the baseline methods CSDI and GP-VAE. This tuning process aimed to find the best settings and ensure the models had a comparable number of trainable parameters. Experiments were conducted on hardware consisting of an NVIDIA RTX A6000 GPU (48GB) and an Intel(R) Xeon(R) Gold 6242R CPU @ 3.10GHz. We report all the results as the average of 5 runs. Our implementation of DiT for imputation is attached in supplementary materials.

### F.1  Real World Datasets

**Dataset Descriptions.**   We utilize two real-world datasets, BeijingAir [Zhang et al., 2017] and ETT_m1, to benchmark the imputation performance of DiT. The **BeijingAir** dataset comprises hourly measurements of six air pollutants and meteorological variables collected from 12 monitoring sites in Beijing. The **ETT_m1** dataset, part of the Electricity Transformer Temperature benchmark, records clients' electricity consumption data, including power load and oil temperature. Detailed statistics for both datasets are provided in Table 5.

| Dataset | Total Samples | Sequence Length | Time Interval | Number of Variables |
|---------|---------------|-----------------|---------------|---------------------|
| Air Quality | 1168 | 30 | 1H | 132 |
| Electricity | 2321 | 48 | 15min | 7 |

Table 5: 80% of the data is used for training, and 20% for testing.

**Results.**   We report the Mean Absolute Error (MAE) in Table 6, the Mean Squared Error (MSE) in Table 7 and the Mean Relative Error (MRE) in Table 8. Results are shown across different missing data rates (10%, 20%, and 50%) for both datasets. The experimental results indicate that DiT consistently outperforms the baseline methods on both datasets, demonstrating its effectiveness, and our mixed-masking strategy can also enhance DiT's performance on real-world datasets.

Figure 4 presents a comparison of imputation results on the ETT_m1 dataset, where we randomly select samples from a 50% missing data scenario. From the plots, it is evident that although both DiT and CSDI generate CRs that largely encompass the true data points, DiT achieves a tighter bandwidth, leading to improved uncertainty quantification performance.

| Model | ETTm_1 (Missing %) | | | BeijingAir (Missing %) | | |
|---|---|---|---|---|---|---|
| | 10% | 20% | 50% | 10% | 20% | 50% |
| CSDI [Tashiro et al., 2021] | 0.1448 (±0.0105) | 0.1521 (±0.0114) | 0.1650 (±0.0097) | 0.1780 (±0.0138) | 0.1800 (±0.0129) | 0.2141 (±0.0119) |
| GP-VAE [Fortuin et al., 2020] | 0.2786 (±0.0077) | 0.3267 (±0.0044) | 0.4666 (±0.0073) | 0.4152 (±0.0088) | 0.4401 (±0.0080) | 0.5265 (±0.0054) |
| DiT | **0.1269 (±0.0076)** | **0.1377 (±0.0095)** | **0.1543 (±0.0102)** | **0.1753 (±0.0094)** | **0.1815 (±0.0208)** | **0.2057 (±0.0145)** |

Table 6: Time Series Imputation MAE Results

| Model | ETT_m1 (Missing %) | | | BeijingAir (Missing %) | | |
|---|---|---|---|---|---|---|
| | 10% | 20% | 50% | 10% | 20% | 50% |
| CSDI [Tashiro et al., 2021] | 0.0615 (±0.0097) | 0.0698 (±0.0106) | 0.0797 (±0.0106) | 0.4196 (±0.1726) | 0.3926 (±0.0790) | 0.4534 (±0.0379) |
| GP-VAE [Fortuin et al., 2020] | 0.1567 (±0.0094) | 0.2138 (±0.0067) | 0.4249 (±0.0127) | 0.4096 (±0.0202) | 0.4777 (±0.0179) | 0.7017 (±0.0189) |
| DiT | 0.0534 (±0.0063) | 0.0606 (±0.0076) | **0.0684 (±0.0070)** | 0.3683 (±0.0351) | 0.4025 (±0.0424) | 0.4255 (±0.0670) |
| **DiT w/ mixed-masking strategy** | **0.0502 (±0.0055)** | **0.0588 (±0.0081)** | 0.0711 (±0.0092) | **0.3428 (±0.0275)** | **0.3864 (±0.0403)** | **0.4229 (±0.0539)** |

Table 7: Time Series Imputation MSE Results

| Model | ETT_m1 (Missing %) | | | BeijingAir (Missing %) | | |
|---|---|---|---|---|---|---|
| | 10% | 20% | 50% | 10% | 20% | 50% |
| CSDI [Tashiro et al., 2021] | 0.1706 (±0.0123) | 0.1808 (±0.0135) | 0.1938 (±0.0114) | 0.2380 (±0.0186) | 0.2420 (±0.0174) | **0.2929 (±0.0159)** |
| GP-VAE [Fortuin et al., 2020] | 0.3285 (±0.0091) | 0.3882 (±0.0052) | 0.5478 (±0.0085) | 0.5598 (±0.0118) | 0.5917 (±0.0107) | 0.7042 (±0.0072) |
| DiT | **0.1592 (±0.0084)** | **0.1701 (±0.0102)** | **0.1825 (±0.0094)** | **0.2154 (±0.0125)** | **0.2578 (±0.0375)** | 0.3073 (±0.0241) |

Table 8: Time Series Imputation MRE Results

