# OpenReview forum: "Diffusion Transformers for Imputation: Statistical Efficiency and Uncertainty Quantification"
_NeurIPS.cc/2025/Conference — NeurIPS 2025 poster_

### Official Review · Reviewer_F7up · 2025-06-30

**Clarity:** 2
**Significance:** 2
**Originality:** 2
**Rating:** 4
**Confidence:** 2

**Summary:**

This paper studies the statistical efficiency of imputation using conditional diffusion models. It provides a theoretical analysis on the sample complexity required for accurate imputation. The main result characterizes sample complexity as a function of the number of training samples, sequence length, and the condition number of the data covariance.  In addition, it proposes a mixed masking procedure that improves the imputation performance empirically.

**Questions:**

1. In Line 146, the paper states that the training procedure assumes fully observed training data. However, in real-world scenarios, training data often contains missing values. How would the proposed framework and theoretical analysis adapt under this more realistic setting?

2. The definitions of S1–S4 in the mixed-reamsking strategy are not clearly provided. Could the authors explicitly define what each of the S1–S4 masking strategies entails, and how they are used during training?

**Ethical Concerns:**

["NO or VERY MINOR ethics concerns only"]

**Final Justification:**

I recommend a borderline acceptance of this paper.
The authors have addressed most of my concerns regarding the practicality of the assumptions made in the theorem. However, I still believe that a more general study would be beneficial. Therefore, I recommend a borderline acceptance of this paper.

**Limitations:**

yes

**Quality:**

2

**Strengths And Weaknesses:**

**Strengths**

1. The paper is well-written and presents its technical contributions in a clear and accessible manner.

2. It addresses an important and underexplored problem: understanding the statistical efficiency of imputation using diffusion models, which is increasingly relevant for structured data applications.

3. The theoretical derivations appear technically sound and novel, offering valuable insights into how sample complexity scales with factors like sequence length and data covariance.

**Weaknesses**

1. **Strong Assumptions Undermine the Practical Utility of Theoretical Results**
   Theorem 2 relies on multiple strong assumptions, including a Gaussian process prior and Assumption 1. More critically, it presumes access to a well-trained diffusion transformer that approximates the score function accurately (per Theorem 1). However, training conditional diffusion models for score estimation is a key open challenge. By assuming this is solved, the theory sidesteps a central difficulty, reducing its real-world applicability.

2. **Limited Empirical Validation of Theoretical Claims**
   While the experiments confirm that imputation performance improves with more training samples, they do not test the predicted \( O(1/\sqrt{n}) \) scaling. Additionally, dependencies on sequence length \( H \) are not empirically explored. This weakens the connection between the theoretical results and observed behavior.

3. **Clarity Issues in Mixed-Masking Strategy**
   The mixed-masking training approach is not clearly described, and the paper provides little intuition for why it should improve imputation performance. A more thorough explanation or comparison with existing masking strategies would make this contribution clearer and more convincing.

---

> ### Author Rebuttal · Authors · 2025-07-30
>
> Thank you very much for your insightful comments and suggestions. We now address the specific questions, weaknesses, and limitations you have raised.
>
> > **Q1**: Gaussian process data assumption undermines practical utility.
>
> **A1**: We respectfully disagree with the view that the Gaussian process (GP) assumption undermines the practical utility of our theoretical contributions and developed methodologies. We impose GP assumption only for theoretical analysis. The insights gained from the analysis motivate our mixed-masking strategy, which applied to real-world data well beyond the GP assumption.
>
> As discussed in the introduction, GPs are not only mathematically convenient but also widely used in real-world time series modeling, including in domains such as climate science [2] and finance. Their ability to capture rich spatio-temporal and long-horizon dependencies makes them highly representative of the core challenges in time series imputation [1]. Moreover, our theoretical analysis intentionally avoids specifying a particular kernel class (lines 119–121), thereby preserving generality across a wide range of covariance structures. This flexibility allows our theoretical findings to cover many practical scenarios beyond the synthetic GP case.
>
> Empirically, we further validate this point through experiments on latent GP data (Section 5.2) and real-world datasets (Appendix), both of which confirm the alignment between our theoretical insights and observed model performance. Also worth noting is that although our theory is developed under GP assumptions, the insights gained from this theory directly informed the development of our mixed-masking strategy. This strategy has demonstrated strong generalizability to real-world applications, clearly indicating that the theoretical assumptions do not limit the practical applicability of our method. This strategy has shown robust generalization across real-world datasets and varying missing patterns.
>
>
> > **Q2**: Experiments lack explicit testing of performance scaling with $\mathcal{O}(1/\sqrt{n})$ and dependencies on sequence length $H$.
>
> **A2**: Thank you for highlighting this. Regarding sample-size scaling ($n$), we summarize the results obtained using training strategy S1 (vanilla strategy with purely random missing) and missing pattern P4 (purely random missing) with $10^5$ training samples in Section 5.1 as follows:
>
> | Sample Size (n)      | $10^3$ | $10^{3.5}$ | $10^4$ | $10^{4.5}$ | $10^5$ |
> |----------------------|--------|------------|--------|------------|--------|
> | CR Coverage Rate (%) |  0.19  |   0.64     | 20.28  |   73.41    | 80.25  |
>
> Approximately, we observe that the difference between the achieved coverage rate and the target rate scales up to constant factors, supporting the theoretical prediction. The Python code snippet below plots the difference versus $1/\sqrt{n}$, and we also perform a linear regression, yielding $R^2 \approx 0.7$, which indicates a reasonably strong linear trend supporting the theoretical prediction.
>
> ```python
> import numpy as np
> from sklearn.linear_model import LinearRegression
> from sklearn.metrics import r2_score
> import matplotlib.pyplot as plt
>
> n = np.array([1e3, 10**3.5, 1e4, 10**4.5, 1e5])
> coverage = np.array([0.19, 0.64, 20.28, 73.41, 80.25])
> diff = 95 - coverage
>
> x_reg = 1 / np.sqrt(n)
> x_reg = x_reg.reshape(-1, 1)
> y_reg = diff
>
> reg_model = LinearRegression()
> reg_model.fit(x_reg, y_reg)
> y_pred_reg = reg_model.predict(x_reg)
>
> r2 = r2_score(y_reg, y_pred_reg)
>
> plt.plot(x_reg, y_reg, marker='o')
> plt.xlabel(r'$1/\sqrt{n}$')
> plt.ylabel('Difference from Target Coverage (%)')
> plt.title('Coverage Gap vs $1/\sqrt{n}$')
> plt.tight_layout()
> plt.show()
> ```
>
> Additionally, following your suggestion, we conducted further experiments to study the impact of sequence length ($H$). Keeping the model architecture fixed (dimension $d=8$) and maintaining a consistent missing rate ($16/96$), we varied sequence lengths $H=16, 32, 64, 96, 128$. Due to computational limitations, we again adopted training strategy S1 and missing pattern P4, with $10^5$ training samples. The results are as follows:
>
> | Sequence Length (H)  | 16     | 32     | 64     | 96     | 128    |
> |----------------------|--------|--------|--------|--------|--------|
> | CR Coverage Rate (%) | 92.67  | 88.63  | 82.14  | 80.25  | 77.81  |
>
> These empirical findings also support our theoretical claims in Theorem 2 and Corollary 1, demonstrating how imputation performance scales with both sample size and sequence length.
>
>
> > **Q3**: Definitions of strategies S1–S4, and why they can improve imputation performance.
>
> **A3**:Thank you for pointing out the need for clarification. Here, we explicitly restate the definitions of the mixed-masking strategies S1–S4 used in Section 5:
>
> - **S1**: 100% random missing pattern (16×1, sixteen randomly placed missing entries).
> - **S2**: 50% random (16×1) + 50% weakly grouped (8×2, eight randomly placed blocks of two consecutive missing entries).
> - **S3**: 33.3% random (16×1) + 33.3% weakly grouped (8×2) + 33.3% moderately grouped (4×4, four randomly placed blocks of four consecutive missing entries).
> - **S4**: 25% random (16×1) + 25% weakly grouped (8×2) + 25% moderately grouped (4×4) + 25% strongly grouped (1×16, one randomly placed block of sixteen consecutive missing entries).
>
> Regarding how these strategies relate to our theoretical results, let us denote the training distributions corresponding to S1 and S4 as $P^{(1)} _ {\mathbf{x} _ {\mathrm{obs}}}$ and $P^{(4)} _ {\mathbf{x} _ {\mathrm{obs}}}$, respectively. Consider a test sample $\mathbf{x}^*  _ {\mathrm{obs}}$ following the strongly grouped missing pattern P1 (consecutive missing entries). Intuitively, the resulting distribution $P _ {\mathbf{x}^*  _ {\mathrm{obs}}}$ is closer to $P^{(4)} _ {\mathbf{x} _ {\mathrm{obs}}}$ than to $P^{(1)} _ {\mathbf{x} _ {\mathrm{obs}}}$. Formally, this implies:
>
> $$
> \mathsf{DS}(P _ {\mathbf{x}^*  _ {\mathrm{obs}}}, P^{(4)} _ {\mathbf{x} _ {\mathrm{obs}}}, \mathcal{G}) < \mathsf{DS}(P _ {\mathbf{x}^*  _ {\mathrm{obs}}}, P^{(1)} _ {\mathbf{x} _ {\mathrm{obs}}}, \mathcal{G}).
> $$
>
> Empirically, in Section 5.1, we calculated the average ratio across all test samples and found that:
>
> $$
> \frac{\mathsf{DS}(P _ {\mathbf{x}^*  _ {\mathrm{obs}}}, P^{(1)} _ {\mathbf{x} _ {\mathrm{obs}}}, \mathcal{G})}{\mathsf{DS}(P _ {\mathbf{x}^*  _ {\mathrm{obs}}}, P^{(4)} _ {\mathbf{x} _ {\mathrm{obs}}}, \mathcal{G})} \approx 47.93.
> $$
>
> This clearly indicates that the mixed-masking training strategy (S4) yields significantly smaller distribution-shift coefficients compared to purely random missingness (S1). According to Corollary 1, this provides strong theoretical support for the superior empirical performance achieved by our mixed-masking strategy. We also provide another example with an even larger difference in distribution-shift coefficients (with the ratio approximately 72), which correspondingly leads to a larger performance gap. Please refer to **A4** of reviewer iceu for further details.
>
> > **Q4**: In Line 146, the paper states that the training procedure assumes fully observed training data. However, in real-world scenarios, training data often contains missing values. How would the proposed framework and theoretical analysis adapt to this more realistic setting?
>
> **A4**: Thank you for raising this important point. Practically, TimeDiT [3] proposes effective strategies to handle naturally occurring missing values when applying DiT to time series tasks. Alternatively, methods like those presented in CSDI [4], which fill missing entries with dummy values (e.g., zeros) and treat them as observed, can also be combined with our framework.
>
> From a theoretical perspective, your insight is highly relevant. In fact, our theoretical framework can readily adapt to scenarios involving training data with missing values, provided the missingness follows a "Missing Completely at Random" scheme (i.e., missing patterns do not depend on observed or missing values themselves). Consider Gaussian-process data with Bernoulli masks applied to each entry: since Bernoulli random variables are also sub-Gaussian (light-tailed), our theoretical analysis naturally extends to this setting with minimal adjustments.
>
>
> However, handling cases where missingness depends on observed or even missing values represents a more challenging and promising direction for theoretical advancement. We currently leave this as an open research direction.
>
>
> ## References
>
> [1] Roberts, Stephen, Michael Osborne, Mark Ebden, Steven Reece, Neale Gibson, and Suzanne Aigrain. "Gaussian processes for time-series modelling." *Philosophical Transactions of the Royal Society A: Mathematical, Physical and Engineering Sciences* 371, no. 1984 (2013): 20110550.
>
> [2] Camps-Valls, Gustau, Jochem Verrelst, Jordi Munoz-Mari, Valero Laparra, Fernando Mateo-Jimenez, and Jose Gomez-Dans. "A survey on Gaussian processes for earth-observation data analysis: A comprehensive investigation." *IEEE Geoscience and Remote Sensing Magazine* 4, no. 2 (2016): 58–78.
>
> [3] Cao, Defu, Wen Ye, Yizhou Zhang, and Yan Liu. "TimeDiT: General-purpose diffusion transformers for time series foundation model." *arXiv preprint arXiv:2409.02322* (2024).
>
> [4] Tashiro, Yusuke, Jiaming Song, Yang Song, and Stefano Ermon. "CSDI: Conditional score-based diffusion models for probabilistic time series imputation." *Advances in Neural Information Processing Systems* 34 (2021): 24804–24816.

---

> > ### Comment · Reviewer_F7up · 2025-08-05
> >
> > Thank you for the detailed and thoughtful rebuttal. I appreciate the effort put into clarifying the assumptions underlying the theoretical analysis and providing additional experiments to validate the scaling property.
> >
> > Your responses have addressed most of my concerns, and I will adjust my score accordingly.

---

> > > ### Author Response · Authors · 2025-08-05
> > >
> > > Thank you for your time and valuable suggestions. We’re glad our clarifications and additional experiments addressed your concerns, and we would be happy to discuss further if there are any remaining questions.

---

### Official Review · Reviewer_iceu · 2025-07-02

**Clarity:** 4
**Significance:** 4
**Originality:** 4
**Rating:** 5
**Confidence:** 3

**Summary:**

This paper takes a highly theoretical approach to the problem of imputing missing values in time series using diffusion transformers (DiT). The authors focus on Gaussian process data and derive detailed statistical sample complexity bounds. They also construct confidence regions for the imputations based on theoretical guarantees. Synthetic experiments with varied missing patterns are used to validate the theory and match intuitive expectations. The paper further explores how to mitigate distribution shift between training and testing, improving robustness. Extensive real-world results are included in the supplement, but the core contribution remains largely theoretical.

**Questions:**

1. In Section 5.1, the missing rate is about 17%, and the real data experiments use 10%, 20%, and 50% missingness. Can DiT handle much sparser settings, such as 90% missing?

2. Can DiT be directly applied (or easily modified) to incorporate time-invariant covariates?

3. Are all simulation settings run with the same transformer architecture, regardless of the condition number $\kappa$?

4. In Table 1, under P1 (first row), there is a noticeable gap in coverage between imputation strategies S1 and S2. While this may be due to empirical variation, is there any theoretical intuition or insight that might help explain this discrepancy?

**Ethical Concerns:**

["NO or VERY MINOR ethics concerns only"]

**Final Justification:**

I believe the author has addressed all my questions with high clarity, and the additional numerical results further highlight the significance of this research. As a result, I will maintain my score of 5.

**Limitations:**

yes

**Quality:**

4

**Strengths And Weaknesses:**

Strength:
1. Figure 1 is well-designed and helps clarify the structure of DiT.
2. The paper is theory-heavy, and I’m impressed by the depth and completeness of the analysis.
3. Theorem 1 provides a detailed characterization of score approximation error in terms of transformer architecture, with explicit dependence on the missing pattern via the condition number. This is much more realistic than assuming missing completely at random.
4. The background section (Appendix A) offers a solid overview of time series imputation, which is helpful for non-experts.

Weaknesses:
No major weaknesses observed.

Suggestion:
In Figure 4, consider plotting CSDI first and then DiT, so that DiT appears on top. Since CSDI’s confidence intervals are wider, they visually overshadow DiT, making it hard to see DiT’s advantage in the current version.

---

> ### Author Rebuttal · Authors · 2025-07-30
>
> We sincerely thank you for your positive evaluation of our work, and we greatly appreciate your helpful suggestion on improving Figure 4 to better highlight our results. We now address the specific questions you raised.
>
> > **Q1**: In Section 5.1, the missing rate is about 17%, and the real data experiments use 10%, 20%, and 50% missingness. Can DiT handle much sparser settings, such as 90% missing?
>
> **A1**: Yes. For Section 5.1 (simulated Gaussian process dataset), we intentionally selected a relatively challenging setup—featuring small model size, complex kernels, long sequence length, and high dimensionality—to clearly differentiate the effects of different mixed-masking strategies. With a larger model, more training samples, or simplified kernels and lower dimensionality, we are confident that DiT can handle much higher missing rates, including up to 90%.
>
> Regarding real-world data experiments, we followed your suggestion and conducted additional experiments for both DiT and CSDI under 90% missingness. Below, we report the Mean Absolute Error (MAE) and Mean Squared Error (MSE) results:
>
> **MAE Results (90% Missing Rate)**
>
> | Model              | ETT _ m1                  | BeijingAir              |
> |--------------------|--------------------------|--------------------------|
> | CSDI               | 0.2654 (±0.0297)         | 0.3340 (±0.0410)         |
> | DiT                | **0.2445 (±0.0244)**     | **0.3012 (±0.0247)**     |
>
> **MSE Results (90% Missing Rate)**
>
> | Model              | ETT _ m1                  | BeijingAir              |
> |--------------------|--------------------------|--------------------------|
> | CSDI               | 0.2167 (±0.0286)         | 0.3733 (±0.0552)         |
> | DiT                | **0.1991 (±0.0163)**     | **0.3178 (±0.0326)**     |
>
> These results demonstrate that DiT continues to perform robustly even in extremely sparse settings, achieving better results than CSDI under 90% missingness.
>
>
> > **Q2**: Can DiT be directly applied (or easily modified) to incorporate time-invariant covariates?
>
> **A2**: Yes, we address this question from both a theoretical and a practical perspective.
>
> Time-invariant covariates indeed play a critical role in time series analysis. From a theoretical perspective, time-invariant covariates can be modeled as conditioning variables in the diffusion process, allowing them to be treated similarly to the observed data. As such, only minimal extensions are needed to incorporate them into our current analytical framework.
>
> On the application side, while this extension is not the main focus of our paper, relevant studies such as TimeDiT [1] have demonstrated practical approaches for integrating time-invariant covariates (as conditions) into DiT for time series forecasting and imputation. DiT can be readily adapted to follow similar strategies.
>
>
> > **Q3**: Are all simulation settings run with the same transformer architecture, regardless of the condition number?
>
> **A3**: Yes, as detailed in Appendix F (Experiment details), we consistently employed the same DiT architecture across all conditions, specifically with a hidden size of 256, 12 transformer layers, and 16 attention heads per layer. This ensured comparability across all experiments.
>
> > **Q4**: In Table 1, under P1 (first row), there is a noticeable gap in coverage between imputation strategies S1 and S2. While this may be due to empirical variation, is there any theoretical intuition or insight that might help explain this discrepancy?
>
> **A4**: Thank you for pointing out this intriguing observation. First, we briefly restate the definitions of the mixed-masking strategies S1 and S2:
>
> - **S1**: 100% random (16×1, sixteen fully random placed missing entries)
> - **S2**: 50% random (16×1) + 50% weakly grouped (8×2, eight randomly placed blocks of two consecutive entries)
>
> In row P1 (missing 16 consecutive entries at the end of the sequence, corresponding to a very challenging condition number of 415.4), strategy S2 significantly outperforms S1. Intuitively, training with S2 exposes the model to a combination of simpler (random) and more challenging (weakly grouped) missing patterns, enhancing the model’s ability to handle difficult imputation scenarios. In contrast, the model trained only on simpler random patterns (S1) is less robust against difficult test patterns like P1.
>
> From a theoretical perspective, denoting training distributions corresponding to strategies S1 and S2 as $P^{(1)} _ {\mathbf{x} _ {\mathrm{obs}}}$ and $P^{(2)} _ {\mathbf{x} _ {\mathrm{obs}}}$, respectively, we find that the test distribution $P _ {\mathbf{x}^*  _ {\mathrm{obs}}}$ (with pattern P1) is closer to $P^{(2)} _ {\mathbf{x} _ {\mathrm{obs}}}$. Empirically, we approximately computed the distribution-shift coefficient ratio over all test samples with pattern P1 and obtained:
>
> $$
> \frac{\mathsf{DS}(P _ {\mathbf{x}^*  _ {\mathrm{obs}}}, P^{(1)} _ {\mathbf{x} _ {\mathrm{obs}}}, \mathcal{G})}{\mathsf{DS}(P _ {\mathbf{x}^*  _ {\mathrm{obs}}}, P^{(2)} _ {\mathbf{x} _ {\mathrm{obs}}}, \mathcal{G})} \approx 72.46.
> $$
>
> Thus, the training distribution under strategy S2 yields a substantially lower distribution-shift coefficient compared to S1 for the test pattern P1. According to Corollary 1, this theoretically confirms our empirical observation that S2 significantly improves imputation performance over S1 for challenging missing patterns.
>
> ## References
>
> [1] Cao, Defu, Wen Ye, Yizhou Zhang, and Yan Liu. "TimeDiT: General-purpose diffusion transformers for time series foundation model." *arXiv preprint arXiv:2409.02322* (2024).

---

> > ### Comment · Reviewer_iceu · 2025-08-03
> >
> > I believe the author has addressed all my questions with high clarity, and the additional numerical results further highlight the significance of this research. As a result, I will maintain my score of 5.

---

> > > ### Author Response · Authors · 2025-08-03
> > >
> > > Thank you for taking the time to read our rebuttal. We truly appreciate your recognition of our additional results, and thank you again for the thoughtful suggestions.

---

### Official Review · Reviewer_PUbh · 2025-07-02

**Clarity:** 2
**Significance:** 3
**Originality:** 3
**Rating:** 4
**Confidence:** 3

**Summary:**

This paper addresses the problem of imputing missing values in time-series data using diffusion-based generative models, specifically focusing on the diffusion transformers (DiT) method. This work contributes a theoretical analysis of their statistical properties under Gaussian Process data. The paper provides sample complexity bounds for the imputation of the DiT architecture. Based on the analysis, confidence regions for quantifying uncertainty in the imputed values can be derived. The analysis also reveals that missingness patterns critically affect both the efficiency and accuracy of imputation. In addition to the theory, the paper proposes a mixed-masking training strategy to improve imputation quality and supports the theoretical findings with an evaluation using synthetic data.

**Questions:**

1. Clarification on Benchmark Experiments
Appendix F presents experimental results on benchmark datasets, which seem to largely confirm prior findings that DiT outperforms CSDI. Could the authors clarify what new insights are gained from this particular experiment in the context of the paper’s theoretical contributions? Additionally, have the authors considered applying the mixed-masking strategy to these benchmark experiments as well? It would be interesting to see whether the proposed strategy generalizes beyond the synthetic setting.
2. Purpose of Algorithm 2
If I understand correctly, Algorithm 2 is introduced solely for analytical purposes and is not used in practice. Is this correct?
3. Connection Between Condition Numbers and Mixed-Masking
It is insightful that the paper identifies the difficulty of imputation as being influenced by the condition numbers of Σcond and Σobs. However, the link between this and the effectiveness of the mixed-masking strategy remains unclear. While it is intuitive that training on a variety of missing patterns might enhance robustness to unseen test patterns, I did not see a formal theoretical justification showing that mixing patterns with different condition numbers leads to more effective training. Could the authors clarify whether such a result exists or whether this remains an empirical hypothesis?
4. Coverage of Masking Pattern Combinations
In the synthetic experiments, not all combinations of S1-S4 were explored. While it is understandable that exhaustive evaluation would be too costly, to support the claim that combining different masking patterns is beneficial, it would be helpful to also report results from training on individual subsets, that is, using only S2/S3/S4. This would make the benefits of combining patterns more evident and quantifiable.

**Ethical Concerns:**

["NO or VERY MINOR ethics concerns only"]

**Final Justification:**

Additional results in the rebuttal clarified my concern on the effectiveness of mixed masking strategy and clarified my misunderstanding on the definition of S1-S4 of the training strategies. I increased the score to 4.

**Limitations:**

Not so many limitations were discussed. It might be useful to include more discussion, for example, the limitations of the theoretical analysis compared with the real use cases, and the need to rely on assuming Gaussian process data.

**Quality:**

2

**Strengths And Weaknesses:**

Strengths
1. The paper addresses a highly relevant problem in time-series imputation using deep learning, an area where theoretical understanding remains limited.
2. The authors present a range of theoretical results, which yield practical insights. For example, based on the condition number of the covariance matrix, we can interpret why one missing pattern is more difficult than the other. Convergence rate is also derived to provide a good theoretical grounding of DiT and also its uncertainty quantification.

Weaknesses
1. The experiments are primarily conducted on a Gaussian Process dataset that satisfies the model’s assumptions. While Appendix F includes benchmark experiments demonstrating the superiority of DiT over CSDI, these results do not directly support the effectiveness of the proposed mixed-masking training strategy, as DiT itself is not introduced in this paper.
2. While it is encouraging that the results support the use of transformer architectures, the theoretical argument does not appear strong enough to imply that CSDI (Conditional Score-based Diffusion Imputer) would be inherently less effective unless the paper explicitly shows structural limitations of CSDI compared to transformers.
3. The work does not propose a novel algorithm for diffusion-based imputation. It focuses solely on the theoretical analysis of an existing DiT model under GP assumptions.
4. The paper lacks a related work section in the main body, making it overly dependent on the supplementary material and limiting its self-contained readability.

While I appreciate the authors' effort in deriving several theoretical results for DiT, I have several concerns that currently lead me to assign a borderline (slightly negative) rating. Please refer to the weaknesses and questions sections for more information.

---

> ### Author Rebuttal · Authors · 2025-07-30
>
> Thank you for your insightful comments and constructive suggestions. We now address the specific questions, weaknesses, and limitations you have raised.
>
> > **Q1**: Assumptions on Gaussian process data.
>
> **A1**: We impose Gaussian process (GP) assumption only for theoretical analysis. The insights gained from the analysis motivate our mixed-masking strategy, which applies to real-world data well beyond the GP assumption.
>
> As discussed in the introduction, GPs naturally exhibit rich spatio-temporal dependencies and long-horizon correlations, which pose central challenges in time series tasks [1] and are widely adopted in practical modeling scenarios such as climate [2] and financial forecasting.  Additionally, as noted in lines 119–121, we intentionally do not restrict the kernel function for GPs, ensuring our theoretical results remain generalizable across various settings.
>
> Empirical validation in Section 5.2 (latent GP data) and the Appendix (real-world datasets) further supports the applicability of our findings beyond strictly synthetic settings. Also worth noting is that although our theory is developed under GP assumptions, the insights gained from this theory directly informed the development of our mixed-masking strategy. This strategy has demonstrated strong generalizability to real-world applications, clearly indicating that the theoretical assumptions do not limit the practical applicability of our method.
>
> > **Q2**:  The theoretical argument does not appear strong enough to imply that CSDI would be inherently less effective.
>
> **A2**: Thank you for raising this point. Theorem 1 demonstrates how transformers capture temporal correlations effectively through the attention mechanism (Theorem 1). While convolutional architectures used by CSDI may also model temporal correlations, theoretical analysis of their efficiency is elusive. Recent study [3] suggests that transformers are inherently suitable for sequential modeling, though rigorously proving DiT’s superiority over CSDI remains open and challenging.
>
> >**Q3**: What are the new insights gained from the real-world data experiment in Appendix F?  Additionally, have the authors considered applying the mixed-masking strategy to these benchmark experiments?
>
> **A3**: Thank you for this valuable suggestion. The original Appendix F experiments primarily confirmed DiT’s superior imputation performance, supporting the theoretical result from Theorem 2 . Following your advice, we conducted additional experiments to specifically validate our mixed-masking strategy on real-world benchmark datasets. In these experiments, we designed mixed-masking strategies inspired by those described in Section 5 (patterns tend to have more grouped missing values rather than purely random). The updated Mean Squared Error results are shown below:
>
> | Model| **ETT _ m1 (Missing %)** ||| **BeijingAir (Missing %)** |||
> |-|-|-|-|-|-|-|
> | | 10% | 20%| 50%| 10%  | 20%| 50% |
> | CSDI                              | 0.0615 (±0.0097)       | 0.0698 (±0.0106)     | 0.0797 (±0.0106)     | 0.4196 (±0.1726)           | 0.3926 (±0.0790)     | 0.4534 (±0.0379)     |
> | GP-VAE                            | 0.1567 (±0.0094)       | 0.2138 (±0.0067)     | 0.4249 (±0.0127)     | 0.4096 (±0.0202)           | 0.4777 (±0.0179)     | 0.7017 (±0.0189)     |
> | **DiT**                           | 0.0534 (±0.0063)       | 0.0606 (±0.0076)     | **0.0684 (±0.0070)** | 0.3683 (±0.0351)           | 0.4025 (±0.0424)     | 0.4255 (±0.0670)     |
> | **DiT w/ mixed-masking strategy** | **0.0502 (±0.0055)**   | **0.0588 (±0.0081)** | 0.0711 (±0.0092)      | **0.3428 (±0.0275)**       | **0.3864 (±0.0403)** | **0.4229 (±0.0539)** |
>
> As can be seen, incorporating our proposed mixed-masking strategy further improves DiT's performance on real-world datasets with varying missing rates, supporting our theoretical and methodological contributions.
>
> > **Q4**: Is Algorithm 2 introduced solely for analytical purposes and is not used in practice?
>
> **A4**: Yes, Algorithm 2 serves for analytical purposes to understand how transformers unroll an algorithm to represent the score function. Our theoretical guarantees in Lemma 1, Theorem 1 and Theorem 2 are built upon the insights from this algorithm.
>
> > **Q5**: Connection between condition numbers and mixed-masking strategies.
>
> **A5**: Thank you for highlighting this important aspect.  First, we clarify the definitions of mixed-masking strategies S1–S4 used in Section 5:
>
> - **S1**: 100% random (16×1, sixteen fully random placed missing entries)
> - **S2**: 50% random (16×1) + 50% weakly grouped (8×2, eight randomly placed blocks of two consecutive entries)
> - **S3**: 33.3% random (16×1) + 33.3% weakly grouped (8×2) + 33.3% moderately grouped (4×4, four randomly placed blocks of four consecutive entries)
> - **S4**: 25% random (16×1) + 25% weakly grouped (8×2) + 25% moderately grouped (4×4) + 25% strongly grouped (1×16, one randomly placed blocks of sixteen consecutive entries)
>
> Our theoretical analysis indeed strongly supports the effectiveness of the mixed-masking strategy by linking it directly to the distribution-shift coefficient $\mathsf{DS}$ (Corollary 1). Using mixed-masking strategies reduces $\mathsf{DS}$, directly improving imputation performance.
>
> To illustrate this clearly, consider the concrete example in our numerical simulations (Section 5). Missing patterns P1–P4 have varying difficulties , with P1 (strongly grouped missing) yielding the highest condition number and P4 (purely random missingness) having the lowest. Specifically, our training strategies S1 and S4 differ as follows:
>
> - **S1** uses only random missing patterns during training (P4-like, low condition number).
> - **S4** includes various missing patterns (both P1-like and P4-like, high and low condition numbers).
>
> Denoting the training distributions corresponding to S1 and S4 as $P^{(1)} _ {\mathbf{x} _ {\mathrm{obs}}}$ and $P^{(4)} _ {\mathbf{x} _ {\mathrm{obs}}}$, respectively, if we consider a new sample $\mathbf{x}^*  _ {\mathrm{obs}}$ with missing pattern P1 (strongly grouped missing), it is intuitive that the resulting distribution $P _ {\mathbf{x}^*  _ {\mathrm{obs}}}$ is closer to $P^{(4)} _ {\mathbf{x} _ {\mathrm{obs}}}$. Correspondingly, we have:
>
> $$
> \mathsf{DS}(P _ {\mathbf{x}^*  _ {\mathrm{obs}}}, P^{(4)} _ {\mathbf{x} _ {\mathrm{obs}}}, \mathcal{G}) < \mathsf{DS}(P _ {\mathbf{x}^*  _ {\mathrm{obs}}}, P^{(1)} _ {\mathbf{x} _ {\mathrm{obs}}}, \mathcal{G}).
> $$
> Empirically, in Section 5.1, we approximately computed the average ratio over all test samples $\mathbf{x}^*  _ {\mathrm{obs}}$ and obtained:
>
> $$
> \frac{\mathsf{DS}(P _ {\mathbf{x}^*  _ {\mathrm{obs}}}, P^{(1)} _ {\mathbf{x} _ {\mathrm{obs}}}, \mathcal{G})}{\mathsf{DS}(P _ {\mathbf{x}^* _ {\mathrm{obs}}}, P^{(4)} _ {\mathbf{x} _ {\mathrm{obs}}}, \mathcal{G})} \approx 47.93.
> $$
> This clearly demonstrates that the mixed-masking training strategy (S4) yields significantly smaller distribution-shift coefficients compared to purely random missingness (S1). According to Corollary 1, this strongly supports the superior imputation performance achieved by our mixed-masking strategy. We also provide another example with an even larger difference in distribution-shift coefficients (with the ratio approximately 72), which correspondingly leads to a larger performance gap. Please refer to **A4** of reviewer iceu for further details.
>
> > **Q6**: Not all combinations of masking patterns were explored; it would be helpful to also report results from training on individual subsets, that is, using only S2/S3/S4.
>
> **A6**: Thank you for suggesting additional masking-pattern evaluations.
>
> For the definition of the mixed-masking strategies, please refer to **A5**. We agree that evaluating the individual patterns (8×2, 4×4, and 1×16 separately) would provide useful insights. Following your suggestion, we performed these additional experiments under the Gaussian process data setting. The updated confidence region coverage (%) results are shown below:
>
> |   | S1 | S2 | S3 | S4 | Only 8×2 | Only 4×4 | Only 1×16 |
> |-|-|-|-|-|-|-|-|
> |P1  |34.58(±1.22)|**66.22(±3.86)**|56.04(±6.48)   |57.27(±5.34)|36.74(±1.31)|34.15(±1.16)|32.68(±1.50)|
> |P2  |58.46(±1.89)|**83.71(±2.86)**|81.05(±2.09)   |79.00(±2.42)|60.51(±1.65)|59.23(±1.88)|54.23(±1.76)|
> |P3  |72.42(±1.66)|74.04(±1.90)   |**74.59(±1.27)**|74.38(±3.00)|71.24(±1.52)|73.08(±1.10)|69.46(±1.53)|
> |P4  |80.25(±1.64)|81.50(±2.12)   |**83.09(±1.48)**|82.74(±2.40)|80.46(±2.01)|79.83(±1.84)|76.72(±2.20)|
>
> These results clearly illustrate that training exclusively on single masking patterns, regardless of their complexity, yields inferior imputation performance compared to appropriately mixing different patterns. This supports our proposed mixed-masking strategies and aligns well with our theoretical insights.
>
>
>
> ## References
>
> [1] Roberts, Stephen, Michael Osborne, Mark Ebden, Steven Reece, Neale Gibson, and Suzanne Aigrain. "Gaussian processes for time-series modelling." *Philosophical Transactions of the Royal Society A: Mathematical, Physical and Engineering Sciences* 371, no. 1984 (2013): 20110550.
>
> [2] Camps-Valls, Gustau, Jochem Verrelst, Jordi Munoz-Mari, Valero Laparra, Fernando Mateo-Jimenez, and Jose Gomez-Dans. "A survey on Gaussian processes for earth-observation data analysis: A comprehensive investigation." *IEEE Geoscience and Remote Sensing Magazine* 4, no. 2 (2016): 58-78.
>
> [3] Wen, Qingsong, Tian Zhou, Chaoli Zhang, Weiqi Chen, Ziqing Ma, Junchi Yan, and Liang Sun. "Transformers in time series: A survey." *arXiv preprint arXiv:2202.07125* (2022).

---

> > ### Comment · Reviewer_PUbh · 2025-08-03
> > **Thank you for the rebuttal**
> >
> > I appreciate the authors for responding to my concerns in detail. Additional results in the rebuttal clarified my concern on the effectiveness of the mixed masking strategy and clarified my misunderstanding of the definition of S1-S4 of the training strategies. I increased the score to 4. It is reassuring to see that the mixed masking strategy can also enhance the benchmark experiment result of DiT.

---

> > > ### Author Response · Authors · 2025-08-03
> > >
> > > Thank you for taking the time to read our rebuttal and for your thoughtful feedback. We truly appreciate your recognition of our additional results and your updated assessment.

---

### Official Review · Reviewer_i349 · 2025-07-02

**Clarity:** 2
**Significance:** 3
**Originality:** 3
**Rating:** 5
**Confidence:** 3

**Summary:**

This paper addresses the problem of theoretically understanding the effectiveness of score-based diffusion models for imputation. Under a Gaussian data assumption, the authors propose a nested optimization algorithm to approximate conditional scores and show that a Transformer can emulate this process via a tailored architecture. A key contribution is the explicit modeling of how missingness patterns affect score estimation. The paper introduces tools such as distribution shift measures and confidence regions to quantify and mitigate this effect. Theoretical guarantees are provided, with non-asymptotic error bounds that scale with the condition number and missing data structure.

**Questions:**

- The conclusion in Remark 1 is not immediately obvious. Could the authors provide further clarification or intuition behind this result?

**Ethical Concerns:**

["NO or VERY MINOR ethics concerns only"]

**Final Justification:**

After reading the authors’ detailed rebuttal and considering the clarifications provided, I am satisfied that my initial concerns have been fully addressed. The authors offered further intuition behind Remark 1, clarified the treatment of GP data, and acknowledged the modeling limitations associated with Gaussian assumptions. These clarifications, along with a more nuanced discussion of related work, have increased my confidence in the paper. While the theoretical exposition in Sections 3 and 4 could still benefit from improved clarity, the paper makes a solid theoretical and algorithmic contribution to the field of imputation with diffusion models. I maintain my rating and recommend acceptance.

**Limitations:**

While the authors acknowledge the assumption of Gaussianity in their analysis, the implications of this modeling choice merit further discussion. The assumption enables theoretical tractability but may limit applicability to real-world data distributions, which are often multimodal or heavy-tailed.

No ethical or broader societal risks are apparent from the method itself.

**Paper Formatting Concerns:**

I did not notice any formatting issues.

**Quality:**

3

**Strengths And Weaknesses:**

## Strengths
- The paper shows that Transformers can approximate conditional score functions through unrolled optimization, supported by a constructive proof (Theorem 1).
- The proposed score approximation under Gaussian assumptions is elegant, despite the simplified distributional setting.
- The method explicitly accounts for the impact of missing patterns and introduces tools like confidence regions and shift metrics. The method explicitly accounts for the impact of missing patterns and introduces tools like confidence regions and shift metrics.
- Theoretical analysis provides non-asymptotic error bounds, which scale with the condition number and model parameters.

## Weaknesses
- While not a major weakness, the paper overlooks several relevant works on missing data imputation using GANs and VAEs. The claim that these models “often spell limitations in expressiveness or training stability” is overly broad and insufficiently supported. Recent methods [1–3] demonstrate that VAE- and GAN-based approaches can be both efficient and accurate. A more balanced discussion would improve the positioning of this work.
- Clarity and flow could be improved, particularly in Sections 3 and 4, where the theoretical framework and algorithmic insights are harder to follow.

## References
[1] Yoon, Jinsung, James Jordon, and Mihaela Schaar. "Gain: Missing data imputation using generative adversarial nets." International conference on machine learning. PMLR, 2018.

[2] Mattei, Pierre-Alexandre, and Jes Frellsen. "MIWAE: Deep generative modelling and imputation of incomplete data sets." International conference on machine learning. PMLR, 2019.

[3] Peis, Ignacio, Chao Ma, and José Miguel Hernández-Lobato. "Missing data imputation and acquisition with deep hierarchical models and hamiltonian monte carlo." Advances in Neural Information Processing Systems 35 (2022): 35839-35851.

---

> ### Author Rebuttal · Authors · 2025-07-30
>
> Thank you for the valuable and constructive comments. We respond to the specific questions, weaknesses, and limitations you have raised in the following.
>
> > **Q1**: Discussions on GAN- and VAE-based methods.
>
> **A1**: While our main focus is on diffusion-based imputation, we appreciate the references to recent GAN- and VAE-based methods [1–3]. We will cite and discuss these works in the Related Work section to present a more balanced view of expressiveness and training stability. We also note that we have already compared our method with a VAE-based approach (GP-VAE) in our experiments. Through our experiments on real-world and synthetic datasets, we observe that diffusion-based methods outperform GP-VAE. Additionally, thank you for highlighting clarity and flow issues in Sections 3 and 4; we will revise these sections to ensure the theoretical framework and algorithmic insights are presented more smoothly.
>
>
> > **Q2**: Assumptions on Gaussian process data, and heavy-tail data extension.
>
> **A2**: We appreciate the limitations you pointed out. Indeed, our theoretical analysis assumes Gaussian process (GP) data. However, as discussed in the introduction, GPs naturally exhibit rich spatio-temporal dependencies and long-horizon correlations, which pose central challenges in time series tasks [4] and are widely adopted in practical modeling scenarios such as climate [5] and financial forecasting. Additionally, as noted in lines 119–121, we allow for a wide range of kernel functions for GPs, ensuring our theoretical results remain generalizable across various settings. Empirical validation in Section 5.2 (latent GP data) and the Appendix (real-world datasets) further supports the applicability of our theoretical findings beyond the synthetic GP settings. It is also worth noting that although our theory is developed under GP assumptions, the insights gained from the theory motivate the development of our mixed-masking strategy. This strategy has demonstrated strong generalizability to real-world applications, clearly indicating that the theoretical assumptions do not limit the practical applicability of our method.
>
> Your insightful point regarding heavy-tailed data distributions common in real-world scenarios is particularly valuable and promising. Yet we suspect that diffusion-based models are relatively weak in capturing heavy-tailed data distributions accurately without careful modifications. Therefore, we leave it as an interesting and meaningful direction for future research.
>
>
>
>
> > **Q3**:  The conclusion in Remark 1 is not immediately obvious. Could the authors provide further clarification or intuition behind this result?
>
> **A3**:  Remark 1 builds directly on Corollary 1 by leveraging the distribution-shift coefficient $\mathsf{DS}$ (defined in Definition 1, measuring how much the test sample deviates from the training distribution). Corollary 1 shows that a large $\mathsf{DS}$ can lead to poor imputation outcomes. Remark 1 explains under **what conditions** $\mathsf{DS}$ becomes large (or small) and **how** we can benefit from this insight:
>
> - **Under what conditions $\mathsf{DS}$ is large:**
>   If $\mathbf{x}^*  _ {\mathrm{obs}}$ lies in the region where the training distribution has a weak coverage (e.g., hardly seen similar examples in the training data), then $\mathsf{DS}$ becomes large, making imputation difficult.
>
> - **How we leverage this insight:**
>   Different missing patterns during training lead to different training distributions $P _ {\mathbf{x} _ {\mathrm{obs}}}$, resulting in varying condition numbers and consequently different $\mathsf{DS}$ values. Training with diverse missing patterns—ranging from easy to hard—helps the model adapt to imputation tasks with varying levels of difficulty by effectively covering more scenarios.
>
> To illustrate this clearly, consider the concrete example in our numerical simulation in Section 5. Missing patterns P1–P4 have varying difficulties (estimated by condition numbers), with P1 (most clustered missingness) yielding the highest condition number and P4 (completely random missingness) having the lowest. Specifically, our training strategies S1 and S4 differ as follows:
>
> - **S1** uses only random missing patterns during training (P4-like, low condition number).
> - **S4** includes various missing patterns (both P1-like and P4-like, high and low condition numbers).
>
> Denoting the training distributions corresponding to S1 and S4 as $P^{(1)} _ {\mathbf{x} _ {\mathrm{obs}}}$ and $P^{(4)} _ {\mathbf{x} _ {\mathrm{obs}}}$, respectively, if we consider a new sample $\mathbf{x}^*  _ {\mathrm{obs}}$ with missing pattern P1, it is intuitive that the resulting distribution $P _ {\mathbf{x}^*  _ {\mathrm{obs}}}$ is closer to $P^{(4)} _ {\mathbf{x} _ {\mathrm{obs}}}$. Hence, we have:
>
> $$
> \mathsf{DS}(P _ {\mathbf{x}^*  _ {\mathrm{obs}}}, P^{(4)} _ {\mathbf{x} _ {\mathrm{obs}}}, \mathcal{G}) < \mathsf{DS}(P _ {\mathbf{x}^*  _ {\mathrm{obs}}}, P^{(1)} _ {\mathbf{x} _ {\mathrm{obs}}}, \mathcal{G}).
> $$
> Empirically, in Section 5.1, we approximately computed the average ratio over all test samples $\mathbf{x}^*  _ {\mathrm{obs}}$ and obtained:
>
> $$
> \frac{\mathsf{DS}(P _ {\mathbf{x}^*  _ {\mathrm{obs}}}, P^{(1)} _ {\mathbf{x} _ {\mathrm{obs}}}, \mathcal{G})}{\mathsf{DS}(P _ {\mathbf{x}^*  _ {\mathrm{obs}}}, P^{(4)} _ {\mathbf{x} _ {\mathrm{obs}}}, \mathcal{G})} \approx 47.93.
> $$
> This clearly demonstrates that the mixed-masking training strategy (S4) yields significantly smaller distribution-shift coefficients compared to purely random missingness (S1). According to Corollary 1, this strongly supports the superior imputation performance achieved by our mixed-masking strategy. We also provide another example with an even larger difference in distribution-shift coefficients (with the ratio approximately 72), which correspondingly leads to a larger performance gap. Please refer to **A4** of reviewer iceu for further details.
>
>
>
> ## References
> [1] Yoon, Jinsung, James Jordon, and Mihaela Schaar. "Gain: Missing data imputation using generative adversarial nets." International conference on machine learning. PMLR, 2018.
>
> [2] Mattei, Pierre-Alexandre, and Jes Frellsen. "MIWAE: Deep generative modelling and imputation of incomplete data sets." International conference on machine learning. PMLR, 2019.
>
> [3] Peis, Ignacio, Chao Ma, and José Miguel Hernández-Lobato. "Missing data imputation and acquisition with deep hierarchical models and hamiltonian monte carlo." Advances in Neural Information Processing Systems 35 (2022): 35839-35851.
>
> [4] Roberts, Stephen, Michael Osborne, Mark Ebden, Steven Reece, Neale Gibson, and Suzanne Aigrain. "Gaussian processes for time-series modelling." *Philosophical Transactions of the Royal Society A: Mathematical, Physical and Engineering Sciences* 371, no. 1984 (2013): 20110550.
>
> [5] Camps-Valls, Gustau, Jochem Verrelst, Jordi Munoz-Mari, Valero Laparra, Fernando Mateo-Jimenez, and Jose Gomez-Dans. "A survey on Gaussian processes for earth-observation data analysis: A comprehensive investigation." *IEEE Geoscience and Remote Sensing Magazine* 4, no. 2 (2016): 58-78.

---

> > ### Comment · Reviewer_i349 · 2025-08-04
> >
> > Thank you for the detailed and thoughtful rebuttal. I appreciate the authors’ clarifications on the theoretical assumptions, the assumptions on GP data, and the additional context provided through new references and explanations—particularly regarding Remark 1. These responses have fully addressed my concerns, and I have accordingly increased my confidence score and will recommend acceptance.

---

> > > ### Author Response · Authors · 2025-08-04
> > >
> > > Thank you for taking the time to read our rebuttal and your thoughtful suggestions on potential discussions. We truly appreciate your recognition of our additional results and explanations regarding Remark 1.

---

### Author Response · Authors · 2025-08-07

We sincerely thank all reviewers for their thoughtful feedback, constructive suggestions, and valuable discussions throughout the review process. Your insights have helped us significantly improve the clarity, empirical validation, and theoretical exposition of our work.

We also extend our thanks to the Area Chair and the reviewers once again for the upcoming AC-reviewer discussion period. We appreciate your time and effort in evaluating our submission.

---

### Decision · Program_Chairs · 2025-09-17

**Decision:**

Accept (poster)

**Comment:**

The paper develops a statistical framework for capturing statistical efficiency and uncertainty of diffusion-based time-series imputation with conditional diffusion transformers (DiT). All reviewers (and area chair) found the paper investigates an important problem in time-series imputation with diffusion models that can lack theoretical guarantees, where the paper provides theory and the architectural insight with clear intuition, and the experimental results highlight the significance of the proposed work.

During rebuttal, the authors provided clarifications on intuition for Remark 1 and its link to the shift coefficient, and a more balanced discussion of VAE/GAN work and existing GP-VAE comparisons. The authors provided additional real-world experiments with high (i.e., 90%) missingness, and mixed-masking strategies, to support DiT’s robustness and exhibited empirical trends. These additions strengthen the paper and resolve the main concerns by reviewers. The reviewers also left some important notes on improving parts of the paper, including theoretical exposition in Sections 3 and 4. We highly encourage the authors to more clearly discuss those points and include the additional analysis in revised paper.

The potential limitation pointed by reviewer is the GP assumption bounds’ formal generality, though classified by authors it is well-motivated and clearly acknowledged. We hope the authors clearly describe the potential limitations of their assumptions, while we still agree the experiments, algorithmic and structure design well support the contribution of this paper to related domains.